# Global ocean dimethyl sulfide climatology estimated from observations and an artificial neural network

Wei-Lei Wang[1], Guisheng Song[2], François Primeau[1], Eric S. Saltzman[1,3], Thomas G. Bell[1,4], and J. Keith Moore[1]

[1]Department of Earth System Science, University of California at Irvine, Irvine, California, USA
[2]School of Marine Science and Technology, Tianjin University, Tianjin, 300072, China
[3]Department of Chemistry, University of California at Irvine, Irvine, California, USA
[4]Plymouth Marine Laboratory, Prospect Place, Plymouth, PL1 3DH, UK

**Correspondence:** Wei-Lei Wang (weilei.wang@gmail.com)

**Abstract.** Marine dimethyl sulfide (DMS) is important to climate due to the ability of DMS to alter Earth's radiation budget. Knowledge of the global-scale distribution, seasonal variability, and sea-to-air flux of DMS is needed in order to improve understanding of atmospheric sulfur, aerosol/cloud dynamics and albedo. Here we examine the use of an artificial neural network (ANN) to extrapolate available DMS measurements to the global ocean and produce a global climatology with monthly temporal resolution. A global database of 82,996 ship-based DMS measurements in surface waters was used along with a suite of environmental parameters consisting of latitude-longitude coordinates, time-of-day, time-of-year, solar radiation, mixed layer depth, sea surface temperature, salinity, nitrate, phosphate, and silicate. Linear regressions of DMS against the environmental parameters show that on a global scale mixed layer depth and solar radiation are the strongest predictors of DMS. These parameters capture ∼9% and ∼7% of the raw DMS data variance, respectively. Multi-linear regression can capture more of the raw data variance (∼39%), but strongly underestimates DMS in high concentrations regions. In contrast, the artificial neural network captures ∼66% of the raw data variance in our database. Like prior climatologies our results show a strong seasonal cycle in surface ocean DMS with highest concentrations and sea to air fluxes in the high-latitude summertime oceans. We estimate a lower global sea-to-air DMS flux ($20.12 \pm 0.43$ Tg S yr$^{-1}$) than the prior estimate based on a map interpolation method (Lana et al., 2011) when the same gas transfer velocity parameterization is used. Our sensitivity test results show that DMS concentration does not change unidirectionally with each of the environmental parameters, which emphasizes the interactions among these parameters. The ANN model suggests that the flux of DMS from the ocean to the atmosphere will increase with global warming. Given that larger DMS fluxes induce greater cloud albedo, this corresponds to a negative climate feedback.

## 1 Introduction

Dimethyl sulfide emitted from the surface ocean is the major precursor for aerosol sulfate in the marine atmosphere. These aerosols play a significant role in the climate system both directly, through aerosol radiative effects and indirectly, through their role as cloud condensation nuclei and influence on cloud radiative properties (Andreae and Rosenfeld, 2008). Assessing the impact of DMS on global climate requires an understanding of the seawater DMS distribution and the factors controlling

variability on a variety of spatial and temporal scales. Dimethyl sulfide is produced in surface waters, mainly via enzymatic cleavage of the biogenic compound dimethyl sulfoniopropionate (DMSP; (e.g. Stefels et al., 2007)). The abundance of DMS in surface waters is a function of numerous factors controlling production, loss rates, and pathways of both DMSP and DMS (Simó, 2001; Toole and Siegel, 2004; Galí et al., 2015). Developing mechanistic and predictive models of surface ocean DMS is challenging due to limitations of the existing observational database and process rate measurements.

Given the biogenic origin of DMS, early efforts focused on the relationship between DMS and Chl a (a proxy for biomass). Positive correlations between DMS and Chl *a* have been reported on basin scales (e.g. Andreae and Barnard, 1984; Yang et al., 1999). However, this positive correlation disappears when more data are used. Kettle et al. (1999) found no significant relationship between DMS and Chl *a* based on the global DMS data set available at the time. The weak relationship may be caused by the so-called "summer DMS paradox", which describes a phenomenon that annual maximum of surface DMS concentration is commonly detected in summer when Chl *a* is at its annual minimum in mid and subtropical low latitude waters (Simó and Pedrós-Alió, 1999). Kettle et al. (1999) also tested linear regression models on a compilation of data, including sea surface salinity and temperature, nitrate, silicate, phosphate, and Chl *a*. The authors then concluded that no simple algorithm based on linear regression could be used to create monthly DMS fields, indicating that more complex mechanisms can control surface DMS concentrations.

Simó and Dachs (2002) achieved a strong linear relationship between heavily binned/averaged DMS and mixed layer depth (MLD) when Chl-$a$/MLD $\geq$ 0.02, and a logarithmic relationship between DMS and Chl-$a$/MLD when Chl-$a$/MLD $<$ 0.02. Vallina and Simó (2007) found a linear relationship between DMS concentration and solar radiation dose (SRD) in the coastal northwestern Mediterranean. They conducted a global scale study by dividing the ocean into $10°$ latitude by $20°$ longitude boxes and correlating SRD and the box averaged DMS concentration. A strong linear relationship was detected in this filtered dataset. Derevianko et al. (2009) reexamined the relationship between SRD/MLD and DMS concentration by using $1°$ by $1°$ bins, and found that only a small fraction (14%) of the DMS variance was captured by a linear model based on SRD or MLD. These authors also pointed out that the previously identified strong relationship between MLD/SRD and DMS "results from the reduction in the total variance in the data due to binning" (Derevianko et al., 2009).

Prognostic models have also been used to obtain climatological DMS distributions. In these models, phytoplankton are divided into different groups based on their ability to produce DMSP, the precursor of DMS. For example, diatoms produce less DMS than coccolithophores and *Phaeocystis* (e.g. Bopp et al., 2003; Vogt et al., 2010; Gypens et al., 2014). Elliott (2009) implicitly incorporated *Phaeocystis* in a model by assuming that DMS yields are simply related to temperature. The work of Wang et al. (2015) explicitly incorporated *Phaeocystis* into the Biogeochemical Elemental Cycling (BEC) model and included DMSP production from each phytoplankton group, along with DMS leakage pathways from algal cells, (grazing, lysis, and exudation). Despite this level of modeling detail, there are still large discrepancies between the model simulations and in situ measurements (Tesdal et al., 2016). Le Clainche et al. (2010) suggested that environmental conditions should be included in future model development because DMS cycling depends strongly on phytoplankton dynamics.

The DMS climatologies used in most climate models were obtained by extrapolating observed DMS to the global ocean using objective analysis schemes (Kettle et al., 1999; Lana et al., 2011). In those climatologies, observational data were first

binned and averaged into 1° by 1° grid squares, which were then grouped into 57 static biogeographic provinces according to Longhurst (1998). Many provinces lacked adequate data to create a reliable climatology (Fig. A1). In those situations, they first generated an annual cycle with monthly means for each province. Temporal interpolations were used to fill the monthly gaps if there were enough data to create a robust annual mean. Otherwise, weighted interpolation from neighboring provinces was used to fill the remaining gaps. Major gaps remain in the observational data base for wintertime in the high latitudes of both hemispheres.

Machine learning is being increasingly used in oceanography and geoscience studies (Bergen et al., 2019). For example, Roshan and DeVries (2017) applied an artificial neural network (ANN) to extrapolate observed dissolved organic carbon (DOC) to the global ocean. Rafter et al. (2019) used an ensemble of neural networks to study oceanic $\delta^{15}$N distribution. ANNs have also been used to study DMS on regional scales (e.g. Humphries et al., 2012). The popularity of machine learning partially stems from one of its inherent advantages: it can detect non-linear relationships that traditional linear regression models are unable to capture. In this study, we explore the relationships between DMS and environmental parameters using a machine learning method. Such relationships are hard to detect using traditional linear regression methods, because environmental parameters do not directly influence DMS concentration. They control the distribution of marine algae that determines the distribution of DMSP (a precursor of DMS) and its conversion to DMS (Kiene et al., 2000; Simó, 2001). The objective of this paper is to discover the relationships between DMS and environmental variables, with the goal of constructing a novel monthly-resolved DMS climatology.

The paper is organized as follows. We begin by exploring the relationships between DMS concentration and various environmental parameters taken one at a time using linear regression. We then do a stepwise multilinear regression to create a reference model to which we compare our neural network model results. Lastly, we train an ANN using DMS measurements and environmental parameters. With the trained networks, we extrapolate the sparse measurements globally to obtain gridded fields of monthly DMS distributions and sea-to-air DMS fluxes.

## 2 Materials and Methods

### 2.1 Data sources and cleaning

Surface ocean DMS data were obtained from the Global Surface Seawater DMS database (PMEL) and from the North Atlantic Aerosol and Marine Ecosystems experiment [NAAMES] (Behrenfeld et al., 2019) (Table A1). In total, there are 93,571 valid measurements (PMEL: 86,785 and NAAMES: 6,786) after removing ultra-low (<0.1 nM) and ultra-high (>100 nM) DMS measurements according to Galí et al. (2015). The number of measurements used are substantially more than the 47,313 used by Lana et al. (2011). The Global Surface Seawater DMS database also includes some ancillary in situ data, such as DMSP (4,620), Chl $a$ (PMEL: 11,491, NAAMES: 6750), sea surface temperature (SST; PMEL: 81,069, NAAMES: 6,786), and salinity (SSS; PMEL: 77,209, NAAMES: 6,786). In situ SST and SSS were used if available. If not, monthly climatology data from other sources (Table A1) were used to fill the gaps. SeaWiFS Chl-$a$ data (monthly average, Level 3-binned, spatial resolution of 9.2 km, last access date May 1st 2020) from December 1997 to March 2010 were matched to DMS data according to

coordinates and sampling date. We compared PMEL in situ Chl $a$ to SeaWiFS Chl $a$, which are well correlated on logarithmic scale ($R^2 = 0.64$) with a slope of 0.67 and an intercept of -0.06, $[log(Chl_{SeaWiFS}) = 0.67log(Chl_{insitu}) - 0.01]$, which means that on logarithmic scale SeaWiFS Chl-$a$ concentrations are on average $\sim$30% lower than those of in situ Chl-$a$ concentrations. This is possibly because SeaWiFS Chl $a$ is calibrated based on HPLC determined Chl $a$ (Morel et al., 2007), which on average is $\sim$40% lower than that determined using Fluorometric method (Sathyendranath et al., 2009). Unfortunately, there is no flag in the database showing how Chl $a$ was determined. For consistency, we use only Chl-$a$ data retrieved from SeaWiFS in the following multilinear and network models.

SeaWiFS photosynthetically available radiation (PAR) and diffuse attenuation coefficient for downwelling irradiance at 490 nm (Kd490) (monthly average, both are L3BIN with spatial resolution of 9.2 km, last access date May 1st 2020) from September 1997 to August 2010 were matched with DMS according to coordinates and sampling date. Mixed layer depth climatologies were obtained from the MIMOC climatology (Schmidtko et al., 2013). Sea ice cover was from a simulation with the ocean component of the Community Earth System Model (CESM) forced with a repeating thirty year cycle (1980-2009) of NCEP reanalysis datasets (Wang et al., 2019). The output was averaged into a monthly climatology and was used as part of the air-sea gas exchange calculations. Nutrient data (nitrate, phosphate, and silicate) from World Ocean Atlas (WOA2013, Garcia et al. (2013)) were also included in the multilinear regression and neural network analyses, since they can exert influence on phytoplankton distribution and thus influence DMS production (Wang et al., 2015; Archer et al., 2009). The ancillary data are then matched with DMS data according to sampling location and time of year.

The entire dataset is subjected to another round of quality control following Galí et al. (2015). Specifically, coastal data with salinity lower than 30 and samples with sampling depth greater than 10 m were removed. Additionally, data with extremely low nutrient concentrations (e.g. DIP < 0.01 $\mu$M, DIN < 0.01 $\mu$M, SiO$_4$ < 0.1 $\mu$M) or low Chl-$a$ concentrations (Chl $a$ < 0.01 mg/m$^3$) were also removed because a) the low concentrations are below traditional method detection limits and b) they cause the data distributions severely left skewed, which significantly affects the performance of a ANN model.

## 2.2 Linear regressions

Linear regression models are conducted on three sets of data to diagnose the predictive skill of each ancillary variable. As a first step, we restrict the regression model to the PMEL data sets where both DMS and the predictor variable are simultaneously available. This selection process yields a total of 10,404 pairs for Chl $a$ and DMS, 4,061 pairs of total DMSP (DMSPt) and DMS, 69,197 pairs of SST and DMS, and 85,150 pairs of SSS and DMS, respectively. In a second step, we conduct regression models on combined PMEL and NAAMES data. Since almost all NAAMES samples are accompanied by in situ measurements of Chl $a$, SSS, and SST, the data pairs increased to 17,153 pairs for Chl $a$ and DMS, 75,983 pairs of SSS and DMS, and 91,936 pairs of SST and DMS, respectively. In a third step, to keep Chl $a$ data sources consistent as described previously, we use satellite Chl $a$; the other unmeasured predictors (i.e. MLD, PAR, Nitrate (DIN), Phosphate (DIP), and Silicate (SiO$_4$), SST, and SSS) are filled in using monthly climatology data from the previously cited sources. DMSPt is not included, because there is no observation based climatological dataset to fill the missing values.

To reduce the dynamic range, we log-transform the DMS, DMSPt, Chl $a$, MLD, DIP, DIN, SiO$_4$, and SST after conversion to absolute temperature to avoid losing data with temperature below or equal to 0 °C. The corresponding predictors are then standardized to their z-score, $Z \equiv (C - \overline{C})/\sigma$, where $C$ is predictor's concentration; $\overline{C}$ is the mean of the variables; and $\sigma$ is standard deviation of the variables. Matlab's `polyfit` function is applied to each pair to fit a first degree polynomial, i.e. a linear regression.

## 2.3 Multilinear regression

We begin by applying a step-wise multi-linear regression model to the environmental data using Matlab's `stepwiselm` function. In a first test, we consider a total of eight potential DMS predictors: PAR, MLD, Chl $a$, SSS, SST, DIN, DIP, and SiO$_4$. In a second test, we combine the above eight potential parameters with sampling location and time parameters (Eq: 1-3). The ANN requires that the predictor fields be available for every DMS data point so we fill missing values in the environmental dataset with climatological data. We eliminate DMS measurements that are under ice cover, leaving us with 82,996 DMS measurements with a complete set of predictors.

The in situ sampling times (months and hours) were converted to periodic functions using sine and cosine functions to address the data continuity issue, such that in a diurnal or seasonal cycle the start (0th hour or January) and the end (24th hour or December) of a cycle share the same properties, but are numerically different. The coordinate space notations have a similar issue in the longitudinal direction. The conversions are conducted according to Gade (2010) and Gregor et al. (2017) as follows:

$$\begin{bmatrix} H1 \\ H2 \end{bmatrix} = \begin{bmatrix} \cos(\text{hour}\frac{2\pi}{24}) \\ \sin(\text{hour}\frac{2\pi}{24}) \end{bmatrix},$$

$$(1)$$

$$\begin{bmatrix} M1 \\ M2 \end{bmatrix} = \begin{bmatrix} \cos(\text{month}\frac{2\pi}{12}) \\ \sin(\text{month}\frac{2\pi}{12}) \end{bmatrix},$$

$$(2)$$

$$\begin{bmatrix} L1 \\ L2 \\ L3 \end{bmatrix} = \begin{bmatrix} \sin(\text{lat}\frac{\pi}{180}) \\ \sin(\text{lon}\frac{\pi}{180})\cos(\text{lat}\frac{\pi}{180}) \\ -\cos(\text{lon}\frac{\pi}{180})\cos(\text{lat}\frac{\pi}{180}) \end{bmatrix}.$$

$$(3)$$

Bayesian Information Criterion (BIC) of 0.01 is used as a criterion for accepting or rejecting a predictor, which means that predictors are removed if they induce a BIC increase of more than 0.01.

## 2.4 Artificial Neural Network (ANN)

To assess the possibility that a non-linear model might provide better prediction, we train artificial neural networks (ANNs) using the `Keras` deep learning toolbox in Python. DMS concentration along with the eight environmental predictors (PAR,

MLD, Chl $a$, SSS, SST, DIN, DIP, and SiO$_4$) are log-transformed. The predictors' dynamic ranges are then constrained to the [-1,1] interval using a minmax normalization, i.e. $C_{norm} \equiv (C - C_{min})/(C_{max} - C_{min})$, where $C_{min}$ and $C_{max}$ are the minimum and maximum values in the data $C$, respectively.

The dataset is then separated into three sets: training, internal testing, and external validating sets. Data from each of the fourteen one-degree-latitude bands (64°N−65°N, 54°N−55°N, 44°N−45°N, 34°N−35°N, 24°N−25°N, 14°N−15°N, 4°N−5°N, 4°N−5°S, 14°S−15°S, 24°S−25°S, 34°S−35°S, 44°S−45°S, 54°S−55°S, 64°S−65°S,) are left out for internal testing (9,084 points). Data from each of the fifteen one-degree-latitude bands (69°N−70°N, 59°N−60°N, 49°N−50°N, 39°N−40°N, 29°N−30°N, 19°N−20°N, 9°N−10°N, 1°N−0°S, 9°S−10°S, 19°S−20°S, 29°S−30°S, 39°S−40°S, 49°S−50°S, 59°S−60°S, 69°S−70°S) are left out for external validation (10,870 points). The remaining data (63,042 points) are used to train the neural network. The data was split into the above sets manually rather than automatically. This is because data collected from the same cruise are highly intercorrelated. The common practice of shuffling and randomly splitting the data produces an over-fitted model because the validating data can be predicted using near-neighbor values. This kind of apparent skill does not generalize to regions with large data gaps, which we need for constructing a robust climatology. We also manually adjust the hyper-parameters (dropout ratio, hidden layers, number of nodes etc.) using the data that has been manually-divided into training, internal testing, and external validation subsets. After obtaining a satisfactory combination of those hyper-parameters (as discussed below), we fix them and fine tune the network using all available data.

The network has one input layer with input nodes corresponding to the number of predictors, two dense hidden layers with 128 nodes each, and one output layer with one node corresponding to the predicted logarithm of DMS concentration. To avoid overfitting, we add two dropout layers with a dropout ratio of 25% after each hidden layer. We also apply a L2 kernel regularizer for each hidden layer with the regulation parameter value set to 0.001. When the network is trained, the mean squared error of the internal validation data is monitored, and the training is stopped when there is no error reduction in 10 epochs. An epoch consists of one forward pass and one backward pass of all the training examples. Only the best model with the lowest validation mean squared error is saved. We tested different network setups - the current setting achieves goodness of fit, but avoids overfitting.

### 2.4.1 Parameter selections

The 15 predictors (8 environmental predictors and 7 time and coordinate signatures) were tested separately. In the first set of tests, we use only time and location parameters. In the second set of tests, we run a series models that examine every possible combination of the eight environmental parameters (a total of 255 combinations) to time and location parameters. The models are then ranked according to the root mean square error of the validation data.

### 2.4.2 Monthly climatology

To obtain monthly DMS climatologies, we interpolate the corresponding predictor variables (PAR, MLD, Chl $a$, SSS, SST, DIN, DIP, and SiO$_4$) onto a 1° by 1° grid. Coordinates and target months are transformed accordingly. We then apply the top

10 (Section: 2.4.1) trained networks to obtain DMS monthly concentrations. Monthly results from 10 models are then used to produce the final monthly climatology and to analyze uncertainties.

## 2.5  Sea-to-air flux

Air-sea gas transfer is estimated using the following bulk formula,

$$F = K_w(C_w - C_a/H), \tag{4}$$

where $F$ is sea-to-air gas exchange flux, $C_a$ and $C_w$ are bulk air and bulk water gas concentrations, and $K_w$ (cm/hr) is the overall gas transfer velocity, expressed in water side units (Liss, 1974). $K_w$ reflects the combined resistance to gas transfer on both sides of the interface, as follows:

$$1/K_w = 1/k_w + 1/(Hk_a)), \tag{5}$$

where $H$ is the dimensionless (gas/liquid) Henry's law constant and $k_a$ and $k_w$ are gas transfer velocities in air and seawater. DMS in the surface ocean is strongly supersaturated with respect to that in the overlying atmosphere ($C_w \gg C_a$), which simplifies the flux Eq. 4 to

$$F = K_w C_w, \tag{6}$$

For this study we used two parameterizations for $K_w$. The Goddijn-Murphy et al. (2012) parameterization (hereafter GM12) is based on regressions between satellite based wind-speed observations with shipboard in situ measurements of DMS gas transfer velocities using eddy covariance. The GM12 parameterization for $K_w$ normalized to a $S_c$ number of 660 is

$$K_{w,660} = 2.1U_{10} - 2.8, \tag{7}$$

where $U_{10}$ is wind speed (m/s) at 10 m above sea surface. Negative $K_{w,660}$ values produced at low wind speeds are set to zero. We also utilized the Nightingale et al. (2000) (hereafter N00), which is based on shipboard $^3$He/SF$_6$ dual tracer experiments. Their parameterization for water side only DMS gas transfer velocity at a Schmidt number of 660 ($\kappa_{w,660}$) is calculated as follows,

$$k_{w,660} = (0.222U_{10}^2 + 0.333U_{10})(Sc_{DMS}/600)^{-0.5}, \tag{8}$$

where $Sc_{DMS}$ is calculated as a function of temperature after Saltzman et al. (1993). A total transfer velocity is obtained from N00 as follows,

$$K_{w,660} = k_{w,660}(1 - \gamma_a), \tag{9}$$

where $\gamma_a$ is atmospheric gradient fraction given by $\gamma_a = 1/(1 + k_a/\alpha k_{w,660})$ (McGillis et al., 2000). Air side DMS transfer velocity is given as $k_a = 659U_{10}(M_{DMS}/M_{H_2O})^{-0.5}$, where $M_{DMS}$ and $M_{H_2O}$ are the molecular weights of DMS and water, respectively (McGillis et al., 2000).

DMS fluxes were calculated using surface ocean DMS concentrations from the ANN results and a satellite-based wind speed climatology (Table A1 and Fig. A2). Because the N00 parameterization was calibrated using in situ wind speeds and has a nonlinear quadratic dependence on wind speed, the use of monthly mean wind speeds will introduce errors. To reconcile the differences between in situ wind speed and monthly mean wind speed, a correction is applied according to Simó and Dachs (2002) by assuming that instantaneous wind speeds follow a Rayleigh distribution. Eq. 8 thus becomes $k_{w,660} = [0.222\eta^2\Gamma(1+2/\xi) + 0.333\eta\Gamma(s)](Sc_{DMS}/600)^{-0.5}$, where $\eta^2 = 4U_{10}^2/\pi$; $s = (1+1/\xi)$, and $\xi = 2$ for Rayleigh distribution (Livingstone and Imboden, 1993). Ice fraction data are from the CESM simulation monthly climatology. DMS fluxes from ice-covered regions are set to zero, although DMS concentration in or below sea ice is not necessarily zero.

## 3 Results and discussion

### 3.1 Linear regressions

The linear regression coefficients and $R^2$ values are summarized in Table 1. For the test using in situ measurements, DMS and DMSPt show the strongest positive correlation with a $R^2$ value of 0.41 (n = 4061). Galí et al. (2018) reported a slightly higher $R^2$ value (0.42) with less data points (n = 3637). It is not surprising to find the strong relationship between total DMSP (DMSPt) and DMS, since DMS derives from the enzymatic cleavage of DMSP (Stefels, 2000; Stefels et al., 2007). Since DMSP is directly produced by phytoplankton and does not undergo sea-to-air gas exchange, it is relatively easy to parameterize in a biogeochemical model (Galí et al., 2015). The strong relationship between DMS and DMSP point toward a potential way to model marine seawater DMS. McParland and Levine (2019) developed a mechanistic model that related intracellular DMSP concentration to environmental stress, and coupled the model with MIT ecosystem model (DARWIN) to estimate global ocean DMSP distribution. Galí et al. (2015) first applied a remote sensing algorithm to obtain a DMSP climatology, from which they predict DMS climatology through an empirical relationship with PAR (Galí et al., 2018).

The second strongest predictor is in situ Chl $a$ ($R^2$ = 0.21, n = 10,404), which is slightly higher than that by Galí et al. (2018) who reported a $R^2$ value of 0.20 (n = 8,141). The positive correlation between Chl $a$ and DMS is possibly due to the fact that the precursor of DMS, namely DMSP, is biogenic. However, when we test the relationship on satellite-based climatological Chl $a$, it becomes weaker (PMEL, $R^2$ = 0.09, n = 81,767; PMEL+NAAMES $R^2$ = 0.09, n = 88,516). The weaker relationship can be caused by several reasons: 1) Greater variance in the larger dataset (81,767 vs 10,404); 2) mismatch between satellite derived Chl-$a$ concentrations and analytical Chl-$a$ concentrations; 3) the in situ Chl-$a$ samples in PMEL database were collected mainly in highly productive regions (Galí et al., 2018), whereas the relationship between Chl-$a$ and DMS may negatively correlated in oligotrophic oceans over the seasonal cycle (Galí and Simó, 2015).

When tested against climatological data with gaps filled-in, PAR has the strongest correlation with DMS (PMEL: $R^2$ = 0.07, n = 82,137; PMEL+NAAMES: $R^2$ = 0.09, n = 88,923) with a positive correlation slope. Climatological MLD is the second strongest predictor (PMEL: $R^2$ = 0.06, n = 81,646; PMEL+NAAMES: $R^2$ = 0.07, n = 88,214) of raw DMS data, with a slope of -0.25 for PMEL and -0.26 for PMEL and NAAMES combined data.

## 3.2 Multilinear regression

A multilinear regression model that uses a combination of predictors or product of predictors has higher predictive ability than a linear regression model. For example, a multilinear regression model using eight environmental parameters has a $R^2$ value of 0.28, which is higher than that of any of the linear models. By adding time and location parameters, the $R^2$ value increases to 0.39 (n = 82,996, Fig. 1.) The results emphasize the importance of including time and location information in the model. Sampling time and location are useful predictors, especially when the output has strong seasonality such as DMS. Given a location and sampling time, the model roughly predicts the level of DMS concentrations (e.g. high latitude DMS concentrations are higher in summer than in winter, $R^2 = 0.24$). However, it is apparent that the multilinear regression model significantly underestimates high DMS concentrations. The generally low correlation coefficient hinders the possibility of reliably extrapolating the model to the global ocean.

## 3.3 ANN

Fig. 1b displays the correlation between DMS observations and ANN predictions. Compared to simple linear and multilinear regression models, ANN captures much more of the observed DMS variance ($R^2 = 0.66$, n= 82,996). Compared to previous extrapolations (Kettle et al., 1999; Lana et al., 2011), the ability of the ANN to build a nonlinear relationship between DMS and environmental predictors allows it to capture more of the variance. The ANN model can also incorporate sampling time and coordinate signals present in the data (see below). As a result, the extrapolation obtained from the ANN considers the relationship with geographical neighbors and also with temporal relationships.

From traditional linear or multilinear models, one can easily determine which parameter is a strong predictor and how a predictor influences the state variable (e.g. the correlation between DMSP and DMS). An ANN model is much more complex, it adjusts weights of each node that connect inputs and outputs. The relationship between inputs and outputs is therefore much more subtle, and that is why ANN models are generally referred to as a "Black Box". In this study, we design experiments that help open this "Black Box" and reveal parameters that drive surface ocean DMS distributions.

As shown in Fig. 2, without using any environmental parameters, sampling location and date alone can explain 44% of the validation data variance (RMSE = 0.65 on natural logarithm scale). Time of day can be another possible predictor if DMS concentration varies diurnally. However, adding time of day to the model increases RMSE slightly (Fig. 2a). Galí et al. (2013c) studied diel cycle at the Mediterranean Sea and Sargasso Sea. Among their four experiments (three in the Mediterranean Sea and one in the Sargasso Sea) regular diel variation was observed at only one experiment in the Mediterranean Sea at summer season, with highest DMS values observed at midnight and lowest values at midday. In all the other experiments, diel variations for both DMS and DMSPt pools were small. Gross community DMS production during the daytime was two to three times higher than that in the nighttime, but the high DMS production was compensated by greater photochemical and microbial consumption (Galí et al., 2013c). The balance between DMS production and consumption appears to dampens DMS diel variation. This may explain why adding time parameters does not improve the ANN model's predictive ability.

Adding environmental parameters can further improve the model performance, however, different parameter combinations show different predictive abilities. Among the top 10 models ranked according to RMSE of validation data (PAR + MLD + SAL + SST, MLD + SST, SAL + SST + DIP + Chl *a*, MLD + SST + DIP, PAR + MLD + SAL + SST + SiO + DIP, PAR + MLD + SST + SiO, MLD + SAL + DIP, PAR + MLD + SST + Chl *a*, PAR + MLD + SST + SiO + DIP, SAL + SST + SiO + Chl *a*), 9 models have SST, 8 models have MLD, 5 models have PAR, SSS, and DIP, 4 models have SiO, and 3 models have

Chl *a* as a predictor, and none of the models have DIN as a predictor. Section 3.7 shows the results of a series of sensitivity tests that demonstrate how each of those parameters influence DMS distribution.

### 3.4   Binned data versus raw data

Simó and Dachs (2002) obtained high $R^2$ values between DMS concentration and the ratio of Chl *a* to MLD (Chl/MLD) when Chl/MLD is greater than or equal to 0.02, and between DMS concentration and ln(MLD) when Chl/MLD is less than 0.02.

We tried exactly the same model on raw PMEL data with in situ Chl-*a* measurements and climatological MLD, and found that both correlations between DMS and Chl/MLD (n = 4,921, $R^2 =\sim 0$ .1) and between DMS and ln(MLD) (n = 5,978, $R^2 =\sim 0$ ) are statistically insignificant. To reduce interannual variability, we binned in situ Chl *a* and DMS into monthly $1° \times 1°$ grid, and retested the above model on the binned data, and found that the correlations are still statistically insignificant.

    Vallina and Simó (2007) reported an $R^2$ of 0.95 (n=14) between DMS concentration and SRD. We applied the same linear

regressions on both raw data and monthly $1° \times 1°$ data, and found no significant correlations between DMS and SRD as calculated according to Vallina and Simó (2007):

$$\text{SRD} = \text{SI} \cdot \frac{1}{\text{Kd490} \cdot \text{MLD}}(1 - e^{-\text{Kd490} \cdot \text{MLD}}), \tag{10}$$

where SI is shortwave irradiance (W m$^{-2}$), which is converted from PAR according to Galí and Simó (2015).

    Compared to Simó and Dachs (2002) and Vallina and Simó (2007), we used significantly more data points. For example,

in this study, there is a total of 10,899 DMS measurements accompanied with simultaneous Chl *a* measurements versus 2,385 data points used in Simó and Dachs (2002), and 83,152 (DMS, MLD) pairs in this study versus 26,400 in Vallina and Simó (2007). Another noticeable difference between the current study and previous analyses is that both Simó and Dachs (2002) and Vallina and Simó (2007) binned the data into large longitude and latitude grids. By doing so, the raw data variance is greatly reduced.

Binning data will necessarily result in loss of information. A lot of information is associated with sampling location and date as shown in Fig. 2a. By binning the data into monthly $1° \times 1°$ grid, the number of data points decreases from 82,996 to only 9,018; sampling date feature (365) will be average to 12 months, and coordinate combinations will be averaged from 87,332x87,332 to 180°x360°, which represents a substantial loss of information. For ANN models, using less data points can lead to overfitting. For example, the averaged RMSE on natural logarithm scale for the 10 best ANN models is 0.608 for the

validating dataset and 0.600 for the training dataset when using the un-binned data, whereas the RMSE is 0.655 (validating) and 0.635 (training) for the model constructed using the binned data (See Fig. 2b).

### 3.5 DMS distributions

Northern and Southern hemisphere monthly mean DMS concentrations are plotted along with results from previous studies (Simó and Dachs, 2002; Vallina and Simó, 2007; Lana et al., 2011; Galí et al., 2018) (Fig. 3a). Overall, all models show similar seasonal patterns with highest concentrations in summer and lowest concentrations in winter. Our predictions are highly consistent with the products derived from satellite data reported by Galí et al. (2018), who used an optimized relationship between DMS, DMSPt, and PAR to obtain DMS climatology from satellite retrieved PAR and DMSPt fields (Galí et al., 2015). In the northern hemisphere, the algorithms by Simó and Dachs (2002) (SD02 hereafter) and by Vallina and Simó (2007) (VS07 hereafter) generate higher concentrations and a smaller seasonal amplitude. From zonal average plots (Fig.4), it is clear that the elevated monthly means from SD02 are caused by high concentrations in high latitude oceans, whereas, high monthly means of VS07 are caused by high DMS concentrations in low and middle latitude. High DMS concentration in high latitude summer (SD02) is driven by a shoaling of the MLD caused by high freshwater content (Galí et al., 2018), while high DMS concentrations at low/middle latitude (VS07) are driven by strong solar radiation dose, which is a joint effect of shallow MLD and strong irradiance.

L11 stands out in the S. hemisphere monthly mean plot (Fig. 3b), with the highest mean concentrations in January and December, when DMS concentrations are ∼2 times higher than other model predictions. Galí et al. (2018) identified five short-comings associated with the direct interpolation method employed by Lana et al. (2011). All shortcomings concern the nature of in situ DMS data, including right-skewed distribution, lack of spatial and temporal coverage, lack of duplicate measurements, and sampling bias towards DMS-productive conditions. Because of the sparsity and skewed distribution, the interpolation/extrapolation method broadcasts small scale features to large scales (Tesdal et al., 2016). This is especially true for the month of January and December when the elevated L11 monthly means were mainly driven by a small amount of extremely high DMS measurements (>40 nM) near the Antarctic continent. On the other hand, empirical models including the ANN model used in this study rely on environmental parameter climatologies to obtain the DMS climatology. Extreme conditions are smoothed out in climatological data, e.g. in the DMS database the 99 percentiles of in situ Chl-$a$ concentration is 12.58 mg/m$^3$, whereas it is only 6.85 mg/m$^3$ in the SeaWiFS climatology. When climatological data are used to generate DMS distribution, a smaller variance than in situ data is expected.

Fig. 5 displays monthly DMS concentration distributions predicted by the ANN. Generally, DMS concentrations in polar regions show strong seasonality. The highest DMS concentrations are in summer when light and temperature are ideal for primary production. For example, in austral summer, the Southern Ocean circumpolar regions display the highest DMS concentration (>10 nM). DMS concentration in the Scotia Sea and Ross Sea display the highest DMS concentration, which gradually decreases and falls below 0.5 nM in the following months when primary production is limited by light or low temperature. In boreal summer, DMS concentration in the Bering Sea and Greenland Sea can exceed 20 nM.

The high DMS concentration during the summertime at high latitudes is believed to accompany blooms of coccolithophores and *Phaeocystis*, which are strong DMSP producers (Neukermans et al., 2018; Wang et al., 2015). The shoaling mixed layer depth during the summer provides favorable conditions, i.e. stable and warm, with adequate irradiation for coccolithophores

and *Phaeocystis* growth (Galí et al., 2019). Additionally, high DMS concentrations at ice edge zones have also been observed. These high concentrations are due to the release of ice algae that are prolific DMSP producers (Stefels et al., 2012; Webb et al., 2019). As an important cryoprotectant and osmolyte, DMSP helps ice algae to cope with the low temperature and high salinity conditions (Thomas and Dieckmann, 2002).

Another interesting region is the Pacific equatorial upwelling region. Large-scale upwelling brings nutrient-rich waters to the surface, which nourish highly productive phytoplankton communities. Overall, the seasonality in the equatorial Pacific is weaker than that in polar regions, but there is still a clear seasonal pattern. In the period from December to April, the tongue with higher DMS concentration ($\sim$3 nM) extends to the west Pacific Ocean reaching the east coast of Australia and the Philippine Sea. The tongue gradually retreats eastward in the following months. From September to November, the tongue is

constrained to the eastern Pacific and DMS concentration falls to its lowest values ($<$2.0 nM). High DMS concentrations in the west Pacific ocean from November to February are also predicted by Lana et al. (2011).

    The subtropical gyres show consistently low DMS concentrations and weak seasonal cycles throughout the year. In the southern hemisphere gyres, DMS concentrations are highest during austral summer, when the ocean is strongly stratified and local primary production is low. There are hot spots where DMS concentration exceeds 3 nM in December and February. DMS

concentrations are generally low ($\leq$ 1 nM) during austral spring and winter seasons. In the period from April to September, DMS concentrations in the S. Atlantic gyre fall below 0.6 nM. In the northern hemisphere gyres, DMS concentrations are high during the boreal summer season. Fig. 6 compares monthly mean Chl-*a* concentrations to DMS concentrations in N. and S. hemisphere gyres. The concentrations are normalized to the range of 0 to 1. It is clear that Chl *a* and DMS are anti-correlated, DMS concentration peaks at summer season when Ch-*a* concentration is generally low. This phenomenon is previously termed

as "summer DMS paradox" (Simó and Pedrós-Alió, 1999). This pattern is more apparent in the S. hemisphere gyres, because the terrestrial influence is smaller in the S. hemisphere than in the N. hemisphere.

### 3.6  Sea-to-air flux

In this study, we computed monthly sea-to-air DMS fluxes using both the GM12 and N00 gas transfer velocity parameterizations (Fig. 7 and Fig. 8). These yield global DMS annual fluxes of 15.89$\pm$0.34 Tg S yr$^{-1}$ [GM12] and 20.12$\pm$0.43 Tg S yr$^{-1}$

[N00], respectively. The uncertainties ($\pm 1\sigma$) are calculated according to DMS distributions from the top 10 ANN models based on different parameter combinations. We also calculated sea-to-air DMS fluxes using the N00 parameterization and previous DMS climatologies from Lana et al. (2011) [L11], Simó and Dachs (2002) [SD02], Vallina and Simó (2007) [VS07], and four from Galí et al. (2018)[Gali18]. Among those climatologies, VS07 produces the highest annual DMS flux (31.59 Tg S yr$^{-1}$), the ensemble of Galí et al. (2018) climatologies produce the lowest flux (18.18 $\pm$ 0.52 Tg S yr$^{-1}$) (Table 2). Generally, our

fluxes are consistent with previous results when the same flux parameterization, wind speed field, sea surface temperature, and ice coverage are used. The sea-to-air flux based on the GM12 parameterization is $\sim$24% lower than that based on N00.

    Geographically, in the high-latitude northern hemisphere, sea-to-air DMS fluxes are low in boreal winter, even though wind speeds are high. The DMS flux tends to increase in the proceeding months and reaches a maximum in boreal summer, despite the lower wind speeds (Fig. A2). The inverse relationship between wind speed and DMS flux indicates that the high DMS

flux is mainly driven by high seawater DMS concentrations. Large sea-to-air DMS fluxes at high latitudes in austral summer are driven jointly by high DMS concentrations and high wind speeds (Fig. 7 and Fig. A2). The eastern tropical Pacific Ocean displays a year-round intermediate sea-to-air DMS flux. This is mainly driven by the high DMS concentration in this region, since the wind speeds here are generally low (Fig. 7 and Fig. A2).

Fig. 8 displays integrated monthly global DMS fluxes for both hemispheres and for the global ocean based on GM12 velocity parameterizations. Globally, DMS fluxes are highest in the winter months (Dec., Jan., and Feb.) and March, which is mainly driven by high DMS flux in the Southern Hemisphere. There is another peak in the months of July and August because of northern hemisphere flux peaks. An interesting feature is that the Northern hemisphere peak is close to Southern hemisphere though, and does not reach the peak level in the Southern hemisphere. This is mainly because of the larger surface area in the Southern hemisphere. High DMS fluxes in the southern hemisphere have profound impact to the Earth's climate because there are less terrestrial and anthropogenic aerosol inputs compared to the northern hemisphere.

## 3.7 Sensitivity tests

Section 3.3 screens key parameter combinations that have the highest prediction skill. To demonstrate how these parameters influence the predicted distribution and sea-to-air flux of DMS, we ran a series of sensitivity tests. In each test, we increase/decrease one environmental parameter at a time. Fig. 9 shows annual mean differences between perturbed models and the control model. These sensitivity tests show regional differences in the sign of the perturbations anomalies. This non-linear behavior of the ANN model is not possible with a simple linear model.

For the temperature sensitivity test, we uniformly increase SST by 2 °C for the whole ocean (Fig. 9a). Compared to the control case, DMS concentrations are lower in most of the low and middle latitude oceans and higher in high latitude oceans, especially in the Southern Ocean, the Bering Sea, and the high latitude N. Atlantic Ocean. In contrast, the linear regression model shows no correlation between SST and DMS. SST alone with date and location parameters has very low prediction ability (ranked 244th over 255 models). When combined with other parameters, SST helps to improve the model performance. For example, the combination of SST and MLD ranks 2nd among all models.

For the mixed layer depth sensitivity test, we decrease MLD by 10% to mimic the stronger stratification in a warming world (Fig. 9c). DMS concentrations increase in most of the ocean , in line with the linear regression result. In the PAR sensitivity test, we uniformly increased PAR by 10% with the expectation that light exposure will increase in the future because of MLD shoaling (Fig. 9e). DMS concentrations increase with increased PAR, in agreement with the linear regression result and also with the physiological role of DMS. First, high radiation negatively influences the bacterial population/activity, which decreases DMS consumption (Galí et al., 2013a, b, c; Royer et al., 2016). Second, high radiation promotes DMS production by inducing oxidative stress within algal cells (Toole et al., 2006; Sunda et al., 2002; Royer et al., 2016).

For the salinity sensitivity test, we uniformly decrease surface ocean salinity by 1 psu. Similar to the temperature sensitivity result, the changes of DMS concentration show regional variations. DMS concentrations increase in most of the Southern Ocean, the high latitude N. Atlantic Ocean, and the Arctic Ocean, whereas DMS levels decrease in the eastern North Pacific Ocean, the Indian Ocean, and South Atlantic Ocean (Fig. 9b). The linear regression model also shows that there is no sig-

nificant correlation between DMS and salinity. As in the case for temperature, salinity works synergistically with the other environmental parameters to predict the DMS concentration.

Figs. 9e and f show the sensitivity tests for DIP and $SiO_4$, respectively. For these tests, we decrease DIP and $SiO_4$ concentrations by 10% with the expectation that increasing ocean stratification due to global warming will decrease the nutrient supply from the deep ocean. In certain regions, the two nutrient perturbations have nearly opposite effects. For example, DMS concentrations drop slightly in the western Pacific and Indian Ocean for the DIP perturbation experiment, whereas the concentrations have almost opposite patterns in those regions for the $SiO_4$ perturbation experiment. In the eastern Pacific Ocean, the Southern Ocean, and high latitude N. and S. Atlantic oceans, reduced DIP concentration triggers an increase of DMS concentrations, which might be related to nutrient stress, which can increase DMSP production by low DMSP producers (e.g. diatoms) (McParland and Levine, 2019). The increase of DMS concentration for the $SiO_4$ perturbation is potentially due to a regime shift away from diatoms, which are low DMSP producers, to other more prolific DMSP producers.

Fig. 9g shows the sensitivity test for Chl $a$. In the test, we decreased Chl $a$ concentration by 10% to mimic the decreased primary production caused by ocean stratification and nutrient depletion. Overall, the most apparent changes are in the subtropical gyres, where DMS concentrations are lower than the control run. DMS concentrations increase in some marginal seas and coastal oceans such as the Arabian Sea and eastern coast of Australia. Previous studies of the relationship between DMS and Chl $a$ have produced contradictory results. Strong correlations have been reported in basin scale studies (e.g. Yang et al., 1999). On the other hand, there are numerous studies that observed no correlation between DMS and Chl $a$ (e.g. Dacey et al., 1998; Kettle et al., 1999; Toole and Siegel, 2004). The inconsistent relationships indicate the complexity of the reduced sulfur cycle.

On a global scale, the increase of temperature does not significantly change sea-to-air flux (15.96 Tg S $yr^{-1}$ compared to 15.89 Tg S $yr^{-1}$ for the control run based on GM12) because the elevated DMS concentrations in the high latitude oceans are compensated by the reduced concentrations in the low latitude oceans. Similar to the case for the temperature perturbation, the salinity perturbation has a small effect on sea-to-air flux of DMS (15.88 compared to 15.89 Tg S $yr^{-1}$). The overall increases of DMS concentration in the MLD, PAR, and $SiO_4$ perturbation tests lead to increases of DMS sea-to-air flux of 0.56, 0.96, and 0.91 Tg S $yr^{-1}$, respectively. The Chl $a$ perturbation model is the only one that shows a slight decrease in the sea-to-air flux of DMS (15.59 Tg S $yr^{-1}$ compared to 15.89 Tg S $yr^{-1}$).

Of course, the ocean is a very complex system and changes in these environmental parameters will be correlated. For example, the projected temperature increase will lead to a stronger surface ocean stratification that will result in shoaling of MLD and reduced nutrient supplies from the deep ocean, which together will decrease primary production in the ocean. Based on our model results, if these effects work jointly, the DMS sea-to-air flux will increase more than each of the individual perturbations. Assuming that larger DMS sea-to-air fluxes induce greater cloud albedo, then we might expect the changes in DMS to represent a negative climate feedback.

## 4    Conclusions

The artificial neural network (ANN) used in this study has some advantages compared to the prior methods used to develop DMS climatologies. Most importantly, the ANN utilizes available measurements to fill regions without DMS observations, using non-linear relationships trained in more data rich regions/seasons. By contrast, objective interpolation methods are spatial/temporal averages of sparse data with weak underlying basis in environmental variability. As a result, the ANN approach captures significantly more of the raw data variance than simple linear/multilinear models. Simple models achieve comparable fits only after heavily binning the DMS observations (e.g. Simó and Dachs, 2002; Galí et al., 2015; Vallina and Simó, 2007; Galí et al., 2018). The ANN is computationally more expensive than the linear/multilinear models, but considerably less expensive than prognostic biogeochemical models (e.g. Vogt et al., 2010; Wang and Moore, 2011; Wang et al., 2015). The principal weakness of the ANN approach is that it does not easily provide scientific insight into the relationships between the parameters. We attempted to overcome this weakness by running a series of sensitivity tests to explore how DMS concentration might change in response to global climate warming. We found that the predicted changes in DMS concentration are almost never unidirectional in response to a change in only one environmental parameter. This reveals the underlying interactions between these environmental parameters, which a linear regression model can not achieve.

The ANN approach is a useful tool for developing trace gas climatologies. It may also be useful as a means of assessing the sensitivity of DMS to past/future changes in climate by coupling the ANN to prognostic biogeochemical models. Caution is warranted in the interpretation of such efforts because there is as yet no basis for assessing whether the relationships obtained by training on contemporary measurements apply to the past or will hold in the future. Such relationships could be investigated using paleoceanographic and ice core data (Osman et al., 2019).

The annual sea-to-air DMS flux calculated in this study is slightly ($\sim$23%) lower than the objective interpolation method of Lana et al. (2011) using the same sea-to-air gas exchange models. DMS concentrations from this study are similar to Lana et al. (2011) where measurements are abundant, so we infer that the difference is likely caused by positive bias in the objective interpolation method for data-sparse regions/seasons.

*Code availability.*    Code for ANN model is available at: https://github.com/weileiw/ANN-DMS-code

*Data availability.*    The data for DMS concentrations and sea-to-air flux are available at DOI: 10.5281/zenodo.3631875.

*Author contributions.*    W.L.W and G.S. initiated the study and drafted the manuscript. W.L.W. built the model with inputs from F.P., E.S.S., and J.K.M.. E.S.S and T.G.B provided new N. Atlantic DMS measurement data. All authors contributed to review the manuscript, and to interpret the data presented.

*Competing interests.* The authors declare that they have no competing financial interests.

*Acknowledgements.* We thank the observational DMS community for making their measurements publicly available. J.K.M., F.P., and W.L.W. are supported by DOE Earth System Modeling program (DE-SC0016539). G.S is supported by the Natural Key Research and Development Program of China (2017YFC1404403). E.S.S. and T.G.B. are supported by the NASA North Atlantic Aerosols and Marine Ecosystems Study (NAAMES), which was funded through the NASA Earth Venture Suborbital program. (NNX#15AF31G). The sources of ancillary data are listed in Table A1.

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

**Table 1.** Results of linear regression models. The $R^2$ values are for log transformed, and normalized data as described in the text.

| Parameter | in situ data | | | PMEL | | | PMEL+NAAMES | | |
|---|---|---|---|---|---|---|---|---|---|
| | $R^2$ | Slope | No. | $R^2$ | Slope | No. | $R^2$ | Slope | No. |
| DMSPt | 0.41 | 0.77 | 4,061 | - | - | - | - | - | - |
| Chl $a$ | 0.21 | 0.43 | 10,404 | 0.09 | 0.30 | 81,767 | 0.09 | 0.29 | 88,516 |
| MLD | - | - | - | 0.06 | -0.25 | 81,646 | 0.07 | -0.26 | 88,214 |
| PAR | - | - | - | 0.07 | 0.26 | 82,137 | 0.09 | 0.29 | 88,923 |
| SST | $\sim$0 | -0.01 | 69,196 | 0.02 | -0.12 | 82,770 | 0.01 | -0.12 | 89,556 |
| SSS | $\sim$0 | -0.08 | 69,196 | 0.01 | -0.10 | 82,759 | 0.02 | -0.13 | 89,545 |
| DIP | - | - | - | 0.01 | 0.11 | 81,868 | 0.02 | 0.12 | 88,654 |
| DIN | - | - | - | 0.01 | 0.10 | 79,083 | $\sim$0 | 0.09 | 85,865 |
| SiO | - | - | - | 0.04 | 0.19 | 81,813 | 0.04 | 0.20 | 88,599 |

**Table 2.** Annually-averaged zonal mean DMS flux (Tg S/yr) for this study (W20), Lana et al. (2011) (L11), Simó and Dachs (2002)(SD02), Vallina and Simó (2007)[VS07], and Galí et al. (2018)[Gali18] for their four parameterization models. L11, SD02, VS07, and Gali18 are computed with the Nightingale et al. (2000) parameterization of the piston velocity[N00]. Flux in this study is calculated using both the Nightingale et al. (2000)[N00], and Goddijn-Murphy et al. (2012)[GM12], parameterizations. Uncertainties are estimated based on top 10 models with different parameterizations. Errorbars correspond to $\pm 1\sigma$.

| Latitude | L11[N00] | SD02[N00] | VS07[N00] | Gali18[N00] | W20[N00] | W20[GM12] |
|---|---|---|---|---|---|---|
| 90°-80°N | 0.00 | 0.00 | 0.00 | 0.00± 0.00 | 0.00±0.00 | 0.00±0.00 |
| 80°-70°N | 0.08 | 0.04 | 0.02 | 0.02± 0.00 | 0.05±0.01 | 0.04±0.01 |
| 70°-60°N | 0.19 | 0.11 | 0.06 | 0.09± 0.01 | 0.13±0.01 | 0.11±0.01 |
| 60°-50°N | 0.78 | 0.52 | 0.30 | 0.38± 0.04 | 0.45±0.03 | 0.35±0.03 |
| 50°-40°N | 1.16 | 1.01 | 0.81 | 0.73± 0.08 | 0.79±0.06 | 0.60±0.05 |
| 40°-30°N | 1.39 | 1.64 | 1.85 | 1.18± 0.07 | 1.13±0.05 | 0.90±0.04 |
| 30°-20°N | 1.43 | 1.89 | 2.84 | 1.33± 0.02 | 1.29±0.05 | 1.07±0.04 |
| 20°-10°N | 2.60 | 2.79 | 4.29 | 1.96± 0.07 | 2.12±0.09 | 1.68±0.07 |
| 10°-0°N | 2.91 | 2.64 | 3.55 | 1.66± 0.03 | 2.11±0.10 | 1.79±0.08 |
| 00°-10°S | 2.90 | 2.40 | 3.54 | 1.84± 0.01 | 2.23±0.13 | 1.91±0.11 |
| 10°-20°S | 3.42 | 2.64 | 4.35 | 2.05± 0.02 | 2.41±0.13 | 1.93±0.11 |
| 20°-30°S | 2.91 | 2.26 | 3.74 | 1.87± 0.02 | 1.93±0.12 | 1.56±0.10 |
| 30°-40°S | 2.91 | 2.42 | 3.00 | 2.19± 0.08 | 2.20±0.19 | 1.71±0.14 |
| 40°-50°S | 2.70 | 2.19 | 2.18 | 2.07± 0.14 | 2.19±0.16 | 1.51±0.11 |
| 50°-60°S | 1.67 | 1.00 | 0.10 | 0.76± 0.07 | 1.01±0.07 | 0.67±0.05 |
| 60°-70°S | 0.18 | 0.08 | 0.08 | 0.04± 0.00 | 0.09±0.01 | 0.06±0.01 |
| 70°-80°S | 0.00 | 0.00 | 0.00 | 0.00± 0.00 | 0.00±0.00 | 0.00±0.00 |
| 80°-90°S | 0.00 | 0.00 | 0.00 | 0.00± 0.00 | 0.00±0.00 | 0.00±0.00 |
| Total | 27.23 | 23.64 | 31.59 | 18.18± 0.52 | 20.12±0.43 | 15.89±0.34 |

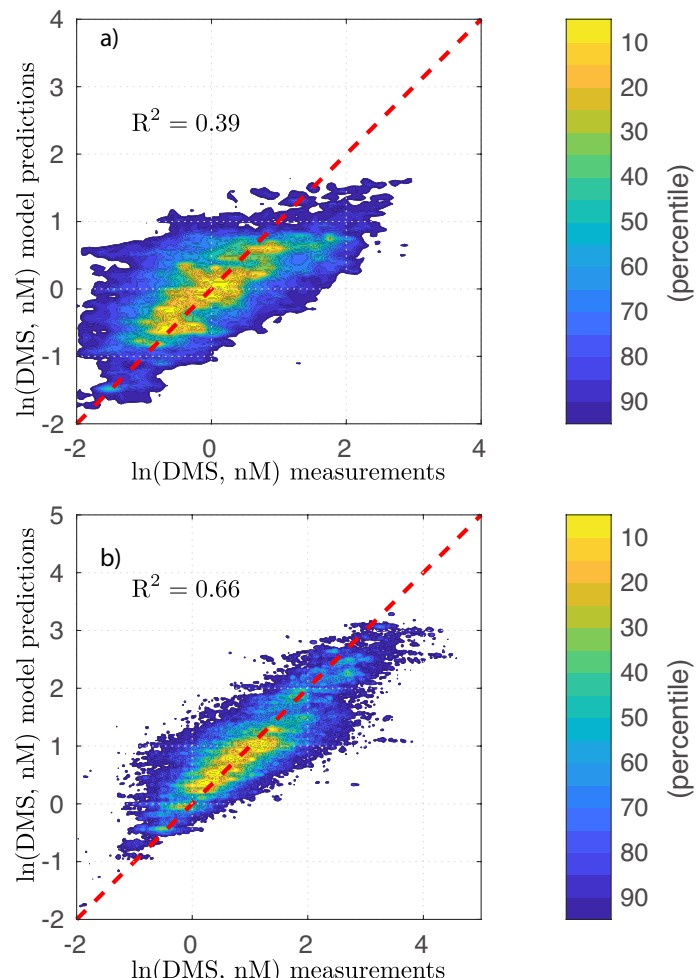

**Figure 1.** Model versus observation plots on logarithmic scale: (a) multilinear regression model; (b) artificial neural network model. The color indicates the fraction of the joint distribution explained as a percentile that falls within a region of concentration space.

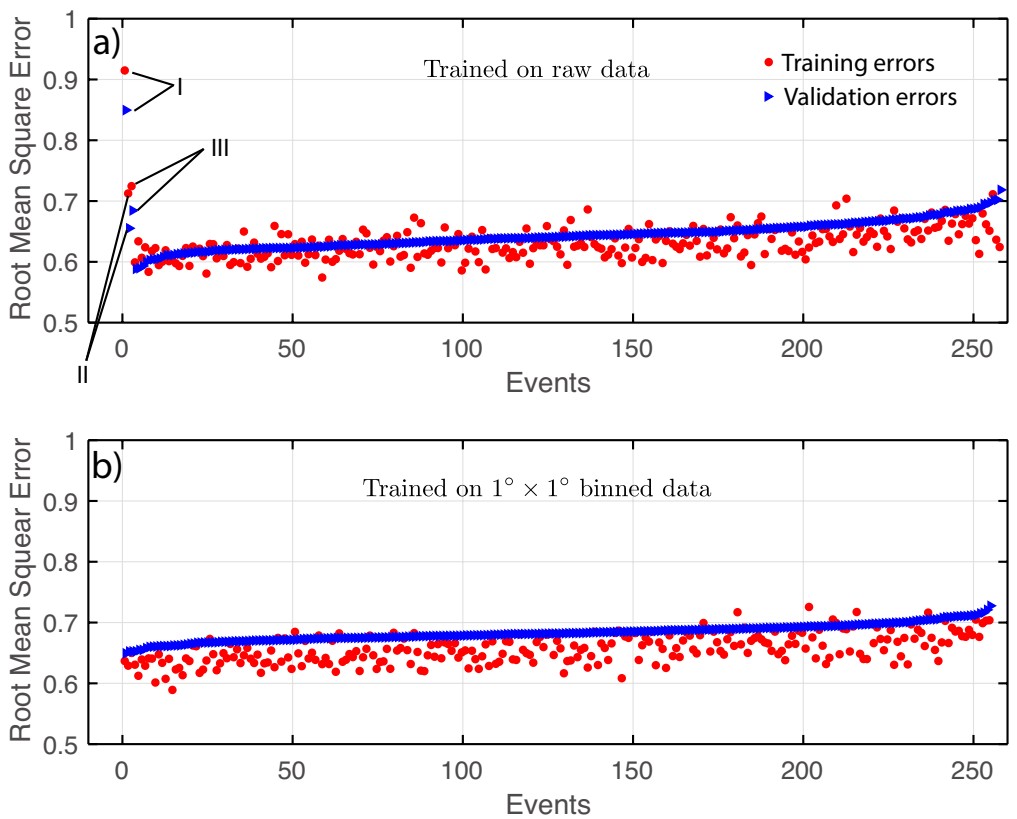

**Figure 2.** Parameter sensitivity tests on raw and binned data. (a) Root mean square error on logarithmic scale for the model trained using raw data; (b) Root mean square error on logarithmic scale for the model trained using binned data . The time and location parameters are tested separately without combining with environmental parameters as shown in the upper panel, (I) with only location parameters; (II) with location and day of year parameters; and (III) with location, day of year, and time of day parameters. The model with three location parameters (I) has a root mean square error on natural logarithmic scale of ∼0.83, which decreases to ∼0.65 by adding sampling day of year parameters (II), however, increases to ∼0.67 by adding time of day parameters (III). We, therefore, do not include sampling time parameters in the following tests. We tested every combination of the eight parameters (PAR, MLD, SST, SAL, Chl *a*, DIP, DIN, and SiO), which in total are 255 tests.

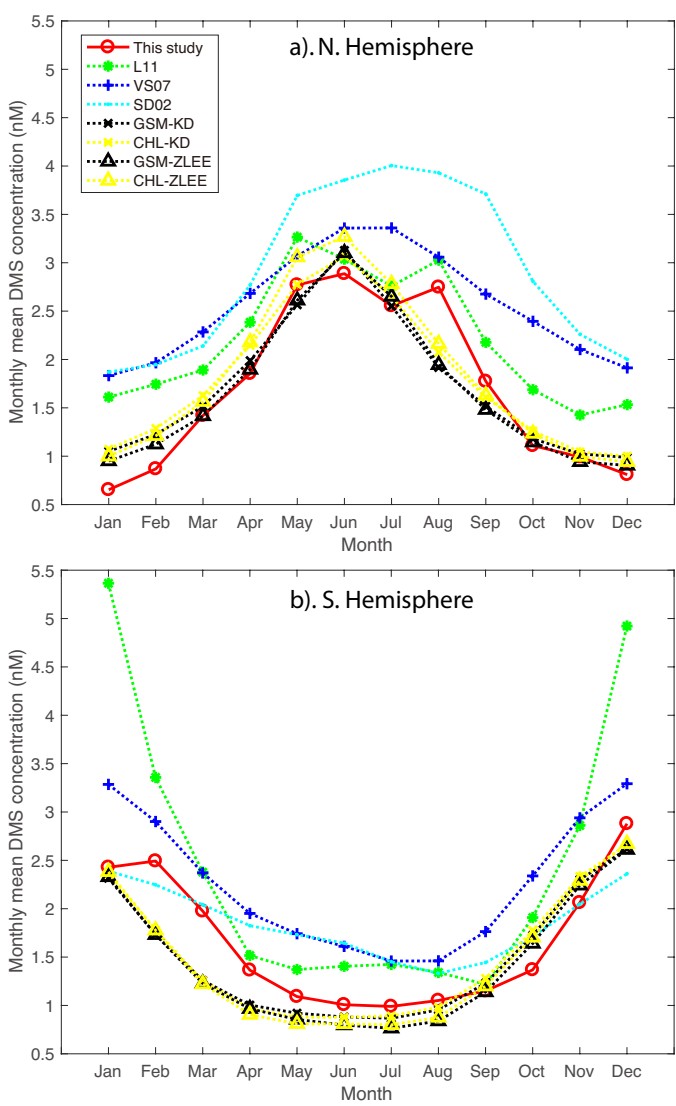

**Figure 3.** Comparisons of monthly mean DMS concentrations to previous studies (Simó and Dachs, 2002; Vallina and Simó, 2007; Lana et al., 2011; Galí et al., 2018). L11, SD02, and VS07 are self-explanatory. GSM-KD, CHL-KD, GSM-ZLEE, and CHL-ZLEE are the four model results from Galí et al. (2018).

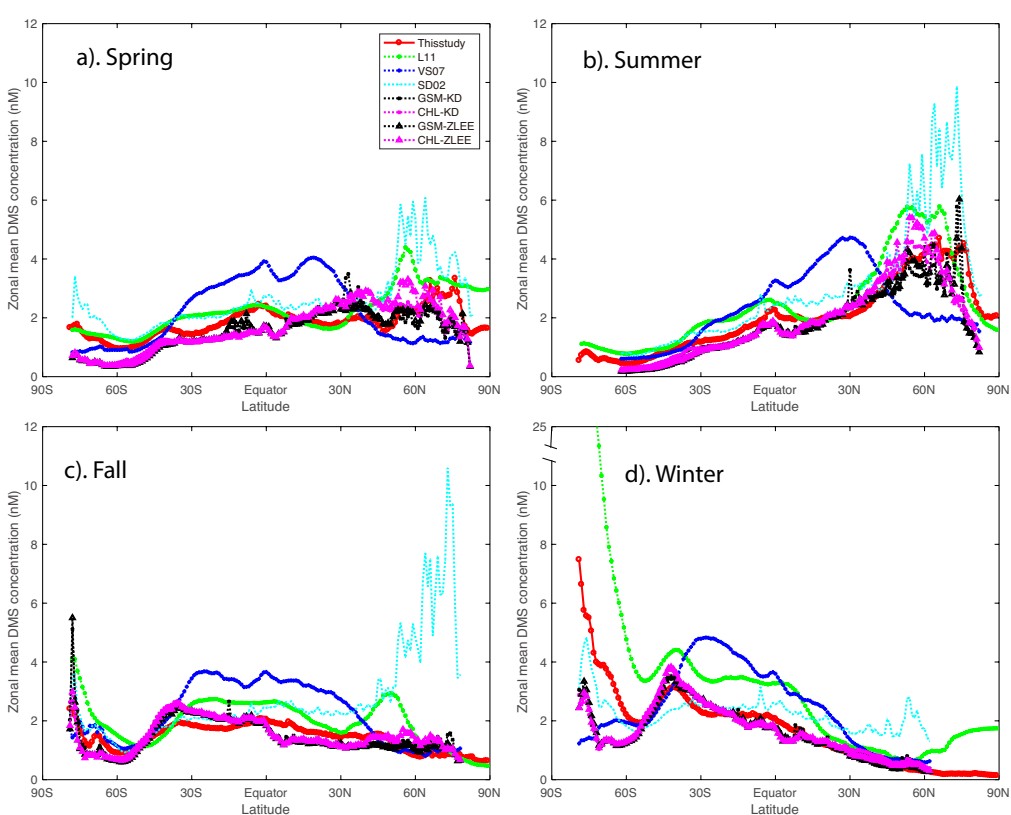

**Figure 4.** Comparisons of zonally mean DMS concentrations to previous studies (Simó and Dachs, 2002; Vallina and Simó, 2007; Lana et al., 2011; Galí et al., 2018). L11, SD02, and VS07 are self-explanatory. GSM-KD, CHL-KD, GSM-ZLEE, and CHL-ZLEE are the four model results from Galí et al. (2018).

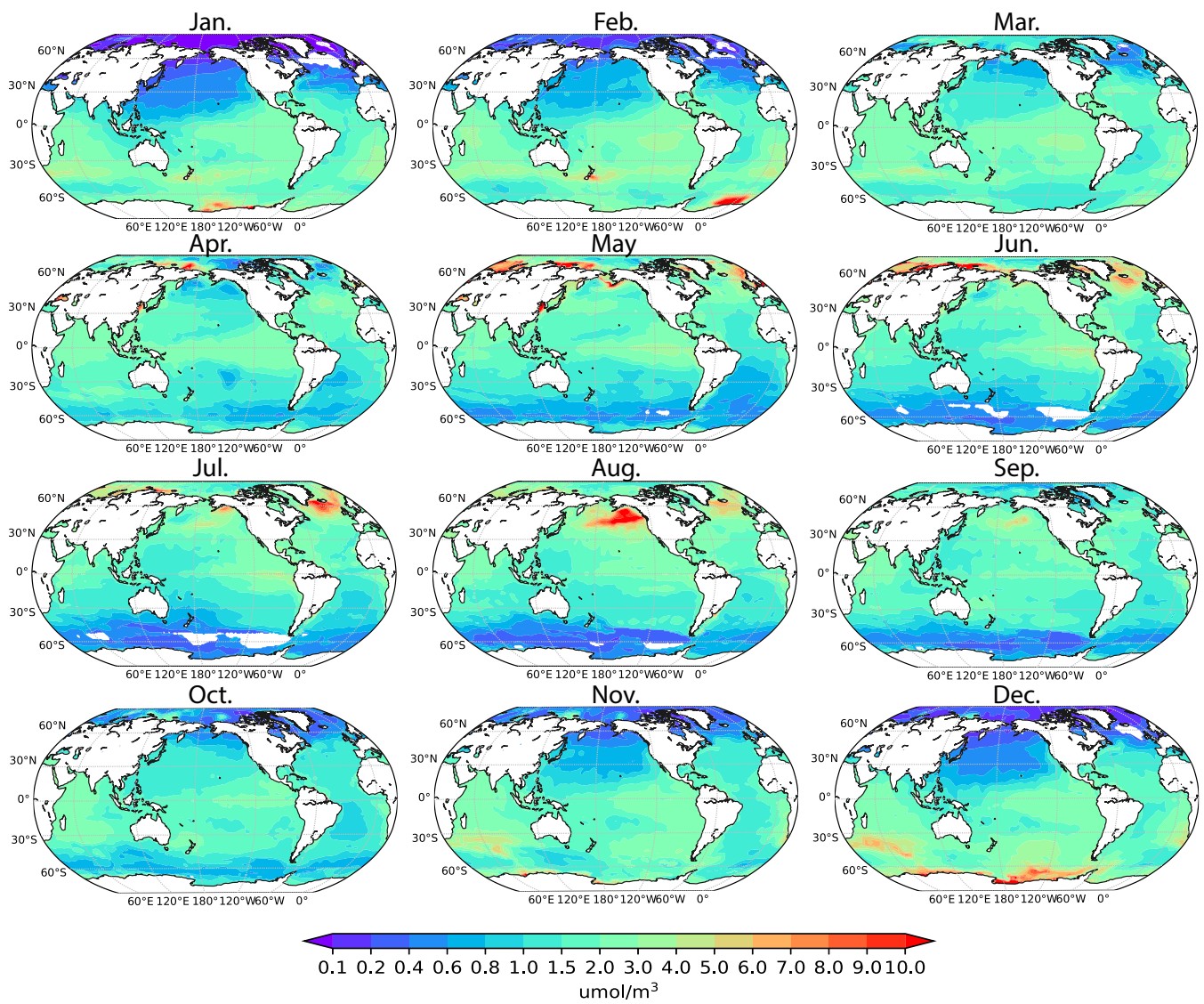

**Figure 5.** Monthly DMS concentration ($\mu$mol m$^{-3}$) estimated based on artificial neural network.

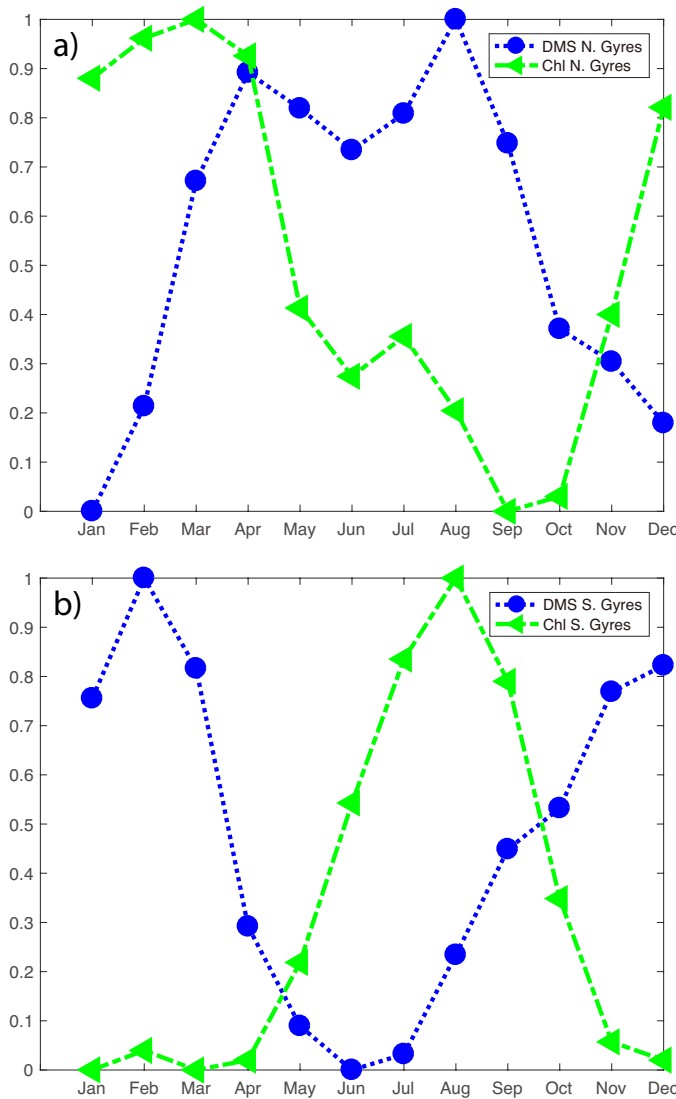

**Figure 6.** Distributions of monthly mean DMS and Chl-*a* concentrations for N. and S. hemisphere gyres. The gyres are defined as regions between $30°$ and equator where annually mean DIP concentration is below 0.2 $\mu$M. Monthly mean concentrations are normalized to the range of 0 to 1.

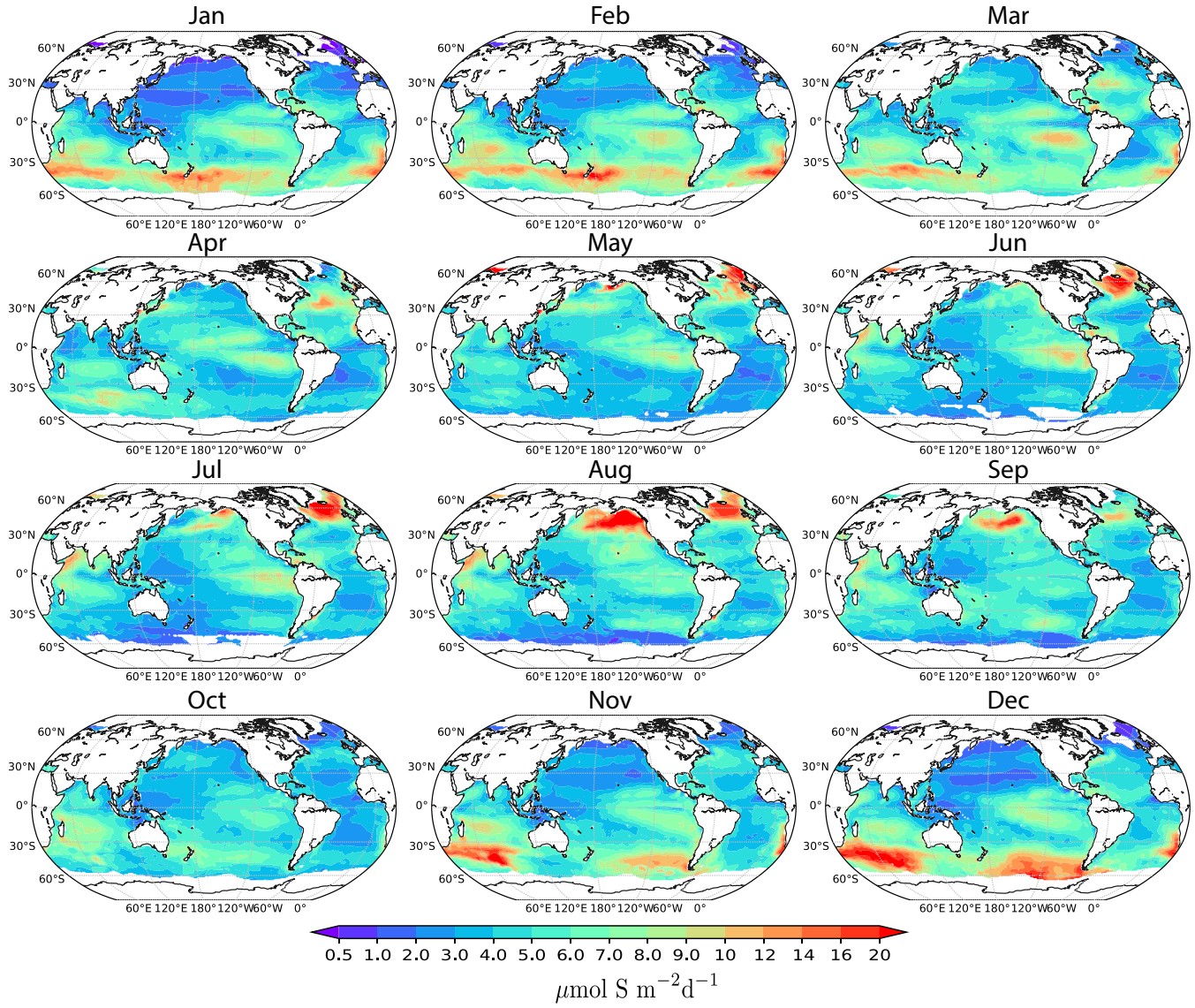

**Figure 7.** Monthly DMS flux ($\mu$mol S m$^{-2}$ day$^{-1}$) calculated based on DMS climatology estimated from the ANN model and Goddijn-Murphy et al. (2012) flux parameterization.

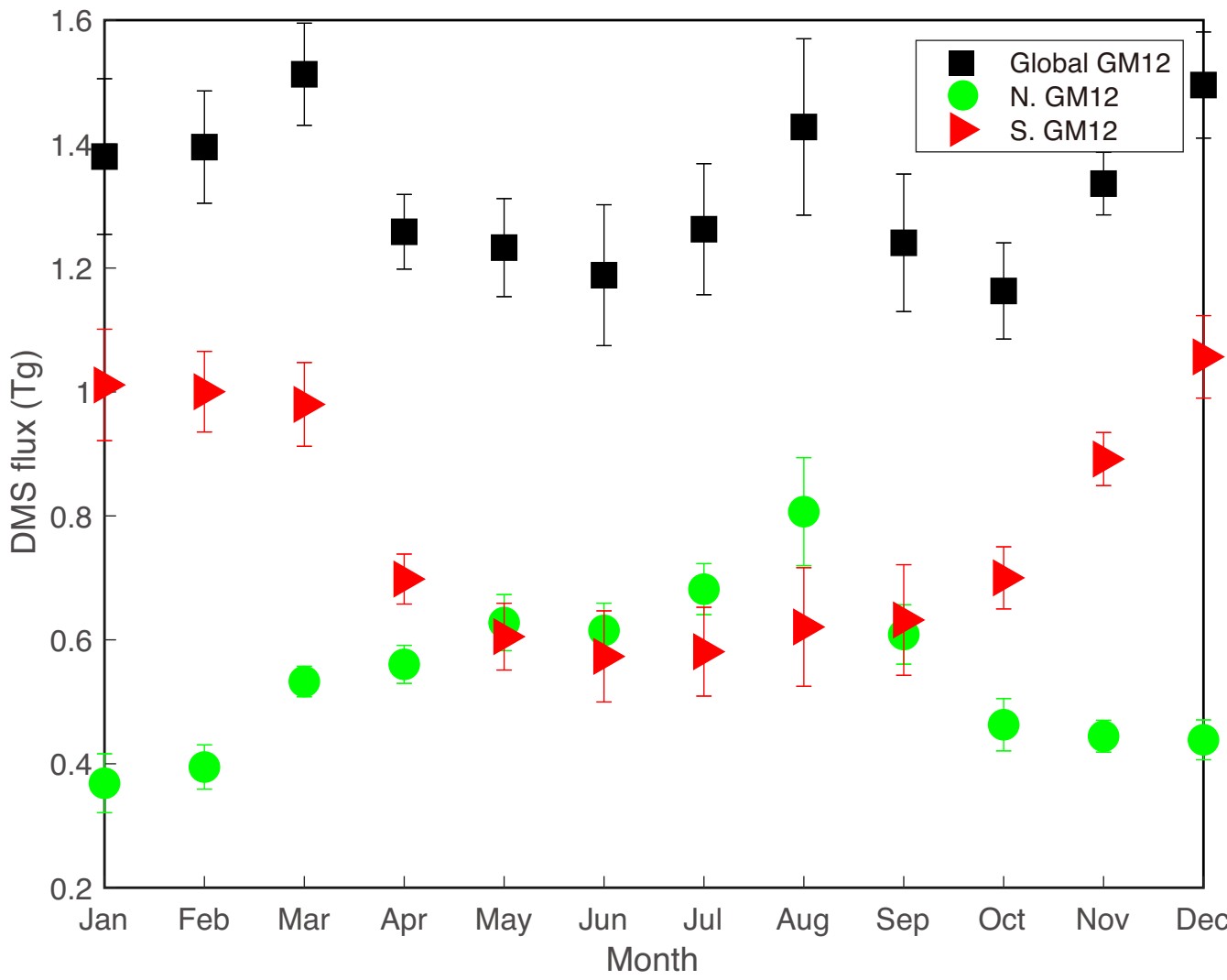

**Figure 8.** Area and month integrated DMS sea-to-air flux (Tg S month$^{-1}$) based on GM12 parameterization. Red triangles represent monthly mean flux of the Southern hemisphere, green dots represent monthly mean flux of the Northern hemisphere, and black squares represent globally monthly mean flux. Uncertainties are estimated based on top 10 models with different parameter combinations. Errorbars correspond to $\pm 1\sigma$.

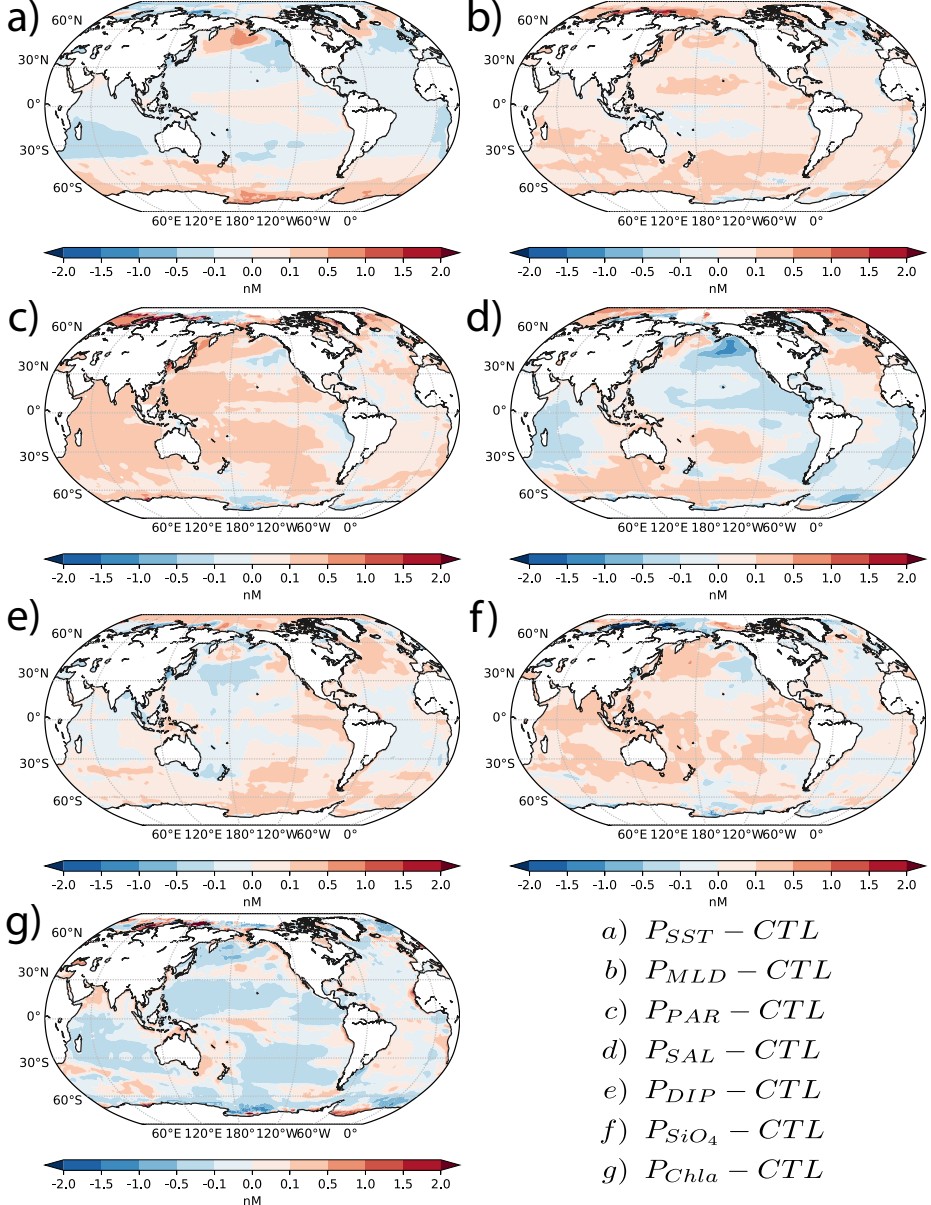

**Figure 9.** Differences of annul mean DMS concentration between perturbation models and the control model. Specific figure indexes are listed in the figure, where $P_{xxx}$ represents a perturbed model and the subscript $_{xxx}$ indicates which parameter is changed. $CTL$ is the control model that is the average of our top 10 model results (Fig. 5).

**Table A1.** DMS and ancillary data sources.

| Variables | Sources | units | References |
|---|---|---|---|
| DMS[1] | http://saga.pmel.noaa.gov/dms/ | nM | (Kettle et al., 1999) |
| DMS[2] | NAAMES | nM | (Behrenfeld et al., 2019) |
| Chl | https://oceandata.sci.gsfc.nasa.gov/SeaWiFS/ | $\mu$g L$^{-1}$ | (NASA, 2018) |
| MLD | https://www.pmel.noaa.gov/mimoc/ | m | (Schmidtko et al., 2013) |
| PAR | https://oceancolor.gsfc.nasa.gov/atbd/par/ | Einsteins m$^{-2}$ d$^{-1}$ | (Frouin et al., 2012) |
| WSP | https://podaac.jpl.nasa.gov/dataset | m s$^{-1}$ | (NASA, 2012) |
| SST | WOA2013 | C | (Garcia et al., 2013) |
| SSS | WOA2013 | psu | (Garcia et al., 2013) |
| DIP | WOA2013 | $\mu$M | (Garcia et al., 2013) |
| DIN | WOA2013 | $\mu$M | (Garcia et al., 2013) |
| SiO | WOA2013 | $\mu$M | (Garcia et al., 2013) |
| ICE | CESM model | - | (Wang et al., 2019) |

[1] Data from the online database. [2] New data from the North Atlantic Aerosol and Marine Ecosystems experiment.

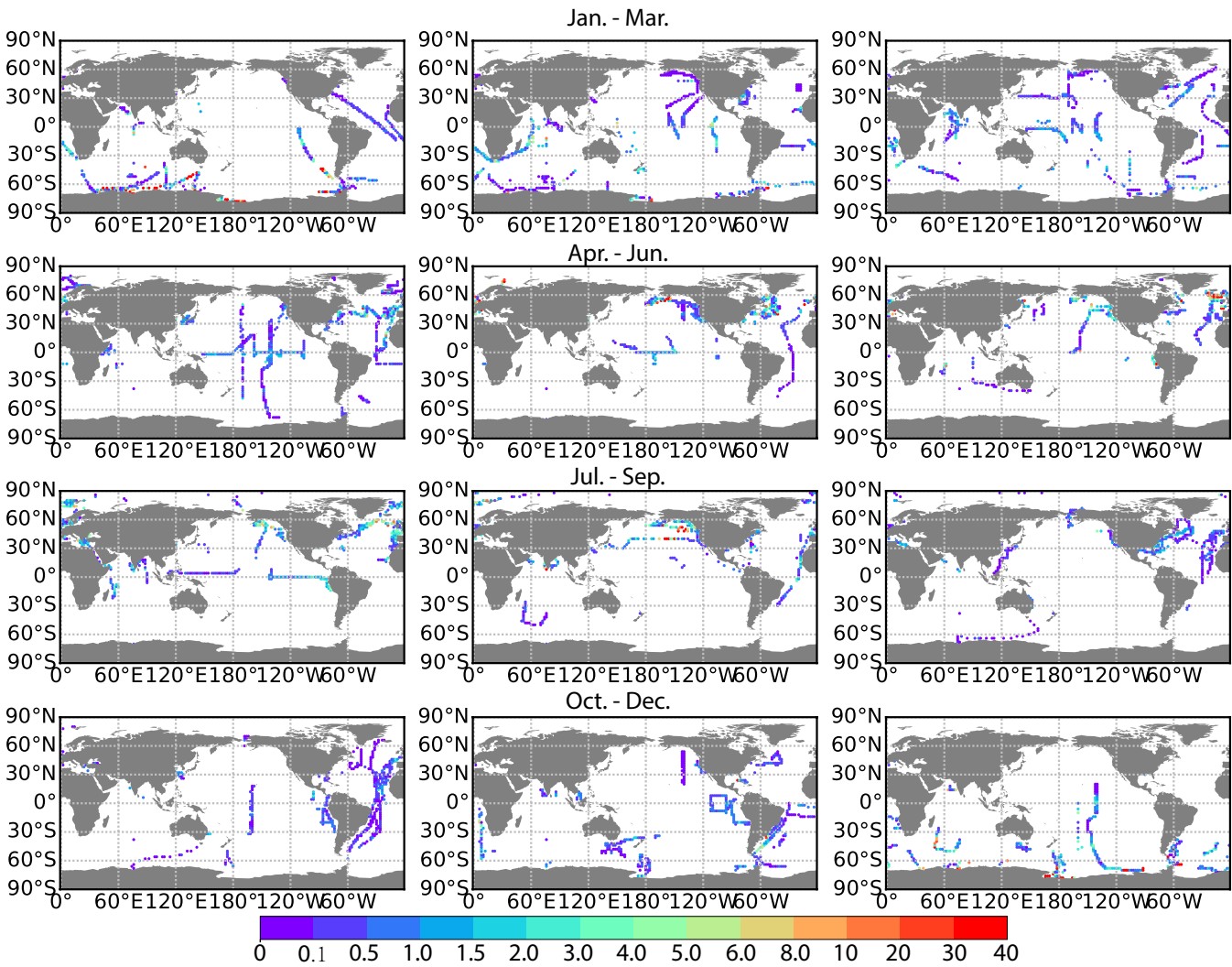

**Figure A1.** Distribution of DMS observations partitioned into each month. The color indicates DMS concentration (nM).

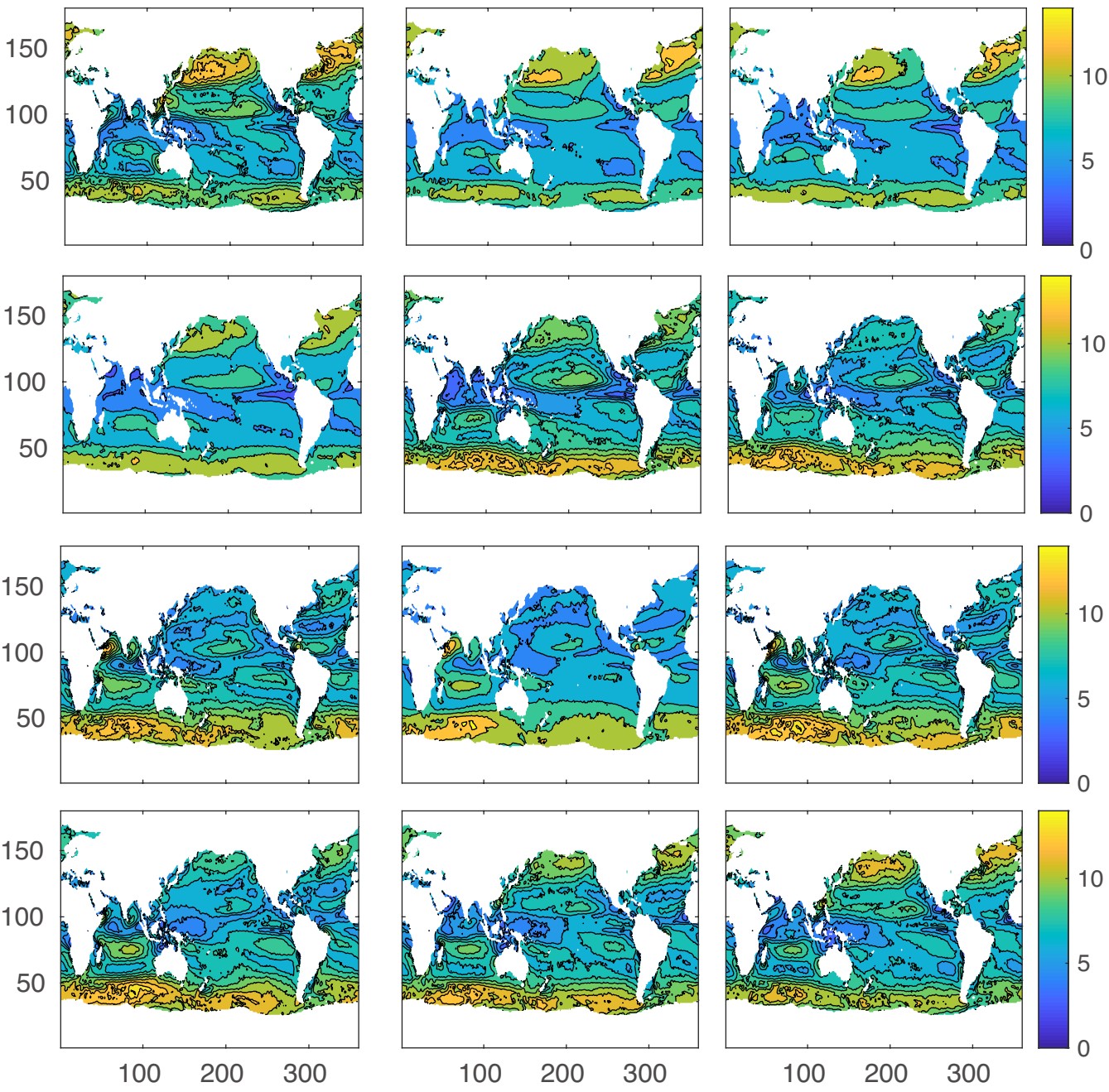

**Figure A2.** Climatological wind speed (m s$^{-1}$).