# Peer review of "Global ocean dimethyl sulfide climatology estimated from observations and an artificial neural network"

_Biogeosciences, 2020_

## Referee Comment (RC1) · Martí Galí (Referee) · 2 Apr 2020

**Review of the manuscript submitted to Biogeosciences Discussions: "Global ocean dimethyl sulfide climatology estimated from observations and an artificial neural network",** by Wei-Lei Wang, Guisheng Song, François Primeau, Eric S. Saltzman, Thomas G. Bell, and J. Keith Moore.

**General comments**

The manuscript by Wang and coauthors proposes an interesting methodological development: the use of artificial neural networks (ANN) to produce a global gridded climatology of dimethylsulfide (DMS) concentration at the sea-surface. This is a relevant topic because DMS emission drives aerosol formation in the remote marine atmosphere, with subsequent effects on aerosol and cloud radiative forcing. Measurements of sea-surface DMS concentration are too sparse to be directly usable in studies of atmospheric chemistry and global sulfur biogeochemistry. Therefore, a number of techniques have been used in the past to produce global gridded DMS fields: from objective interpolation (on which the standard climatological product is based; Lana et al., 2011) to empirical remote sensing algorithms or prognostic ocean biogeochemistry models.

The "artificial intelligence" approach proposed by Wang and coauthors is a necessary step to improve existing global DMS products. The article is generally well written and I appreciated the succinct style. However, in its current form the study suffers from a number of important shortcomings:

- Repetition of results that have been presented more in depth in previous papers. The results of the correlation analysis between DMS(P) and environmental variables are of little interest as they are extremely similar to those reported in previous papers, where they were analyzed more in depth and with a more solid theoretical underpinning. The stepwise multilinear regression (sMLR), which is used mainly to contrast its limited predictive power against the greater predictive power of the ANN, is only partially described.
- 2. Failure to perform appropriate quality control of the raw DMS, DMSP and chlorophyll (Chl) data, for example following procedures detailed by Galí et al. 2015 for the same global DMS database used by Wang et al. The main flaw is the use of in situ fluorometric Chl measurements and satellite-retrieved Chl as if they were equivalent –they are not.
- 3. Inaccurate reasoning regarding the utility of data binning for the purpose of calculating monthly climatologies. What is the value of using raw (non-binned) measurements if (1) most of them are matched to climatological fields of the predictor variables? and (2) the final purpose is calculating a monthly climatology at coarser spatial resolution, which by definition aims at smoothing out interannual and small-scale variability? For example, to what extent the introduction of more than 10,000 new measurements, taken at high resolution in a relatively small region that was already quite well documented (temperate NW Atlantic), adds relevant information when it comes to computing monthly climatologies? Wouldn't it be more appropriate to bin all measurements beforehand to the coarsest resolution at which predictor variables are available, and then train the ANN? The authors should treat these issues more accurately and provide evidence for the advantages (if any) of using raw DMS data (including high-resolution transects), e.g. comparing statistics for ANN trained on raw vs. binned data. This being said, I agree that capturing the weight of extremes, ie the non-normal statistical distribution of DMS, is important (Galí et al., 2018).
- 4. Limited discussion of the advantages of the ANN approach, especially in regions that are challenging for prognostic and empirical models. For example, the ANN method does not outperform the gridded climatology (Lana et al., 2011) in the subarctic Northeast Pacific in August and September, when DMS concentrations are much higher than what one would expect based on global-scale relationships. If the ANN does not outperform the (admittedly limited) objective interpolation approach in a region that contains data, how can we trust ANN predictions in regions with no DMS measurements? An analysis comparing seasonal

DMS patterns across different biogeochemical regimes (eg ANN vs. objective interpolation and remote sensing algorithms) would be very welcome and would strengthen the arguments for adopting the ANN as a standard method to compute climatologies.

5. Misuse or inappropriate citation of some key references (e.g. Simó & Dachs 2002, Toole and Siegel 2004) and, more generally, omission of relevant references from the past 10-20 years. I think the view of marine DMS(P) cycling presented in this article is a bit outdated, especially regarding (1) upper-ocean DMS budgets and DMS turnover times due to biological processes, which ultimately control concentrations, and (2) the role of heterotrophic organisms and processes, which decouple DMS from phytoplankton abundance and taxonomy in much of the global ocean.

In addition to the points above, I have to admit that approaches such as ANN leave me, as a reader, with the feeling I did not learn much about the global distribution of DMS and its controlling factors. This would not necessarily be a criticism as long as the choices the authors made to configure and tune the ANN were sufficiently justified, but I missed some depth of information in this regard. The ANN itself remains a mysterious black box to me and, even if the overall results look reasonable (with exceptions, as highlighted above), I am unable to appreciate whether Wang and coauthors made an optimal implementation of the ANN.

I prompt the authors to address the formal, conceptual and technical criticisms made above. I honestly hope these constructive criticisms will improve the study and, more broadly, pave the way towards sensible implementation of AI techniques to compute DMS climatologies (which will likely become routine in the near future).

Finally, note that in the specific comments below I will frequently refer to my own papers, simply because some of them are very relevant for the present study and, in some cases, the only ones available. Of course, the authors are free to decide what citations they incorporate. For all these reasons, and for the sake of transparency, I decided to sign the review.

Martí Galí

**Specific comments**

**Abstract**

Please reshape taking into account the main criticisms, especially concerning the amount of variance captured when using raw or binned data (general point 3). Raw DMS data variance could be biased towards high-resolution data representative of small-scale variability if no homogenization of the spatial-temporal scales covered by the measurements is applied (the PMEL database consists mostly of coarse resolution data).

**Introduction**

L19: OK, but the approach proposed here does not reveal the factors controlling DMS variability. Rephrasing suggested.

L24: If the authors insist on the mechanistic point of view (not sure is the right line of thought in this paper), I suggest adding "process rate measurements" here. They hamper predictive models even more strongly than limited observations of DMS concentration.

L30: To the best of my knowledge, the term "summer paradox" was coined by Simó & Pedrós-Alió 1999 (Nature), so I suggest crediting them for it.

L49: Relevant citations here are Le Clainche et al. 2010 (GBC) (S cycling model inter-comparison seeking to understand the processes responsible for the summer paradox) and Tesdal et al. 2016

(Env Chem) (the most extensive comparison among gridded climatologies, empirical and prognostic models published so far, to my knowledge).

L61: DMS is produced by some marine algae, some bacteria, and mostly as a result of food web interactions (Kiene et al. 2000; Simó et al. 2001, Stefels et al. 2007, Curson et al. 2011, Moran et al. 2012, etc.). Please nuance and refine.

**Methods**

L73: Over 10,000 measurements came from NAAMES alone. Not binning the data might give too much weight to particular conditions sampled during NAAMES, at least in the multilinear regression.

L75: The DMSPt, Chla, SST and SSS data in the PMEL database require some quality control. This was documented by Galí et al. 2015. Quality controlled datasets with stringent satellite match-up criteria (ie minimizing the use of climatological coarse resolution data), as well as a piece of code used to clean data, are publicly available on github: https://github.com/mgali/DMS-SAT\_DATA\_DEV\_VAL.

L75: Fluorometric Chl is on average around 40% higher than HPLC Chl (Sathyendranath et al., 2009), and the proportion sometimes varies quite a bit. Satellite Chl is validated against HPLC measurements (e.g. Morel et al., 2007).

L77: Less than 3.5 years do not make a good Chl climatology in many ocean areas in my experience. Data products covering much longer periods are available on NASA's ocean colour website. Please update datasets, and specify also what reprocessing was used.

L79: The more recent climatology of Holte et al. 2017 seems to outperform that of Schmidtko et al. 2013 in areas of deep winter convection (subpolar North Atlantic) or where deep mixing prevails (Southern Ocean circumpolar current). In some cases the differences are important. Please consider using the Holte et al. 2017 MLD climatology.

L81: Please specify the nutrient datasets used.

L105-107: I see some contradiction here. Data extremes typically arising from nonlinear dynamics are often smoothed out when averaging data. Your predicted variable (DMS) retains full variability but predictor fields do not, because apart from SST they largely originate from monthly climatologies. How can meaningful nonlinear relationships be identified?

L128-130: Are these parameters default ones, or tuned manually to achieve reasonable fits in this particular study?

L137: Inclusion of time of day is interesting, although diel variability was not been mentioned earlier in the manuscript. Are hourly predictions useful for computing climatologies? Although DMS can oscillate on diel time scales (Galí et al., 2013; Royer et al., 2016) diel cycles do not seem to follow a fixed pattern, at least in low-latitude high-resolution datasets (e.g. Royer et al., 2015).

**Results**

L170 and 177: Note Galí et al. (2018) reported an R2 of 0.42 (r of 0.65) between DMS and DMSPt using the same datasets with stricter quality control. Similarly for DMS vs. Chl: R2 of 0.20 (r of 0.45).

L172-176: These sentences look a bit contradictory and may need further elaboration (or else, can be removed). Is it straightforward or not to predict DMS from DMSPt? Are measurements sufficient or not? Regarding DMSP prediction, another relevant study is that by McParland & Levine (2018). Regarding the relationship between DMSPt and DMS in the global PMEL database, Galí et al. (2018) is a relevant reference.

L179: The weaker relationship between DMS and Chl in the entire dataset probably results from the higher proportion of oligotrophic low latitude data (where DMS is anticorrelated to Chl over the seasonal cycle) compared to in situ Chl-DMS data pairs. The difference between in situ fluorometric Chl and satellite Chl may also play a role. Finally, note that the global PMEL DMS database is biased

towards productive conditions (Galí et al. 2018; figure 7) which influences global DMS-Chl correlations. In summary, the correlation between DMS and Chl in global datasets is not really meaningful as it depends strongly on how evenly represented are the different ocean biomes.

L187-190: This is incorrect. Dilution is not the main explanation for the negative relationship between MLD and DMS (as originally proposed by Aranami and Tsunogai, 2004). The main explanation is the different balance between biological DMS sources and sinks, as explained by Galí & Simó (2015). In the handful of studies that have made DMS budgets including the vertical mixing term, vertical DMS transport has never been found to dominate DMS budgets in the MLD over relevant (~daily) time scales. Check for example Bailey et al. 2008 (DSR), Herrmann et al. 2012 (CSR), Galí et al. 2013 (GBC), Royer et al. 2016 (Sci Rep), etc. DMS turnover in the surface layer due to vertical transport is generally an order of magnitude slower than biological turnover or biological + photochemical turnover, at least. Please correct.

L191-199: The strongest evidence for light-driven DMS production in natural plankton assemblages comes from recent work by myself and colleagues (Galí et al. 2013a, b, c; Royer et al. 2016). Evidence for light-driven DMS production in Toole et al. (2006) (otherwise, a great piece of work!) is indirect as that study focused on DMS removal processes.

Section 3.2: see general comment 3 on data binning.

L201-204: It is unclear here if the authors made the appropriate comparisons with Simó & Dachs 2002 and Vallina & Simó 2007 empirical models. Was DMS compared with surface PAR or with the solar radiation dose in the mixed layer as done by Vallina and Simó 2007? Similarly, did the authors correctly apply the double algorithm used by Simó and Dachs 2002, where different equations are used depending on the value of ChI/MLD? Or just computed a single regression of DMS against ChI/MLD?

Section 3.3: Methods section mentions 8 initial variables (PAR, MLD, Chl a, SSS, SST, DIN, DIP, and SiO), but, what predictor variables were included in the final multilinear regression model (MLR)? Does the R2 of the MLR refer to linear of log space?

Section 3.4: Since this subsection describes the main technical innovation of this paper, a deeper explanation of why the ANN gives these results would be very welcome. See general comment 4.

Section 3.5: For the authors' information, global DMS fields produced with the remote sensing algorithm of Galí et al. 2018, as well as the algorithms of Simó & Dachs 2002 and Vallina & Simó 2007, are available in this repository: https://doi.org/10.5281/zenodo.2558511. Corresponding Matlab and R codes are available on a linked github repository:

https://github.com/mgali/DMS-SAT\_ALGORITHM.

L235: These references are not appropriate here. Please cite studies that actually documented DMS(P) dynamics in subpolar or polar blooms of coccolithophores or *Phaeocystis*.

L247-248: I do not find reasonable that DMS decreases below 0.1 nM in a subtropical gyre in winter. By examining the maps in Fig. 3 I would say ANN DMS is mostly between 0.1 and 0.5 nM, which still looks a bit low but more realistic according to my experience. DMS concentrations lower than 0.1 nM are extremely rare in both the PMEL database and in global estimates made with empirical algorithms (see Galí et al. 2018 figure 7).

L250-254: Please check Galí and Simó 2015 (GBC) for a mechanistic explanation of the summer paradox.

In general, I suggest making a figure showing the climatological seasonal cycles in different ocean biomes or regions, to better support the description made in the text.

L273-293: Here I strongly suggest citing Tesdal et al. 2016.

Figures

Figure 3 and 5: I suggest using a color scale with different colors to help readers appreciate concentration patterns.

Figure 6: I strongly recommend splitting results into northern and southern hemisphere given the strong seasonality of DMS (also wind speed and SST), which results in opposed seasonal patterns.

**Minor corrections**

L131: What does "epochs" mean in this context? Please use synonym for readers that are not expert in ANN or similar techniques.

L218: The "tracer-tracer" term is not needed here (quite specific to bgc modelling).

**Reviewer references (only if not cited by the authors)**

Aranami, K., & Tsunogai, S. (2004). Seasonal and regional comparison of oceanic and atmospheric dimethylsulfide in the northern North Pacific: Dilution effects on its concentration during winter. *Journal of Geophysical Research: Atmospheres, 109*(D12).

Bailey, K. E., Toole, D. A., Blomquist, B., Najjar, R. G., Huebert, B., Kieber, D. J., ... & Del Valle, D. A. (2008). Dimethylsulfide production in Sargasso Sea eddies. *Deep Sea Research Part II: Topical Studies in Oceanography*, *55*(10-13), 1491-1504.

Curson, A. R., Todd, J. D., Sullivan, M. J., & Johnston, A. W. (2011). Catabolism of dimethylsulphoniopropionate: microorganisms, enzymes and genes. *Nature Reviews Microbiology*, *9*(12), 849-859.

Galí, M., Ruiz-González, C., Lefort, T., Gasol, J. M., Cardelús, C., Romera-Castillo, C., & Simó, R. (2013). Spectral irradiance dependence of sunlight effects on plankton dimethylsulfide production. *Limnology and oceanography*, *58*(2), 489-504.

Galí, M., Simó, R., Vila-Costa, M., Ruiz-González, C., Gasol, J. M., & Matrai, P. (2013). Diel patterns of oceanic dimethylsulfide (DMS) cycling: Microbial and physical drivers. *Global Biogeochemical Cycles*, *27*(3), 620-636.

Galí, M., Simó, R., Pérez, G., Ruiz Gonzalez, C., Sarmento, H., Royer, S. J., ... & Gasol, J. M. (2013). Differential response of planktonic primary, bacterial, and dimethylsulfide production rates to static vs. dynamic light exposure in upper mixed-layer summer sea waters.

Galí, M., & Simó, R. (2015). A meta-analysis of oceanic DMS and DMSP cycling processes: Disentangling the summer paradox. *Global Biogeochemical Cycles*, *29*(4), 496-515.

Herrmann, M., Najjar, R. G., Neeley, A. R., Vila-Costa, M., Dacey, J. W., DiTullio, G. R., ... & Vernet, M. (2012). Diagnostic modeling of dimethylsulfide production in coastal water west of the Antarctic Peninsula. *Continental Shelf Research*, *32*, 96-109.

Holte, J., Talley, L. D., Gilson, J., & Roemmich, D. (2017). An Argo mixed layer climatology and database. *Geophysical Research Letters*, 44(11), 5618-5626.

Kiene, R. P., Linn, L. J., & Bruton, J. A. (2000). New and important roles for DMSP in marine microbial communities. *Journal of Sea Research*, *43*(3-4), 209-224.

Le Clainche, Y., Vézina, A., Levasseur, M., Cropp, R. A., Gunson, J. R., Vallina, S. M., ... & Bopp, L. (2010). A first appraisal of prognostic ocean DMS models and prospects for their use in climate models. *Global biogeochemical cycles*, *24*(3).

Moran, M. A., Reisch, C. R., Kiene, R. P., & Whitman, W. B. (2012). Genomic insights into bacterial DMSP transformations. *Annual review of marine science*, *4*, 523-542.

Morel, A., Huot, Y., Gentili, B., Werdell, P. J., Hooker, S. B., & Franz, B. A. (2007). Examining the consistency of products derived from various ocean color sensors in open ocean (Case 1) waters in the perspective of a multi-sensor approach. *Remote Sensing of Environment*, *111*(1), 69-88.

Royer, S. J., Mahajan, A. S., Galí, M., Saltzman, E., & Simó, R. (2015). Small-scale variability patterns of DMS and phytoplankton in surface waters of the tropical and subtropical Atlantic, Indian, and Pacific Oceans. *Geophysical Research Letters*, *42*(2), 475-483.

Royer, S. J., Galí, M., Mahajan, A. S., Ross, O. N., Pérez, G. L., Saltzman, E. S., & Simó, R. (2016). A high-resolution time-depth view of dimethylsulphide cycling in the surface sea. *Scientific reports*, *6*, 32325.

Sathyendranath, S., Stuart, V., Nair, A., Oka, K., Nakane, T., Bouman, H., ... & Platt, T. (2009). Carbon-to-chlorophyll ratio and growth rate of phytoplankton in the sea. *Marine Ecology Progress Series*, *383*, 73-84.

Simó, R., & Pedrós-Alió, C. (1999). Role of vertical mixing in controlling the oceanic production of dimethyl sulphide. *Nature*, 402(6760), 396-399.

Tesdal, J. E., Christian, J. R., Monahan, A. H., & von Salzen, K. (2016). Evaluation of diverse approaches for estimating sea-surface DMS concentration and air–sea exchange at global scale. *Environmental Chemistry*, *13*(2), 390-412.

---

## Referee Comment (RC2) · Anonymous Referee #2 · 6 Apr 2020

The manuscript proposes a new global ocean DMS climatology, or a method to construct it, based on an Artificial Neural Network (ANN). This methodology uses a number of variables and their intelligent combinations as predictors of DMS concentration distribution. It is meant to overcome the limitations of objective analysis based on inter- and extrapolations as well as the limitations of simple linear or logarithmic regressions with few predictors, and to provide better fits of predictions to observations. While developing their ANN application and to claim its better performance, the authors conduct parallel applications of previously published models. Eventually, they indeed obtain a better fit, but very similar seasonal and geographic distributions. The global annual emission to the atmosphere is revised towards the lower end of the hitherto most accepted estimate.

The topic is timely since, after years of having DMS been dismissed for its role in new particle formation, recent studies are recognizing it again as a central agent in ocean-atmosphere-climate interactions. Atmospheric chemistry and climate models require updated climatologies of DMS emissions.

The text is generally well written and the diplay items are clear and informative, with one exception (see particulars below).

That said, the manuscript reads as though it was written 10 years ago. Even though the ANN methodology is probably state-of-the-art (I am not an expert and can hardly assess every technical aspect), the interpretation arguments are outdated, ignoring many of the discoveries in the last decade. This adds to some bad referencing. But most importantly, when the authors intend to make relevant comparisons with previous similar efforts, they miss the point of the studies they are comparing to, or use them in the wrong way. Finally, besides presenting their new method, they fail to discuss what is new in their findings, they just repeat what is already well known and with much poorer arguments, rather than stressing what is unveiled and why. I will develop these and other concerns hereafter, as they come up in the order of the manuscript.

L28-30: "The weak relationship may be caused by the so-called "summer DMS paradox", which describes a phenomenon where a maximum DMS concentration is commonly detected in low latitude waters when phytoplankton biomass is low (Toole and Siegel, 2004; Vallina et al., 2008)." This is not the summer DMS paradox (a term, by the way, suggested by Simo & Pedros-Alio Nature 1999), which states that the annual maximum of surface DMS commonly occurs in summer, even at the mid and subtropical latitudes where chlorophyll-a (chl-a) is at its annual minimum.

L34-35: "Simó and Dachs (2002) achieved a strong relationship between heavily binned and averaged DMS data and mixed layer depth (MLD)." This is not true. Simo & Dachs (2002) correlated DMS to the MLD and to chl-a/MLD, depending on a chl-a/MLD

threshold.

L53-54: "Many provinces lacked adequate data to create a reliable climatology (Fig. A1). In those situations, temporal interpolations were used to fill the blanks, and to create a first-guess map." This was done where monthly data gaps existed to complete the seasonality. Where data were lacking to even outline a seasonality, this was taken from a neighboring province and adjusted to the existing data.

L61: "Since DMS is produced by marine that algae..." This is totally outdated. There are tens of papers showing that this is an oversimplification. DMSP is mainly produced by marine algae, and it is transformed into DMS by marine algae, bacteria and with involvement of zooplankton.

L93-94: "We do not log-transform SST to avoid losing data with temperature below (equal to) zero." You may have other reasons to not log transform SST, but not this one. A common practice to log transform SST if desired is to convert it to K (Kelvin) first.

If I understand it correctly, you use chl-a data where available, otherwise you take it from SeaWiFS. What efforts have you done to reconcile in situ with satellite chl-a? It is well known that algorithms for satellite estimates of chl-a are developed and calibrated against HPLC chl-a, and there is an important shift between this and Turner fluorometric chl-a. Therefore, putting together in situ (Turner, perhaps HPLC too?) and satellite chl-a data will mess up your statistics.

Calculation of air-sea fluxes: I agree that Nightingale 2000 is quite a standard. But, why not using a more updated linear relationship of Kw to u10? Marandino proposed one with one of the coauthors. Also, you use monthly means of wind speed. Since you are using a nonlinear dependence of Kw on the u10, how do you deal with the fact that a mean u10 will not give the same result as a mean Kw?

L170-176: It reads as though you did not know of the existence of Gali et al. BGS 2016.

L182: "On the other hand, negative correlations between DMS and Chl a have also been detected in coastal waters of the Mediterranean and in the Sargasso Sea (Toole and Siegel, 2004)." Toole & Siegel did not do anything with Med Sea data. The original data from the Sargasso Sea were from Dacey et al DSR 1996, and data from the coastal Med Sea were reported by Vila-Costa et al. LO 2008.

L185-190: This is a very poor interpretation of the DMS vs MLD coupling, and a misuse of the original relationship suggested by Simo & Dachs GBC 2002. As a matter of fact, you cite Simo & Pedros-Alio GBC 1999 because they brought it up for the first time, but the occurrence of a negative relationship between DMS and MLD over large regions of the global ocean was reported by Simo & Dachs. However, the relationship was logarithmic, DMS = aÂůLn(MLD) + b, and there are reasons for this to occur, related to exposure to solar radiation. Trying to correlate DMS directly to MLD (or in a log-log manner) is not expected to provide good prediction.

L189-199: There are a number of papers that should be cited here – besides Toole et al. and Sunda et al, sevarl papers by Marti Gali deal exactly with the effects of solar radiation, and particularly UV, on enhancing DMS production and concentration.

"Climatological PAR is the second strongest predictor (R2 = 0.12, n = 54,683) of raw DMS data with a positive correlation. (. . .) Strong correlation between monthly binned and averaged solar radiation dose (SRD) and DMS concentration has been reported (R2 = 0.94) at the Blanes Bay Microbial Observatory located in the coast of northwest Mediterranean (Vallina and Simó, 2007)." Again, you compare your statistics with that of a previous study, but applying a different calculation. According to the methods description, you used monthly PAR, i.e., monthly surface irradiance. Vallina & Simo 2007, conversely, computed what they called the solar radiation dose, which is the daily averaged solar radiation integral in the mixed layer. This is very different from surface irradiance, because it takes into account the mixed layer depth (and a median light attenuation coefficient). Later on, in L200-211, you infer that, contrasting to Vallina & Simo, you did not get a good correlation to light, and attribute it to the number of

original data and to data binning. But you did not use the same light metrics as the other authors, and ignored the arguments given by V&S to use the SRD instead of the surface irradiance, and ignoring the Gali & Simo GBC 2015 meta-analysis too.

L201: "Simó and Dachs (2002) obtained a high R2 value between DMS concentration and the ratio of Chl a and MLD (Chl/MLD)." This is not true. As already mentioned above, the Simo & Dachs (2002) model correlated DMS to the MLD (logarithmic) and to chl-a/MLD (linear), depending on a chl-a/MLD threshold.

All in all, if you are to compare your statistics to those of S&D 2002 and V&S 2007, everything here has to be recomputed and rewritten.

The arguments against binning the data are poor. It is true that binning reduces the variance, but you are using monthly climatologies (heavily averaged and also binned) to relate raw DMS data to potential predictors. Also, binning must be used if you want to avoid giving too much predictive weight to the regions thoroughly sampled over the undersampled. This is becoming more important as we are bringing in underway data at unprecedented spatial resolution, like the NAAMES data incorporated here.

L231-233: "The summertime high DMS concentration at high latitudes is consistent with the hypothesis that phytoplankton use DMSP as a cryoprotectant (Karsten et al., 1992). It is found that the same phytoplankton (Antarctic macroalga) contains higher DMSP concentration in the polar regions than in the temperate regions (Karsten et al., 1990)." Poor again, if not wrong. See recent papers on DMS in polar regions (e.g. Webb et al Sci Rep 2018, Gali et al. PNAS 2019). And macroalgae are not phytoplankton.

Subsequent discussion: The seasonality and geographic distribution of DMS have been profusely (and much better) discussed by Lana et al. GBC 2011 and others, including regional studies. You should rather focus on new features unveiled with respect to others, particularly Lana 2011.

L305-306: "By contrast, objective interpolation methods are spatial/temporal averages of sparse data with no underlying basis in environmental variability." Again, this is not totally true. In Lana et al. 2011, to create a first guess field, biogeographic provinces were used, which is an informed approach to extrapolation. These provinces are defined from environmental descriptors. And a distance weighted interpolation from original data was used for interpolation.

Figure 2: An annual average is not very informative. I would even argue it is misleading in the case of highly seasonal variables like DMS, because summer maxs and winter mins cancel out each other. I would recommend splitting the map into two or four seasons to show hemispherical patterns.

Figure 4: Some differences are outstanding but you do not discuss them. For instance, Lana 2011 captures the September max of DMS concentration in the subarctic NE Pacific, because it is well covered with data. Conversely, your ANN does not capture it. This warrants some discussion, as it will reveal some of the caveats of the ANN approach.

In summary, I think that the ANN is an interesting approach that will help improve DMS (and other) climatologies, especially where data are lacking, as it will do better than inter- and extrapolations. However, the present manuscript does not go much beyond the application of the ANN; when it intends to do so, too often it uses the wrong arguments and is not fair with previous studies. It fails to mobilize what we have learned about DMS in the last one or two decades.

---

## Referee Comment (RC3) · Anonymous Referee #3 · 23 Apr 2020

Review of Wang et al., Global ocean dimethyl sulfide climatology estimated from observations and an artificial neural network

This manuscript describes a novel methodology for deriving a global ocean dimethyl sulfide (DMS) climatology, using an artificial neural network (ANN). The authors demonstrate that the ANN is able to explain a greater fraction of variance in the raw available observations of surface ocean DMS concentrations, as compared with a multiple linear regression approach. They also contrast this approach with previous work that used spatial and temporal gap-filling to estimate DMS concentrations, including in data-sparse regions. Instead, the approach presented here derives relationships between observed environmental parameters and observed oceanic DMS DMS concentrations (using the multiple regression or ANN), and uses these to predict/extrapolate DMS concentrations globally.

The paper is clearly written, the methods are straightforward and appropriate, and it represents a valuable contribution to work on understanding and representing the present-day climatological distribution of DMS concentrations in the surface ocean. Improved climatologies of DMS would be useful for Earth System models, especially if they can offer more insights into how the DMS production would change under past/future climate states. It's unclear (to me, at least) whether a machine learning approach will be able to offer such physical insights. Nevertheless, such approaches can offer a better estimate of the present-day state, and this is useful in itself for Earth System modeling. The uncertainty in ocean DMS climatologies is still quite large, despite advances during the past decade, and new advances in statistical approaches that can reduce errors in these datasets are welcome.

I have only a few minor comments, as follows:

I agree with the comments of the two previous reviewers that the arguments made against data binning are weak. The authors imply that it is an inherently inferior approach, but, this is not necessarily true a prior. There can be good arguments in favor of data binning before analysis, e.g., to harmonize the temporal and spatial scales of multiple datasets before analyzing the relationships between them. When in situ DMS measurements (essentially instantaneous) are being predicted via monthly mean values of chl-a, MLD, etc., it is not at all obvious that it is appropriate to perform the analysis without first binning the data. This point should be treated with more nuance, taking into account the details of the datasets and the processes involved.

p. 5, l. 128-130: I was glad to see that the authors have considered the issue of potential overfishing, but they don't explain how they determined that the setup they used for the ANN is not overfitting (i.e., what methods or criteria were used to determine

this). It's common to use multiple rounds of cross-validation (such as k-fold cross-validation or related methods) in order to determine whether a statistical model may be overfitting and to assess the uncertainty in the fit. If I am understanding the description of the method correctly, it seems that while the authors divided the data into training and validation subsets, they did so only once. In this case, the results of the ANN will be sensitive to the specific subset of data that was used for training it. It should be explained how the training/validation subsets were selected, and also whether a multi-round cross-validation method was employed (and if not, why not). Or, if appropriate, the authors could simply carry out a more thorough cross-validation and update the manuscript, since I expect this should not require much effort.

p. 5, l. 133-134: It was not obvious to me what the "random states" refer to – is this a random seed controlling initial parameter values?

p. 8, l. 220: here, it is stated that ANN is able to "capture more of the variance" than "previous extrapolations (Kettle et al., 1999; Lana et al., 2011)". This is a key claim of the paper in terms of the claimed improvement over previous methods, and I can believe this is probably true, but I think the claim ought to be supported by a quantitative value – i.e., the percentage of variance captured by the two previous climatologies – so that readers can compare and see the improvement in this metric. Perhaps these values are in the manuscript somewhere and I overlooked them – in that case I think they should be featured somewhere that is easier to find (e.g., in the abstract or in a table).

p. 11: I tested the links for the code and data availability; the data doi link at zenodo works, but the github link does not seem to be available.

I also noticed a couple of typos: p. 2, l. 40: "result" -> "results" or "result[s]" p. 5, l. 31: "deduction" -> "reduction" p. 5, l. 133: "assemble" -> "ensemble" (?) p. 7, l. 189: "wasters" -> "waters"

---

## Author Comment (AC1) · 7 Jun 2020

[i]Review of the manuscript submitted to Biogeosciences Discussions: "Global ocean dimethyl sulfide climatology estimated from observations and an artificial neural network", by Wei-Lei Wang, Guisheng Song, François Primeau, Eric S. Saltzman, Thomas G. Bell, and J. Keith Moore.

General comments
The manuscript by Wang and coauthors proposes an interesting methodological development: the use of artificial neural networks (ANN) to produce a global gridded climatology of dimethylsulfide (DMS) concentration at the sea-surface. This is a relevant topic because DMS emission drives aerosol formation in the remote marine atmosphere, with subsequent effects on aerosol and cloud radiative forcing. Measurements of sea-surface DMS concentration are too sparse to be directly usable in studies of atmospheric chemistry and global sulfur biogeochemistry. Therefore, a number of techniques have been used in the past to produce global gridded DMS fields: from objective interpolation (on which the standard climatological product is based; Lana et al., 2011) to empirical remote sensing algorithms or prognostic ocean biogeochemistry models.

Thank you for the positive words.

The "artificial intelligence" approach proposed by Wang and coauthors is a necessary step to improve existing global DMS products. The article is generally well written and I appreciated the succinct style. However, in its current form the study suffers from a number of important shortcomings:

1. Repetition of results that have been presented more in depth in previous papers. The results of the correlation analysis between DMS(P) and environmental variables are of little interest as they are extremely similar to those reported in previous papers, where they were analyzed more in depth and with a more solid theoretical underpinning. The stepwise multilinear regression (sMLR), which is used mainly to contrast its limited predictive power against the greater predictive power of the ANN, is only partially described.

Response: Our goal with the sMLR model is indeed to gradually demonstrate that ANN is better than traditional linear/multilinear models. In the revised version, we added more in-depth discussion of each model results, and added more details about the multilinear model. We also tried to minimize any repetition of previous findings.

2. Failure to perform appropriate quality control of the raw DMS, DMSP and chlorophyll (Chl) data, for example following procedures detailed by Galí et al. 2015 for the same global DMS database used by Wang et al. The main flaw is the use of in situ fluorometric Chl measurements and satellite-retrieved Chl as if they were equivalent –they are not.

Response: This is a good point. Thank you for pointing this out.
In the revised version, we followed the guideline introduced by Galí et al. 2015 to conduct quality control. Specifically, we removed DMS data with concentration less than 0.1 nM and greater than 100 nM, we also removed data with salinity less than 30 psu, so that we focus our

study on the open ocean. We removed DMSPt data that are less than 1 nM.  Other than that, we did not do any binning and averaging to preserve the original data variance. Finally, there are 10404 pairs of DMS-Chl-*a* and 4061 pairs DMS-DMSPt, which is substantially more than what was reported by Galí et al. 2018 (with 3637 DMS-DMSPt and 8141 DMS-Chl pairs). This is because, the PMEL database has been greatly expanded, it now has over 80K DMS data points.

For Chl a data, we have added the following discussion (l.84 – 1.92).
"SeaWiFS Chl-*a* data (Level 3-binned, spatial resolution of 9.2 km) from December 1997 to March 2010 were matched to DMS data according to coordinates and sampling date. We compared PMEL in situ Chl *a* to SeaWiFS Chl *a*, which are well correlated on logarithmic scale ($R^2$ = 0.64) with a slope of 0.67 and an intercept of -0.06, [$\log(Chl_{SeaWiFS}) = 0.67 \log(Chl_{in-situ}) - 0.01$], which means that on logarithmic scale SeaWiFS Chl-*a* concentrations are on average ~30% lower than those of in situ Chl-*a* concentrations. This is possibly because SeaWiFS Chl *a* is calibrated based on HPLC determined Chl *a* (Morel et al., 2007), which on average is ~40% lower than that determined using Fluorometric method (Sathyendranath et al., 2009). Unfortunately, there is no flag in the database showing how Chl *a* was determined. For consistency, we use only Chl-*a* data retrieved from SeaWiFS in the following multilinear and network models."

3. Inaccurate reasoning regarding the utility of data binning for the purpose of calculating monthly climatologies. What is the value of using raw (non-binned) measurements if (1) most of them are matched to climatological fields of the predictor variables? and (2) the final purpose is calculating a monthly climatology at coarser spatial resolution, which by definition aims at smoothing out interannual and small-scale variability? For example, to what extent the introduction of more than 10,000 new measurements, taken at high resolution in a relatively small region that was already quite well documented (temperate NW Atlantic), adds relevant information when it comes to computing monthly climatologies? Wouldn't it be more appropriate to bin all measurements beforehand to the coarsest resolution at which predictor variables are available, and then train the ANN? The authors should treat these issues more accurately and provide evidence for the advantages (if any) of using raw DMS data (including high-resolution transects), e.g. comparing statistics for ANN trained on raw vs. binned data. This being said, I agree that capturing the weight of extremes, ie the non-normal statistical distribution of DMS, is important (Galí et al., 2018).

Response: Since our initial submission, the PMEL database dramatically expanded. Now there are a total of 86,785 valid DMS measurements (concentration greater than 0.1 nM and less than 100 nM according to your instructions), that is 71% larger than the number of data we initially used (51,161). For the expanded data set, ~93% of DMS are accompanied with in situ SST measurements, ~81% are accompanied with in situ salinity measurements.  More importantly, each data point has their unique location and sampling time signatures. As shown in the following figure, sampling time (date) and location information is a strong DMS predictor, which together can decrease DMS root mean square error to 0.64 (on natural logarithm scale). Adding other climatological predictors can further improve the model performance.

The NAAMES dataset has 6,786 valid data points, which are ~7% of the total data points (93,571 = 86785+6786). All data are accompanied with in situ Chl $a$, SST, and SAL measurements. For parameters without in situ measurements, high resolution data are used to match DMS measurements, 0.0417° ×0.0417° for PAR, 0.5° ×0.5° for MLD, and 1° ×1° for NO3, which ensures most of DMS have a set of unique predictors. As shown in Table 1, merging NAAMES data with PMEL data does not significantly change the statistic.

Moreover, binning the data will reduce data variance, which has been demonstrated by Derevianko et al. (2009). The objective of this study is to train an ANN with as much data as possible, and let the ANN do any fitting. The statement "the final purpose is calculating a monthly climatology at coarser spatial resolution" is only partially true. The model can be applied to coarse resolution predictor fields, but also to very fine resolution predictor fields. For example, we have applied the network to fine resolution NAAMES fields for comparison with in situ DMS measurements (Bell et al., in preparation).

Lastly, binning data will necessarily result in loss of information. A great amount of information is associated with sampling location and date as shown in the following figure (Fig. 2a in MS). By binning the data into monthly 1°× 1° grid, the number of data points decreases significantly from 82,996 to only 9,018; sampling date feature (365) will be average to 12 months, and coordination combinations will be averaged from 87,332 × 87,332 to 180° × 360°, which represents a substantial loss of information. For ANN models, using less data points can lead to overfitting (See Fig. 2b).

[Figure]

Fig. 2 Parameter sensitivity tests on raw and binned data. (a) Root mean square error on logarithmic scale for the model trained using raw data; (b) Root mean square error on logarithmic scale for the model trained using binned data. The time and location parameters are tested separately without combining with environmental parameters as shown in the upper panel, (I) with only location parameters; (II) with location and day of year parameters; and (III) with location, day of year, and time of day parameters. The model with three location parameters (I) has a root mean square error on natural logarithmic scale of ~0.83, which decreases to ~0.65 by adding sampling day of year parameters (II), however, increases to ~0.67 by adding sampling time parameters (III). We, therefore, do not include sampling time parameters in the following tests. We tested every possible combination of the eight parameters (PAR, MLD, SST, SAL, Chl a, DIP, DIN, and SiO), which in total are 255 tests.

4. Limited discussion of the advantages of the ANN approach, especially in regions that are challenging for prognostic and empirical models. For example, the ANN method does not outperform the gridded climatology (Lana et al., 2011) in the subarctic Northeast

Pacific in August and September, when DMS concentrations are much higher than what one would expect based on global-scale relationships. If the ANN does not outperform the (admittedly limited) objective interpolation approach in a region that contains data, how can we trust ANN predictions in regions with no DMS measurements? An analysis comparing seasonal DMS patterns across different biogeochemical regimes (eg ANN vs. objective interpolation and remote sensing algorithms) would be very welcome and would strengthen the arguments for adopting the ANN as a standard method to compute climatologies.

Response: We have added extensive discussion about the ANN approach on page 9 (l. 259 - l.274).

We also have added discussion emphasizing on comparison with previous models (P.12 l. 342 – l.364 and Figs. 3 and 4).

5. Misuse or inappropriate citation of some key references (e.g. Simó & Dachs 2002, Toole and Siegel 2004) and, more generally, omission of relevant references from the past 10- 20 years. I think the view of marine DMS(P) cycling presented in this article is a bit outdated, especially regarding (1) upper-ocean DMS budgets and DMS turnover times due to biological processes, which ultimately control concentrations, and (2) the role of heterotrophic organisms and processes, which decouple DMS from phytoplankton abundance and taxonomy in much of the global ocean.

Response: Thank you for the suggestions. We have updated our reference and revised the biogeochemical description of DMS/DMSP.

In addition to the points above, I have to admit that approaches such as ANN leave me, as a reader, with the feeling I did not learn much about the global distribution of DMS and its controlling factors. This would not necessarily be a criticism as long as the choices the authors made to configure and tune the ANN were sufficiently justified, but I missed some depth of information in this regard. The ANN itself remains a mysterious black box to me and, even if the overall results look reasonable (with exceptions, as highlighted above), I am unable to appreciate whether Wang and coauthors made an optimal implementation of the ANN.

This is an important point. It motivated us to do more tests to help open the "black box" as discussed in the revised MS. (l.259 – l. 264).

"From traditional linear or multilinear models, one can easily figure out which parameter is a strong predictor and how a predictor influence the state variable (e.g. the correlation between DMSP and DMS). An ANN model is much more complex, it adjusts weights of each node that connect inputs and outputs, therefore, the relationship between inputs and outputs is subtle. That's why ANN models are generally referred to as a "Black Box". In this study, we design experiments that help open this "Black Box" and reveal parameters that drive surface ocean DMS distributions."

And please also read the subsequent discussion (l.264 – l.317).

I prompt the authors to address the formal, conceptual and technical criticisms made above. I honestly hope these constructive criticisms will improve the study and, more broadly, pave the way towards sensible implementation of AI techniques to compute DMS climatologies (which will likely become routine in the near future).

Thank you for your constructive comments.
I hope you will find that the paper has been greatly improved based on comments of you and two other anonymous reviewers.

Finally, note that in the specific comments below I will frequently refer to my own papers, simply because some of them are very relevant for the present study and, in some cases, the only ones available. Of course, the authors are free to decide what citations they incorporate. For all these reasons, and for the sake of transparency, I decided to sign the review.
We enjoyed reading your papers.

Specific comments
Abstract
Please reshape taking into account the main criticisms, especially concerning the amount of variance captured when using raw or binned data (general point 3). Raw DMS data variance could be biased towards high-resolution data representative of small-scale variability if no homogenization of the spatial-temporal scales covered by the measurements is applied (the PMEL database consists mostly of coarse resolution data).
We have taken most of your advice, and the manuscript has been thoroughly revised. More tests have been conducted, and our results do not significantly change, which means our method is robust. The abstract has been edited accordingly.

Introduction

L19: OK, but the approach proposed here does not reveal the factors controlling DMS variability. Rephrasing suggested.

We have added more tests to figure out how the environmental changes can influence surface ocean DMS distribution. With this revision, we believe that the sentence here is appropriate.

L24: If the authors insist on the mechanistic point of view (not sure is the right line of thought in this paper), I suggest adding "process rate measurements" here. They hamper predictive models even more strongly than limited observations of DMS concentration.

The term "process rate measurements" has been added.
Meanwhile, we did more experiments to exploit the mechanisms.

L30: To the best of my knowledge, the term "summer paradox" was coined by Simó & Pedrós-Alió 1999 (Nature), so I suggest crediting them for it.

Corresponding reference has been added. Thank you.

L49: Relevant citations here are Le Clainche et al. 2010 (GBC) (S cycling model inter-comparison seeking to understand the processes responsible for the summer paradox) and Tesdal et al. 2016 (Env Chem) (the most extensive comparison among gridded climatologies, empirical and prognostic models published so far, to my knowledge).

Corresponding references have been added. Thank you.

L61: DMS is produced by some marine algae, some bacteria, and mostly as a result of food web interactions (Kiene et al. 2000; Simó et al. 2001, Stefels et al. 2007, Curson et al. 2011, Moran et al. 2012, etc.). Please nuance and refine.

Corresponding references have been added, and now the text is as follows,

"The precursor of DMS, DMSP, is mainly produced by marine algae (e.g. Kiene et al., 2000; Curson et al., 65 2011), and a small fraction of DMSP is transformed to DMS by marine algae and/or bacteria lyases (Simó, 2001; Stefels et al., 2007; Curson et al., 2011; Moran et al., 2012), and mostly as a result of food web interactions (Kiene et al., 2000; Simó, 2001)."

Methods
L73: Over 10,000 measurements came from NAAMES alone. Not binning the data might give too much weight to particular conditions sampled during NAAMES, at least in the multilinear regression.

The valid NAAMES data points are 6,939, which accounts for ~7% of the total data with expanded PMEL database. For the NAAMES data, we matched the DMS observation with super high-resolution satellite products, which ensures most of the data points have unique predictor combinations. We have added more results and discussion to argue why binning the data is not a good choice.

L75: The DMSPt, Chla, SST and SSS data in the PMEL database require some quality control. This was documented by Galí et al. 2015. Quality controlled datasets with stringent satellite match-up criteria (ie minimizing the use of climatological coarse resolution data), as well as a piece of code used to clean data, are publicly available on github: https://github.com/mgali/DMS-SAT_DATA_DEV_VAL.

Thank you for the useful tips, we followed your instruction to clean up the data. Specifically, we removed DMS concentration higher than 100 nM and lower than 0.1 nM, DMSPt concentration lower than 1 nM. We also removed data with salinity lower than 30 psu to focus on open ocean.

L75: Fluorometric Chl is on average around 40% higher than HPLC Chl (Sathyendranath et al., 2009), and the proportion sometimes varies quite a bit. Satellite Chl is validated against HPLC measurements (e.g. Morel et al., 2007).

Yes, Good point.

For Chl a data, we have added the following discussion (l.84 – 1.92).

"SeaWiFS Chl-$a$ data (Level 3-binned, spatial resolution of 9.2 km) from December 1997 to March 2010 were matched to DMS data according to coordinates and sampling date. We compared PMEL in situ Chl $a$ to SeaWiFS Chl $a$, which are well correlated on logarithmic scale ($R^2$ = 0.64) with a slope of 0.67 and an intercept of -0.06, $[\log(Chl_{SeaWiFS}) = 0.67 \log(Chl_{in-situ}) - 0.01]$, which means that on logarithmic scale SeaWiFS Chl-$a$ concentrations are on average ~30% lower than those of in situ Chl-$a$ concentrations. This is possibly because SeaWiFS Chl $a$ is calibrated based on HPLC determined Chl $a$ (Morel et al., 2007), which on average is ~40% lower than that determined using Fluorometric method (Sathyendranath et al., 2009). Unfortunately, there is no flag in the database showing how Chl $a$ was determined. For consistency, we use only Chl-$a$ data retrieved from SeaWiFS in the following multilinear and network models."

L77: Less than 3.5 years do not make a good Chl climatology in many ocean areas in my experience. Data products covering much longer periods are available on NASA's ocean colour website. Please update datasets, and specify also what reprocessing was used.

Good point. We updated our climatology so that the current climatology used is from 1997-2010. The new climatology is Level 3-binned (L3BIN with spatial resolution of 9.2 km) from SeaWiFS.

The change of Chl a climatology does not significantly change our results.

L79: The more recent climatology of Holte et al. 2017 seems to outperform that of Schmidtko et al. 2013 in areas of deep winter convection (subpolar North Atlantic) or where deep mixing prevails (Southern Ocean circumpolar current). In some cases the differences are important. Please consider using the Holte et al. 2017 MLD climatology.

Good point. We updated our MLD climatology and used MIMMOC one in the revised model.

The change of MLD climatology does not significantly change our results.

L81: Please specify the nutrient datasets used.

Good point. We added descriptions about the nutrient data sources (l.98 – l.101).

L105-107: I see some contradiction here. Data extremes typically arising from nonlinear dynamics are often smoothed out when averaging data. Your predicted variable (DMS) retains full variability but predictor fields do not, because apart from SST they largely originate from monthly climatologies. How can meaningful nonlinear relationships be identified?

Good point, but it is partially true. Except for SST that has the most in situ observational data (81069 for PMEL data), in situ salinity was reported for ~74% of the DMS data. More importantly, every data point has their unique time-space signatures (5 parameters in the

model).  We used high resolution Chl *a* (0.418), PAR (0.418), and MLD (0.5×0.5) climatologies for the PMEL data. For the NAMMS data, the Chl *a*, SST, and PAR have even higher resolution (0.0417×0.0417). The high-resolution data and unique time-space ensure that each data has a unique signature. See also our response to your general question No. 3.

L128-130: Are these parameters default ones, or tuned manually to achieve reasonable fits in this particular study?

These parameters are called hyper-parameters in machine learning language. We manually tuned the parameters to prevent over-/under- fitting the data (l.159 – l.163).

L137: Inclusion of time of day is interesting, although diel variability was not been mentioned earlier in the manuscript. Are hourly predictions useful for computing climatologies? Although DMS can oscillate on diel time scales (Galí et al., 2013; Royer et al., 2016) diel cycles do not seem to follow a fixed pattern, at least in low-latitude high-resolution datasets (e.g. Royer et al., 2015).
Good point, we retested the diurnal parameters (two time parameters). Adding them slightly worsen the performance of the model (Fig. 2a in the MS). We have added the following discussion (l.265 – l.274),

"Given the strong correlation between solar radiation and DMS concentration reported by Vallina and Simó (2007), one would expect that adding sampling time would improve the model performance. However, it increases RMSE slightly (Fig. 2a). Galí et al. (2013c) studied diel cycle at the Mediterranean Sea and Sargasso Sea. Among their four experiments (three in the Mediterranean Sea and one in the Sargasso Sea) regular diel variation was observed at only one experiment in the Mediterranean Sea at summer season, with highest DMS values observed at midnight and lowest values at midday. In all the other experiments, diel variations for both DMS and DMSPt pools were small. Gross community DMS production during the daytime was two to three times higher than that in the nighttime, but the high DMS production was compensated by greater photochemical and microbial consumption (Galí et al., 2013c). The balance between DMS production and consumption appears to dampens DMS diel variation. This may explain why adding time parameters does not improve the ANN model's predictive ability."

Results
L170 and 177: Note Galí et al. (2018) reported an R2 of 0.42 (r of 0.65) between DMS and DMSPt using the same datasets with stricter quality control. Similarly for DMS vs. Chl: R2 of 0.20 (r of 0.45).

We followed your instructions to clean up the data. Our new $R^2$ value for DMS and in situ Chl *a* is 0.21 (n = 10,404 compared to 8141 in Gali et al. (2018)), for DMS and DMSPt is 0.41 (n = 4060, compared to 3637 in Gali et al. (2018)). Both pairs have larger data than those by Gali et al. (2018) for two reasons, 1) the expansion of the online database; 2) no averaging being done.

L172-176: These sentences look a bit contradictory and may need further elaboration (or else, can be removed). Is it straightforward or not to predict DMS from DMSPt? Are measurements sufficient or not? Regarding DMSP prediction, another relevant study is that by McParland & Levine (2018). Regarding the relationship between DMSPt and DMS in the global PMEL database, Galí et al. (2018) is a relevant reference.

We have removed the corresponding sentence. Meanwhile, we have added more discussion as follows (l.226 – l.229).

"McParland and Levine (2019) developed a mechanistic model that related intracellular DMSP concentration to environmental stress, and coupled the model with MIT ecosystem model (DARWIN) to estimate global ocean DMSP distribution. Galí et al. (2015) first applied a remote sensing algorithm to obtain a DMSP climatology, from which they predict DMS climatology through an empirical relationship with PAR (Galí et al., 2018)."

L179: The weaker relationship between DMS and Chl in the entire dataset probably results from the higher proportion of oligotrophic low latitude data (where DMS is anticorrelated to Chl over the seasonal cycle) compared to in situ Chl-DMS data pairs. The difference between in situ fluorometric Chl and satellite Chl may also play a role. Finally, note that the global PMEL DMS database is biased towards productive conditions (Galí et al. 2018; figure 7) which influences global DMS-Chl correlations. In summary, the correlation between DMS and Chl in global datasets is not really meaningful as it depends strongly on how evenly represented are the different ocean biomes.
Thank you for pointing this out.
We made the following corrections (l.314 – l.317).

"On the other hand, there are numerous studies that observed no correlation between DMS and Chl a (e.g. Dacey et al., 1998; Kettle et al., 1999; Toole and Siegel, 2004). The inconsistent relationships indicate the complexity of the biogeochemical reduced sulfur cycle. As suggested by Simó (2001), not only can phytoplankton biomass, taxonomy, and activity influence DMS production, but so does food-web structure and dynamics. The inconsistent relationship may also explain the low ranking of Chl a in the models."

L187-190: This is incorrect. Dilution is not the main explanation for the negative relationship between MLD and DMS (as originally proposed by Aranami and Tsunogai, 2004). The main explanation is the different balance between biological DMS sources and sinks, as explained by Galí & Simó (2015). In the handful of studies that have made DMS budgets including the vertical mixing term, vertical DMS transport has never been found to dominate DMS budgets in the MLD over relevant (~daily) time scales. Check for example Bailey et al. 2008 (DSR), Herrmann et al. 2012 (CSR), Galí et al. 2013 (GBC), Royer et al. 2016 (Sci Rep), etc. DMS turnover in the surface layer due to vertical transport is generally an order of magnitude slower than biological turnover or biological + photochemical turnover, at least. Please correct.
Thank you for pointing this out.
We have clarified the reasoning in the text and also as follows (l.289 – l.296).

"MLD is another important predictor. High DMS concentrations in the open ocean have been detected when the water column is most stratified (Simó and Pedrós-Alió, 1999). The authors proposed that a stratified (high light) environment nourishes strong DMSP producers, or that phytoplankton cellular DMSP quota increases in such an environment. High conversion rates from DMSP to DMS in stratified waters is another reason for high DMS concentrations when MLD is shallow. Meanwhile, the biological DMS consumption rate decreases in oligotrophic oceans (Galí and Simó, 2015). A dilution model was also proposed to explain the anti-correlation between DMS concentration and MLD (Aranami and Tsunogai, 2004). The authors proposed that mixed layer deepening entrains water with little or no DMS into surface waters and dilutes surface DMS concentrations, but recent studies have shown that DMS loss rate via vertical mixing is orders of magnitude lower than production/consumption rates (e.g. Galí et al., 2013c; Royer et al., 2016)."

L191-199: The strongest evidence for light-driven DMS production in natural plankton assemblages comes from recent work by myself and colleagues (Galí et al. 2013a, b, c; Royer et al. 2016). Evidence for light-driven DMS production in Toole et al. (2006) (otherwise, a great piece of work!) is indirect as that study focused on DMS removal processes.
We have updated the reference. Thank you.

Section 3.2: see general comment 3 on data binning.
Please refer to our response to your comments #2.

L201-204: It is unclear here if the authors made the appropriate comparisons with Simó & Dachs 2002 and Vallina & Simó 2007 empirical models. Was DMS compared with surface PAR or with the solar radiation dose in the mixed layer as done by Vallina and Simó 2007? Similarly, did the authors correctly apply the double algorithm used by Simó and Dachs 2002, where different equations are used depending on the value of Chl/MLD? Or just computed a single regression of DMS against Chl/MLD?
Yes, we used exactly the same model, and have made this clear in the text as follows (l.319 – l.329),

"Simó and Dachs (2002) obtained high $R^2$ values between DMS concentration and the ratio of Chl a to MLD (Chl/MLD) when Chl/MLD is greater than or equal to 0.02, and between DMS concentration and ln(MLD) when Chl/MLD is less than 0.02. We tried exactly the same model on raw PMEL data with in situ Chl-a measurements and climatological MLD, and found that both correlations between DMS and Chl/MLD (n = 4,921, $R^2$ =~ 0 .1) and between DMS and ln(MLD) (n = 5,978, $R^2$ =~ 0 ) are statistically insignificant. To reduce interannual variability, we binned in situ Chl a and DMS into monthly 1° × 1° grid, and retested the above model. We found that the correlations are still statistically insignificant. ($R^2$=~ 0)

Vallina and Simó (2007) reported an R2 of 0.95 (n=14) between DMS concentration and SRD. We applied the same linear regressions on both raw data and monthly 1° × 1° data, and found

no significant correlations between DMS and SRD as calculated according to Vallina and Simó (2007):,

$$SRD = SI \cdot \frac{1}{Kd490 \cdot MLD} (1 - \exp(-Kd490 \cdot MLD),$$

where SI is solar insolation (W m-2); Monthly SI data are from a CESM simulation (Wang et al., 2019)."

Section 3.3: Methods section mentions 8 initial variables (PAR, MLD, Chl a, SSS, SST, DIN, DIP, and SiO), but, what predictor variables were included in the final multilinear regression model (MLR)? Does the R2 of the MLR refer to linear of log space?

The Choice of parameter combinations are based on BIC criterion.
Yes, $R^2$ value is for natural log space. We have made this clear in the revised paper.

Section 3.4: Since this subsection describes the main technical innovation of this paper, a deeper explanation of why the ANN gives these results would be very welcome. See general comment 4.

Yes, we have extensively discussed the ANN model on pages 9.

Section 3.5: For the authors' information, global DMS fields produced with the remote sensing algorithm of Galí et al. 2018, as well as the algorithms of Simó & Dachs 2002 and Vallina & Simó 2007, are available in this repository: https://doi.org/10.5281/zenodo.2558511.  Corresponding Matlab and R codes are available on a linked github repository: https://github.com/mgali/DMS-SAT_ALGORITHM.

Good resources. We have plotted the data together with our new prediction (Fig.3 and Fig. 4).

L235: These references are not appropriate here. Please cite studies that actually documented DMS(P) dynamics in subpolar or polar blooms of coccolithophores or Phaeocystis.

Good point, we updated the discussion as follows (l.371 – l.376),

"The summertime high DMS concentration at high latitudes is believed to be linked to the release of ice algae that are prolific DMSP producers (Stefels et al., 2012; Webb et al., 2019). As an important cryoprotectant and osmolyte, DMSP helps ice algae to cope with the low temperature and high salinity conditions (Thomas and Dieckmann, 2002). High DMS concentrations at high latitudes have also been observed to accompany blooms of coccolithophores and Phaeocystis, which are strong DMSP producers (Neukermans et al., 2018; Wang et al., 2015). The shoaling of mixed layer depth in summer provides favorable conditions, i.e. stable and warm, with adequate irradiation for coccolithophores and Phaeocystis growth (Galí et al., 2019)"

L247-248: I do not find reasonable that DMS decreases below 0.1 nM in a subtropical gyre in winter. By examining the maps in Fig. 3 I would say ANN DMS is mostly between 0.1 and 0.5 nM, which still looks a bit low but more realistic according to my experience. DMS concentrations lower than 0.1 nM are extremely rare in both the PMEL database and in global estimates made with empirical algorithms (see Galí et al. 2018 figure 7).

In the revised MS, we rerun the model, and now there is not region with DMS concentration lower than 0.1 nM.

L250-254: Please check Galí and Simó 2015 (GBC) for a mechanistic explanation of the summer paradox.
We have changed the wording as follows (l.389 – l.393),

"Fig. 6 compares monthly mean Chl-a concentrations to DMS concentrations in N. and S. hemisphere gyres. The concentrations are normalized to the range of 0 to 1. It is clear that Chl a and DMS are anti-correlated, DMS concentration peaks at summer season when Ch-a concentration is generally low. This phenomenon is previously termed as "summer DMS paradox" (Simó and Pedrós-Alió, 1999). This pattern is more apparent in the S. hemisphere gyres, because the terrestrial influence is smaller in the S. hemisphere than in the N. hemisphere."
In general, I suggest making a figure showing the climatological seasonal cycles in different ocean biomes or regions, to better support the description made in the text.

Good point.
We plotted monthly mean Chl-$a$ and DMS concentrations for N. and S. hemisphere gyres as shown in the following figures. It is clear that DMS and Chl $a$ are anti-correlated in the gyres. DMS peaks in the summer when Chl $a$ is at annual minimum.

[Figure]

Figure 8. Distributions of monthly mean DMS and Chl-*a* concentrations for N. and S. hemisphere gyres. The gyres are defined as regions between 30∘ and equator where annually mean DIP concentration is below 0.2 μM. Monthly mean concentrations are normalized to the range of 0 to 1.

L273-293: Here I strongly suggest citing Tesdal et al. 2016.
Reference has been cited, and more discussion has been added.

Figures
Figure 3 and 5: I suggest using a color scale with different colors to help readers appreciate concentration patterns.

Thank you for this suggestion, we have updated the colormaps for both figures.

Figure 6: I strongly recommend splitting results into northern and southern hemisphere given the strong seasonality of DMS (also wind speed and SST), which results in opposed seasonal patterns.

Yes, another good suggestion, we updated the figure accordingly.

Minor corrections
L131: What does "epochs" mean in this context? Please use synonym for readers that are not expert in ANN or similar techniques.

one **epoch** = one forward pass and one backward pass of *all* the training examples

In terms of artificial neural networks, an epoch refers to one cycle through the full training dataset. Usually, training a neural network takes more than a few epochs. In other words, if we feed a neural network the training data for more than one epoch in different patterns, we hope for a better generalization when given a new "unseen" input (test data).

We have added more explaination in the revised MS (l.167 – l.169),
An epoch consists of one forward pass and one backward pass of all the training examples.

L218: The "tracer-tracer" term is not needed here (quite specific to bgc modelling).

Thank you. We have removed the term.

Reviewer references (only if not cited by the authors)
Aranami, K., & Tsunogai, S. (2004). Seasonal and regional comparison of oceanic and atmospheric dimethylsulfide in the northern North Pacific: Dilution effects on its concentration during winter. Journal of Geophysical Research: Atmospheres,109(D12).
Bailey, K. E., Toole, D. A., Blomquist, B., Najjar, R. G., Huebert, B., Kieber, D. J., ... & Del Valle, D. A. (2008). Dimethylsulfide production in Sargasso Sea eddies. Deep Sea Research Part II: Topical Studies in Oceanography,55(10-13), 1491-1504.
Curson, A. R., Todd, J. D., Sullivan, M. J., & Johnston, A. W. (2011). Catabolism of dimethylsulphoniopropionate: microorganisms, enzymes and genes. Nature Reviews Microbiology, 9(12), 849-859.
Galí, M., Ruiz-González, C., Lefort, T., Gasol, J. M., Cardelús, C., Romera-Castillo, C., & Simó, R. (2013). Spectral irradiance dependence of sunlight effects on plankton dimethylsulfide production. Limnology and oceanography, 58(2), 489-504.

Galí, M., Simó, R., Vila-Costa, M., Ruiz-González, C., Gasol, J. M., & Matrai, P. (2013). Diel patterns of oceanic dimethylsulfide (DMS) cycling: Microbial and physical drivers. Global Biogeochemical Cycles, 27(3), 620-636.

Galí, M., Simó, R., Pérez, G., Ruiz Gonzalez, C., Sarmento, H., Royer, S. J., ... & Gasol, J. M. (2013). Differential response of planktonic primary, bacterial, and dimethylsulfide production rates to static vs. dynamic light exposure in upper mixed-layer summer sea waters.

Galí, M., & Simó, R. (2015). A meta-analysis of oceanic DMS and DMSP cycling processes: Disentangling the summer paradox. Global Biogeochemical Cycles,29(4), 496-515.

Herrmann, M., Najjar, R. G., Neeley, A. R., Vila-Costa, M., Dacey, J. W., DiTullio, G. R., ... & Vernet, M. (2012). Diagnostic modeling of dimethylsulfide production in coastal water west of the Antarctic Peninsula. Continental Shelf Research,32, 96-109.

Holte, J., Talley, L. D., Gilson, J., & Roemmich, D. (2017). An Argo mixed layer climatology and database. Geophysical Research Letters, 44(11), 5618-5626.

Kiene, R. P., Linn, L. J., & Bruton, J. A. (2000). New and important roles for DMSP in marine microbial communities. Journal of Sea Research, 43(3-4), 209-224.

Le Clainche, Y., Vézina, A., Levasseur, M., Cropp, R. A., Gunson, J. R., Vallina, S. M., ... & Bopp, L. (2010). A first appraisal of prognostic ocean DMS models and prospects for their use in climate models. Global biogeochemical cycles, 24(3).

Moran, M. A., Reisch, C. R., Kiene, R. P., & Whitman, W. B. (2012). Genomic insights into bacterial DMSP transformations. Annual review of marine science, 4, 523-542.

Morel, A., Huot, Y., Gentili, B., Werdell, P. J., Hooker, S. B., & Franz, B. A. (2007). Examining the consistency of products derived from various ocean color sensors in open ocean (Case 1) waters in the perspective of a multi-sensor approach. Remote Sensing of Environment,111(1), 69-88.

Royer, S. J., Mahajan, A. S., Galí, M., Saltzman, E., & Simó, R. (2015). Small-scale variability patterns of DMS and phytoplankton in surface waters of the tropical and subtropical Atlantic, Indian, and Pacific Oceans. Geophysical Research Letters, 42(2), 475-483.

Royer, S. J., Galí, M., Mahajan, A. S., Ross, O. N., Pérez, G. L., Saltzman, E. S., & Simó, R. (2016). A high-resolution time-depth view of dimethylsulphide cycling in the surface sea. Scientific reports,6, 32325.

Sathyendranath, S., Stuart, V., Nair, A., Oka, K., Nakane, T., Bouman, H., ... & Platt, T. (2009). Carbon-to-chlorophyll ratio and growth rate of phytoplankton in the sea. Marine Ecology Progress Series,383,73-84.

Simó, R., & Pedrós-Alió, C. (1999). Role of vertical mixing in controlling the oceanic production of dimethyl sulphide. Nature,402(6760), 396-399.

Tesdal, J. E., Christian, J. R., Monahan, A. H., & von Salzen, K. (2016). Evaluation of diverse approaches for estimating sea-surface DMS concentration and air–sea exchange at global scale. Environmental Chemistry,13(2), 390-412.

---

## Author Comment (AC2) · 7 Jun 2020

The manuscript proposes a new global ocean DMS climatology, or a method to con- struct it, based on an Artificial Neural Network (ANN). This methodology uses a number of variables and their intelligent combinations as predictors of DMS concentration distribution. It is meant to overcome the limitations of objective analysis based on inter- and extrapolations as well as the limitations of simple linear or logarithmic regressions with few predictors, and to provide better fits of predictions to observations. While developing their ANN application and to claim its better performance, the authors conduct parallel applications of previously published models. Eventually, they indeed obtain a better fit, but very similar seasonal and geographic distributions. The global annual emission to the atmosphere is revised towards the lower end of the hitherto most accepted estimate.

The topic is timely since, after years of having DMS been dismissed for its role in new particle formation, recent studies are recognizing it again as a central agent in ocean- atmosphere- climate interactions. Atmospheric chemistry and climate models require updated climatologies of DMS emissions.

Thank you for your positive comments.

The text is generally well written and the display items are clear and informative, with one exception (see particulars below).

That said, the manuscript reads as though it was written 10 years ago. Even though the ANN methodology is probably state-of-the-art (I am not an expert and can hardly assess every technical aspect), the interpretation arguments are outdated, ignoring many of the discoveries in the last decade. This adds to some bad referencing. But most importantly, when the authors intend to make relevant comparisons with previous similar efforts, they miss the point of the studies they are comparing to, or use them in the wrong way. Finally, besides presenting their new method, they fail to discuss what is new in their findings, they just repeat what is already well known and with much poorer arguments, rather than stressing what is unveiled and why. I will develop these and other concerns hereafter, as they come up in the order of the manuscript.

L28-30: "The weak relationship may be caused by the so-called "summer DMS para- dox", which describes a phenomenon where a maximum DMS concentration is commonly detected in low latitude waters when phytoplankton biomass is low (Toole and Siegel, 2004; Vallina et al., 2008)." This is not the summer DMS paradox (a term, by the way, suggested by Simo & Pedros-Alio Nature 1999), which states that the annual maximum of surface DMS commonly occurs in summer, even at the mid and subtropical latitudes where chlorophyll-a (chl-a) is at its annual minimum.

Thank you for your correction.
We have rephrased the statement and added corresponding reference as follows,

"The weak relationship may be caused by the so-called "summer DMS paradox", which describes a phenomenon that annual maximum of surface DMS concentration is commonly

detected in summer when Chl *a* is at its annual minimum in mid and subtropical low latitude waters (Simó and Pedrós-Alió, 1999)."

L34-35: "Simó and Dachs (2002) achieved a strong relationship between heavily binned and averaged DMS data and mixed layer depth (MLD)." This is not true. Simo & Dachs (2002) correlated DMS to the MLD and to chl-a/MLD, depending on a chl-a/MLD threshold.

We have corrected the corresponding statement as follows:

"Simó and Dachs (2002) achieved a strong linear relationship between heavily binned/averaged DMS and mixed layer depth (MLD) when Chl-a/MLD ≥ 0.02, and a logarithmic relationship between DMS and Chl-a/MLD when Chl-a/MLD < 0.02."

L53-54: "Many provinces lacked adequate data to create a reliable climatology (Fig. A1). In those situations, temporal interpolations were used to fill the blanks, and to create a first-guess map." This was done where monthly data gaps existed to complete the seasonality. Where data were lacking to even outline a seasonality, this was taken from a neighboring province and adjusted to the existing data.

Thank you for pointing this out. We have adjusted the description as follows,

"Many provinces lacked adequate data to create a reliable climatology (Fig. A1). In those situations, they first generated an annual cycle with monthly means for each province. Temporal interpolations were used to fill the monthly gaps if there were enough data to create a robust annual mean. Otherwise, interpolation from neighboring provinces was used to fill the remaining gaps."

L61: "Since DMS is produced by marine that algae. . ." This is totally outdated. There are tens of papers showing that this is an oversimplification. DMSP is mainly produced by marine algae, and it is transformed into DMS by marine algae, bacteria and with involvement of zooplankton.

We have rephrased the statement as follows,

"The precursor of DMS, DMSP, is mainly produced by marine algae (e.g. Kiene et al., 2000; Curson et al., 2011), and a small fraction of DMSP is transformed to DMS by marine algae and/or bacteria lyases (Simó, 2001; Stefels et al., 2007; Curson et al., 2011; Moran et al., 2012), and mostly as a result of food web interactions (Kiene et al., 2000; Simó, 2001)."

L93-94: "We do not log-transform SST to avoid losing data with temperature below (equal to) zero." You may have other reasons to not log transform SST, but not this one. A common practice to log transform SST if desired is to convert it to K (Kelvin) first.

We have re-run the model using log-transformed K.

If I understand it correctly, you use chl-a data where available, otherwise you take it from SeaWiFS. What efforts have you done to reconcile in situ with satellite chl- a? It is well known

that algorithms for satellite estimates of chl-a are developed and calibrated against HPLC chl-a, and there is an important shift between this and Turner fluorometric chl-a. Therefore, putting together in situ (Turner, perhaps HPLC too?) and satellite chl-a data will mess up your statistics.

Thank you for pointing this out. In the revised MS, we used only satellite Chl-*a*, and added more discussion as follows (l.84 – 1.92),

"SeaWiFS Chl-*a* data (Level 3-binned, spatial resolution of 9.2 km) from December 1997 to March 2010 were matched to DMS data according to coordinates and sampling date. We compared PMEL in situ Chl *a* to SeaWiFS Chl *a*, which are well correlated on logarithmic scale ($R^2$ = 0.64) with a slope of 0.67 and an intercept of -0.06, $[\log(Chl_{SeaWiFS}) = 0.67 \log(Chl_{in-situ}) - 0.01]$, which means that on logarithmic scale SeaWiFS Chl-*a* concentrations are on average ~30% lower than those of in situ Chl-*a* concentrations. This is possibly because SeaWiFS Chl *a* is calibrated based on HPLC determined Chl *a* (Morel et al., 2007), which on average is ~40% lower than that determined using Fluorometric method (Sathyendranath et al., 2009). Unfortunately, there is no flag in the database showing how Chl *a* was determined. For consistency, we use only Chl-*a* data retrieved from SeaWiFS in the following multilinear and network models."

Calculation of air-sea fluxes: I agree that Nightingale 2000 is quite a standard. But, why not using a more updated linear relationship of Kw to u10? Marandino proposed one with one of the coauthors. Also, you use monthly means of wind speed. Since you are using a nonlinear dependence of Kw on the u10, how do you deal with the fact that a mean u10 will not give the same result as a mean Kw?

In the paper we used two flux parameterizations, GM12 according to Goddijn-Murphy et al .2012 and N00 according to Nightingale et al 2000. GM12 DOES describe a linear relationship between Kw and u10, and it is more updated that Marandino et al 2008 (if this is the reference that you referred to).

We also used N00, because we can compare to previous results that used the same parameterization.

We did a correction on mean u10 as described below (l.209 – l.214),

"Because the N00 parameterization was calibrated using in situ wind speeds and has a nonlinear quadratic dependence on wind speed, the use of monthly mean wind speeds will introduce errors. To reconcile differences between in situ wind speed and monthly mean wind speed, a correction is applied according to Simó and Dachs (2002) by assuming that instantaneous wind speeds follow a Rayleigh distribution. Eq. 8 thus becomes $k_{w,660}$ =$[0.222\eta^2\Gamma(1+2/\xi)+0.333\eta\Gamma(s)](Sc_{DMS}/600)^{-0.5}$, where $\eta^2$ =$4U_{10}^2/\pi$; s=(1+1/ξ), and ξ = 2 for Rayleigh distribution (Livingstone and Imboden, 1993)."

L170-176: It reads as though you did not know of the existence of Gali et al. BGS 2016.

We have added the following discussion (l.226 – l.229),

"McParland and Levine (2019) developed a mechanistic model that related intracellular DMSP concentration to environmental stress, and coupled the model with MIT ecosystem model (DARWIN) to estimate global ocean DMSP distribution. Galí et al. (2015) first applied a remote sensing algorithm to obtain a DMSP climatology, from which they predict DMS climatology through an empirical relationship with PAR (Galí et al., 2018)."

L182: "On the other hand, negative correlations between DMS and Chl a have also been detected in coastal waters of the Mediterranean and in the Sargasso Sea (Toole and Siegel, 2004)." Toole & Siegel did not do anything with Med Sea data. The original data from the Sargasso Sea were from Dacey et al DSR 1996, and data from the coastal Med Sea were reported by Vila-Costa et al. LO 2008.

Thank you for pointing this out.
We realized that it is that Dacey et al. (1998) reported the original Sargasso Sea data, and Toole and Siegel analyzed the correlation between DMS and Chl a.
We have added the original reference and changed the wording.

"On the other hand, there are numerous studies that failed to correlate DMS and Chl a (e.g. Dacey et al., 1998; Kettle et al., 1999; Toole and Siegel, 2004)."

L185-190: This is a very poor interpretation of the DMS vs MLD coupling, and a misuse of the original relationship suggested by Simo & Dachs GBC 2002. As a matter of fact, you cite Simo & Pedros-Alio GBC 1999 because they brought it up for the first time, but the occurrence of a negative relationship between DMS and MLD over large regions of the global ocean was reported by Simo & Dachs. However, the relationship was logarithmic, DMS = a*Ln(MLD) + b, and there are reasons for this to occur, related to exposure to solar radiation. Trying to correlate DMS directly to MLD (or in a log-log manner) is not expected to provide good prediction.

Thank you for pointing this out. We have added more discussion as follows,

"It is proposed that stratified (high light) environment nourishes strong DMSP producers, or phytoplankton cellular DMSP quota increases in such an environment. High conversion rate from DMSP to DMS in stratified waters is another reason for high DMS concentrations when MLD is shallow. Meanwhile, the biological DMS consumption rate decreases in oligotrophic oceans (Galí and Simó, 2015). Dilution model, which describes a phenomenon that when the mixed layer deepens water with no or little DMS is entrained into the surface waters and dilutes surface DMS concentrations, was also proposed to explain the anti-correlation between DMS concentration and MLD (Aranami and Tsunogai, 2004). However, recent studies show that DMS loss rate via vertical mixing is orders of magnitude lower than production/consumption rates (e.g. Galí et al., 2013c; Royer et al., 2016)."

L189-199: There are a number of papers that should be cited here – besides Toole et al. and Sunda et al, several papers by Marti Gali deal exactly with the effects of solar radiation, and particularly UV, on enhancing DMS production and concentration.

"Climatological PAR is the second strongest predictor (R2 = 0.12, n = 54,683) of raw DMS data with a positive correlation. (. . .) Strong correlation between monthly binned and averaged solar radiation dose (SRD) and DMS concentration has been reported (R2 = 0.94) at the Blanes Bay Microbial Observatory located in the coast of northwest Mediterranean (Vallina and Simó, 2007)." Again, you compare your statistics with that of a previous study, but applying a different calculation. According to the methods description, you used monthly PAR, i.e., monthly surface irradiance. Vallina & Simo 2007, conversely, computed what they called the solar radiation dose, which is the daily averaged solar radiation integral in the mixed layer. This is very different from surface irradiance, because it takes into account the mixed layer depth (and a median light attenuation coefficient). Later on, in L200-211, you infer that, contrasting to Vallina & Simo, you did not get a good correlation to light, and attribute it to the number of original data and to data binning. But you did not use the same light metrics as the other authors, and ignored the arguments given by V&S to use the SRD instead of the surface irradiance, and ignoring the Gali & Simo GBC 2015 meta-analysis too.

Good points.
We have rebuilt the SRD vs DMS model, recomputed the results, and revised our discussion as follows,

"Vallina and Simó (2007) reported an $R^2$ of 0.95 (n=14) between DMS concentration and SRD. We applied the same linear regressions on both raw data and monthly 1°× 1° data, and found no significant correlations between DMS and SRD that is calculated according to Vallina and Simó (2007) using the following equation,

$$SRD = SI \cdot \frac{1}{Kd490 \cdot MLD}(1 - \exp(-Kd490 \cdot MLD)$$

where SI is shortwave irradiance (W m$^{-2}$), which is converted from PAR according to Galí and Simó (2015)."
L201: "Simó and Dachs (2002) obtained a high R2 value between DMS concentration and the ratio of Chl a and MLD (Chl/MLD)." This is not true. As already mentioned above, the Simo & Dachs (2002) model correlated DMS to the MLD (logarithmic) and to chl-a/MLD (linear), depending on a chl-a/MLD threshold.
Yes, we used exactly the same model, and have made it clear in the text as follows (l.319 – l.329),

"Simó and Dachs (2002) obtained high $R^2$ values between DMS concentration and the ratio of Chl a to MLD (Chl/MLD) when Chl/MLD is greater than or equal to 0.02, and between DMS concentration and ln(MLD) when Chl/MLD is less than 0.02. We tried exactly the same model on raw PMEL data with in situ Chl-a measurements and climatological MLD, and found that both correlations between DMS and Chl/MLD (n = 4,921, $R^2$ =∼ 0 .1) and between DMS and ln(MLD) (n = 5,978, $R^2$ =∼ 0 ) are statistically insignificant. To reduce interannual variability, we binned in situ Chl a and DMS into monthly 1° × 1° grid, and retested the above model. We found that the correlations are still statistically insignificant. ($R^2$=∼ 0)"
."

All in all, if you are to compare your statistics to those of S&D 2002 and V&S 2007, everything here has to be recomputed and rewritten.

The results were recomputed and the text was rewritten. Please see our responses to your above comments.

The arguments against binning the data are poor. It is true that binning reduces the variance, but you are using monthly climatologies (heavily averaged and also binned) to relate raw DMS data to potential predictors. Also, binning must be used if you want to avoid giving too much predictive weight to the regions thoroughly sampled over the undersampled. This is becoming more important as we are bringing in underway data at unprecedented spatial resolution, like the NAAMES data incorporated here.

Since this issue was brought up by all three reviewers, we have added adequate discussion/arguments to this point as shown in the following text.

The PMEL database expanded dramatically. Now there are a total of 86,785 valid DMS measurements (concentration greater than 0.1 nM and less than 100 nM according to your instructions), that is 71% larger than the number of data we initially used (51,161). For the expanded data set, ~93% of DMS are accompanied with in-situ SST measurements, ~81% are accompanied with in-situ salinity measurements. More importantly, each data point has their unique location and sampling time signatures. As shown in the following figure, sampling time (date) and location information is a strong DMS predictor, which together can decrease DMS root mean square error to 0.64 (on natural logarithm scale). Adding other climatological predictors can further improve the model performance.

The NAAMES dataset has 6,786 valid data points, which are ~7% of the total data points (93,571 = 86785+6786). All data are accompanied with in-situ Chl a, SST, and SAL measurements. For parameters without in-situ measurements, high resolution data are used to match DMS measurements, $0.0417° \times 0.0417°$ for PAR, $0.5° \times 0.5°$ for MLD, and $1° \times 1°$ for NO3, which ensures most of DMS have a set of unique predictors. As shown in Table 1, merging NAAMES data with PMEL data does not significantly change the statistic.

Moreover, binning the data will reduce data variance, which has been demonstrated by Derevianko et al. (2009). The objective of this study is to train an ANN with as many data as possible, so that the model is generalized. It not only can apply to coarse resolution predictor fields, but also can apply to very fine resolution field, for example, we have applied the network to fine resolution NAAMES fields for comparison with in-situ DMS measurements (Bell et al., in prep.).

Lastly, binning data will also result in loss of information. A great amount of information is associated with sampling time and date as shown in the following figure (Fig. 2a in MS). By binning the data into monthly $1° \times 1°$ grid, the valid DMS data points will decrease significantly from 82,996 to 9,018; sampling date feature (365) will be average to 12 months, and coordination combinations will be averaged from $87,332 \times 87,332$ to $180° \times 360°$, which

represents great information reduction. For ANN models, less data points usually lead to overfitting Fig. 2b.

[Figure]

Fig. 2 Parameter sensitivity tests on raw and binned data. (a) Root mean square error on logarithmic scale for the model trained using raw data; (b) Root mean square error on logarithmic scale for the model trained using binned data. The time and location parameters are tested separately without combining with environmental parameters as shown in the upper panel, (I) with only location parameters; (II) with location and day of year parameters; and (III) with location, day of year, and time of day parameters. The model with three location parameters (I) has a root mean square error on natural logarithmic scale of ~0.83, which decreases to ~0.65 by adding sampling day of year parameters (II), however, increases to ~0.67 by adding sampling time parameters (III). We, therefore, do not include sampling time parameters in the following tests. We tested every possible combination of the eight parameters (PAR, MLD, SST, SAL, Chl a, DIP, DIN, and SiO), which in total are 255 tests.

L231-233: "The summertime high DMS concentration at high latitudes is consistent with the hypothesis that phytoplankton use DMSP as a cryoprotectant (Karsten et al., 1992). It is found that the same phytoplankton (Antarctic macroalga) contains higher DMSP concentration in the polar regions than in the temperate regions (Karsten et al., 1990)." Poor again, if not wrong. See recent papers on DMS in polar regions (e.g. Webb et al Sci Rep 2018, Gali et al. PNAS 2019). And macroalgae are not phytoplankton.

Good points. We have updated the reference, and added more discussion as follows,

Good point. We have updated the reference, and added more discussion as follows (l.371 – l.376),

"The summertime high DMS concentration at high latitudes is believed to be linked to the release of ice algae that are prolific DMSP producers (Stefels et al., 2012; Webb et al., 2019). As an important cryoprotectant and osmolyte, DMSP helps ice algae to cope with the low temperature and high salinity conditions (Thomas and Dieckmann, 2002). High DMS concentrations at high latitudes have also been observed to accompany blooms of coccolithophores and Phaeocystis, which are strong DMSP producers (Neukermans et al., 2018; Wang et al., 2015). The shoaling of mixed layer depth in summer provides favorable conditions, i.e. stable and warm, with adequate irradiation for coccolithophores and Phaeocystis growth (Galí et al., 2019)"

Subsequent discussion: The seasonality and geographic distribution of DMS have been profusely (and much better) discussed by Lana et al. GBC 2011 and others, including regional studies. You should rather focus on new features unveiled with respect to others, particularly Lana 2011.

In the revised MS, we have shortened the spatial distribution discussion, and added more discussion on the comparison with previous results.

L305-306: "By contrast, objective interpolation methods are spatial/temporal averages of sparse data with no underlying basis in environmental variability." Again, this is not totally true. In Lana et al. 2011, to create a first guess field, biogeographic provinces were used, which is an informed approach to extrapolation. These provinces are defined from environmental descriptors. And a distance weighted interpolation from original data was used for interpolation.

We agree that the provinces are defined from environmental descriptors, however, we also noticed that the provinces are static with no seasonality. In the revised version, we weakened our expression as follows,

"By contrast, objective interpolation methods are spatial/temporal averages of sparse data with weak underlying basis in environmental variability."

Figure 2: An annual average is not very informative. I would even argue it is misleading in the case of highly seasonal variables like DMS, because summer maxs and winter mins cancel out

each other. I would recommend splitting the map into two or four seasons to show hemispherical patterns.

Good point. We split the global map into Southern and Northern hemispheres, and plotted seasonal cycles for each hemisphere (Fig. 3). Also, we made zonally mean average for each season (Fig. 4).

Figure 4: Some differences are outstanding but you do not discuss them. For instance, Lana 2011 captures the September max of DMS concentration in the subarctic NE Pacific, because it is well covered with data. Conversely, your ANN does not capture it. This warrants some discussion, as it will reveal some of the caveats of the ANN approach.

This is an interesting point.

Based on Fig. A1, the observational data do not show high DMS hot spots/or good coverage in September in the subarctic NE Pacific, instead, there are some high DMS measurements in August in this region. Both Lana et al, 2011 and our ANN model capture the high DMS concentrations in August in this region.

We believe that in September the high DMS concentrations in this region are interpolated from August. On the other hand, our ANN model predicts moderate DMS concentration at this region in September.

In any event, we agree that more discussion is needed here. The following discussion was added to address the methodology difference between objective interpolation method and ANN method (l.353 – l.364).

"L11 stands out in the S. hemisphere monthly mean plot (Fig. 3b), with the highest mean concentrations in January and December, when DMS concentrations are ~2 times higher than other model predictions. Galí et al. (2018) identified five short-comings associated with the direct interpolation method employed by Lana et al. (2011). All shortcomings concern the nature of in situ DMS data, including right-skewed distribution, lack of spatial and temporal coverage, lack of duplicate measure- ments, and sampling bias towards DMS-productive conditions. Because of the sparsity and skewed distribution, the interpo- lation/extrapolation method broadcasts small scale features to large scales (Tesdal et al., 2016). This is especially true for the month of January and December when the elevated L11 monthly means were mainly driven by a small amount of extremely high DMS measurements (>40 nM) near the Antarctic continent. On the other hand, empirical models including the ANN model used in this study rely on environmental parameter climatologies to obtain the DMS climatology. Extreme conditions are smoothed out in climatological data, e.g. in the DMS database the maximum in situ Chl-a concentration is > 800 mg/m3, whereas it is ~50 mg/m3 in the SeaWiFS climatology. When climatological data are used to generate DMS distribution, a smaller variance than in situ data is expected."

In summary, I think that the ANN is an interesting approach that will help improve DMS (and other) climatologies, especially where data are lacking, as it will do better than inter- and extrapolations. However, the present manuscript does not go much beyond the application of the ANN; when it intends to do so, too often it uses the wrong arguments and is not fair with previous studies. It fails to mobilize what we have learned about DMS in the last one or two decades.

In the revised MS, we have added more tests and more discussion accordingly. More specifically, we designed experiments that help to open the network "black box", and based on the results, we thoroughly discussed the parameters that exert an impact of the prediction ability of the model. The text is from line 253-line 303.

---

## Author Comment (AC3) · 7 Jun 2020

Review of Wang et al., Global ocean dimethyl sulfide climatology estimated from observations and an artificial neural network.

This manuscript describes a novel methodology for deriving a global ocean dimethyl sulfide (DMS) climatology, using an artificial neural network (ANN). The authors demonstrate that the ANN is able to explain a greater fraction of variance in the raw available observations of surface ocean DMS concentrations, as compared with a multiple linear regression approach. They also contrast this approach with previous work that used spatial and temporal gap-filling to estimate DMS concentrations, including in data-sparse regions. Instead, the approach presented here derives relationships between observed environmental parameters and observed oceanic DMS DMS concentrations (using the multiple regression or ANN), and uses these to predict/extrapolate DMS concentrations globally.

The paper is clearly written, the methods are straightforward and appropriate, and it represents a valuable contribution to work on understanding and representing the present-day climatological distribution of DMS concentrations in the surface ocean. Improved climatologies of DMS would be useful for Earth System models, especially if they can offer more insights into how the DMS production would change under past/future climate states. It's unclear (to me, at least) whether a machine learning approach will be able to offer such physical insights. Nevertheless, such approaches can offer a better estimate of the present-day state, and this is useful in itself for Earth System modeling. The uncertainty in ocean DMS climatologies is still quite large, despite advances during the past decade, and new advances in statistical approaches that can reduce errors in these datasets are welcome.

Thank you for your positive comments.

I have only a few minor comments, as follows:

I agree with the comments of the two previous reviewers that the arguments made against data binning are weak. The authors imply that it is an inherently inferior approach, but, this is not necessarily true a prior. There can be good arguments in favor of data binning before analysis, e.g., to harmonize the temporal and spatial scales of multiple datasets before analyzing the relationships between them. When in situ DMS measurements (essentially instantaneous) are being predicted via monthly mean values of chl-a, MLD, etc., it is not at all obvious that it is appropriate to perform the analysis without first binning the data. This point should be treated with more nuance, taking into account the details of the datasets and the processes involved.

This point was also raised by the other two reviewers. We therefore dealt with it very carefully, and added the following arguments.

The PMEL database expanded dramatically. Now there are a total of 86,785 valid DMS measurements (concentration greater than 0.1 nM and less than 100 nM according to your instructions), that is 71% larger than the number of data we initially used (51,161). For the

expanded data set, ~93% of DMS are accompanied with in-situ SST measurements, ~81% are accompanied with in-situ salinity measurements. More importantly, each data point has their unique location and sampling time signatures. As shown in the following figure, sampling time (date) and location information is a strong DMS predictor, which together can decrease DMS root mean square error to 0.64 (on natural logarithm scale). Adding other climatological predictors can further improve the model performance.

The NAAMES dataset has 6,786 valid data points, which are ~7% of the total data points (93,571 = 86785+6786). All data are accompanied with in-situ Chl a, SST, and SAL measurements. For parameters without in-situ measurements, high resolution data are used to match DMS measurements, $0.0417° \times 0.0417°$ for PAR, $0.5° \times 0.5°$ for MLD, and $1° \times 1°$ for NO3, which ensures most of DMS have a set of unique predictors. As shown in Table 1, merging NAAMES data with PMEL data does not significantly change the statistic.

Moreover, binning the data will reduce data variance, which has been demonstrated by Derevianko et al. (2009). The objective of this study is to train an ANN with as many data as possible, so that the model is generalized. It not only can apply to coarse resolution predictor fields, but also can apply to very fine resolution field, for example, we have applied the network to fine resolution NAAMES fields for comparison with in-situ DMS measurements (Bell et al., in prep.).

Lastly, binning data will also result in loss of information. A great amount of information is associated with sampling time and date as shown in the following figure (Fig. 2a in MS). By binning the data into monthly $1° \times 1°$ grid, the valid DMS data points will decrease significantly from 82,996 to 9,018; sampling date feature (365) will be average to 12 months, and coordination combinations will be averaged from $87,332 \times 87,332$ to $180° \times 360°$, which represents great information reduction. For ANN models, less data points usually lead to overfitting Fig. 2b.

[Figure]

Fig. 2 Parameter sensitivity tests on raw and binned data. (a) Root mean square error on logarithmic scale for the model trained using raw data; (b) Root mean square error on logarithmic scale for the model trained using binned data. The time and location parameters are tested separately without combining with environmental parameters as shown in the upper panel, (I) with only location parameters; (II) with location and day of year parameters; and (III) with location, day of year, and time of day parameters. The model with three location parameters (I) has a root mean square error on natural logarithmic scale of ~0.83, which decreases to ~0.65 by adding sampling day of year parameters (II), however, increases to ~0.67 by adding sampling time parameters (III). We, therefore, do not include sampling time parameters in the following tests. We tested every possible combination of the eight parameters (PAR, MLD, SST, SAL, Chl a, DIP, DIN, and SiO), which in total are 255 tests.

p. 5, l. 128-130: I was glad to see that the authors have considered the issue of potential overfishing, but they don't explain how they determined that the setup they used for the ANN is not overfitting (i.e., what methods or criteria were used to determine this). It's common to use multiple rounds of cross-validation (such as k-fold crossvalidation or related methods) in order to determine whether a statistical model may be overfitting and to assess the uncertainty in the fit. If I am understanding the description of the method correctly, it seems that while the authors divided the data into training and validation subsets, they did so only once. In this case, the results of the ANN will be sensitive to the specific subset of data that was used for training it. It should be explained how the training/validation subsets were selected, and also whether a multiround cross-validation method was employed (and if not, why not). Or, if appropriate, the authors could simply carry out a more thorough cross-validation and update the manuscript, since I expect this should not require much effort.

Good point.
There are two general guidelines when one separates the data to training and validating sets, representation and generalization. That is to say that your training data has to be representative, and your model has to have the ability to generalize. The online DMS data are organized by contributor ID, while when you do cross validations, the data are drawn section by section as the following figure shows. One section of data may be from a specific contributor who collected data from a specific region, therefore, the data may not be representative, which results in an over-trained or a less-trained model (we have uploaded the cross-validation model to github ((https://github.com/weileiw/ANN-DMS-code), so interested readers can play with it.). To make the selection more representative, a common practice is to shuffle the data, and then randomly draw a fraction from the shuffled data. For DMS data (or maybe other oceanography data too), data collected from the same cruise are highly intercorrelated, so that shuffling and randomly splitting will "leak" information to the model and cause an overfitting (we have tested shuffling and random drawing method, it indeed leads to overfitting. (Code is also available at Github directory.)) .

Another purpose of doing cross-validation is to allow your model to see as many data as possible. This is useful when you do not have enough data to train your network. To achieve a similar effect, we first manually adjust the hyper-parameters (dropout ratio, hidden layers, number of nodes etc., they are key parameters to determine the model performance) using manually-divided training, internal testing, and external validation data. After we get a satisfactory combination of those hyper-parameters, we fix them and fine tune the network using all available data (because the data are intercorrelated, shuffle and randomly split training and testing does the work.).

Lastly, in the parameter selection experiments, we examined a total of 255 models (every combination of eight environmental parameters). We then ranked the model according to root mean squared error (RMSE) on validation data as shown in Fig. 2a. Compared to RMSEs on the training data, there are no apparent overfittings for the top 10 models. The models with larger RMSEs generally overfit the training data. Meanwhile, overfitting occurs with almost every model when binned data are used (Fig.2b).

Accordingly, we have added more explanations in the revised MS as follows (l.154 – l.162):

"The data was split into sets manually rather than automatically. The online DMS data are organized by contributor ID, and automatic splitting draws a continuous portion from the data. The data portion may come from a specific contributor who collected data from a specific region and it may therefore not be representative. This would result in an over-trained or a under-trained model. To make the selected data more representative, a common practice is to shuffle the data, and then randomly draw a fraction from the shuffled data. For DMS, data collected from the same cruise are highly intercorrelated, so that shuffling and randomly splitting "leaks" information to the model and causes overfitting. We manually adjust the hyper-parameters (dropout ratio, hidden layers, number of nodes etc.) using the data that has been manually-divided into training, internal testing, and external validation subsets. After obtaining a satisfactory combination of those hyper-parameters (as discussed below), we fix them and fine tune the network using all available data."

[Figure]

Figure cited from https://scikit-learn.org/stable/modules/cross_validation.html.

p. 5, l. 133-134: It was not obvious to me what the "random states" refer to – is this a random seed controlling initial parameter values?

This is a good point and following is the explanation.
In the ANN, there are at least two places using random states, 1) it uses random state to decide the Dropout nodes, 2) it uses random state to separate internal testing data from training data. The random states do not control initial parameter values, but different random states produce slightly different results. To make our model results reproduceable, we fixed the random state at 64 in the revised model. The uncertainly analyses are now based on different parameter combinations.

p. 8, l. 220: here, it is stated that ANN is able to "capture more of the variance" than "previous extrapolations (Kettle et al., 1999; Lana et al., 2011)". This is a key claim of the paper in terms of the claimed improvement over previous methods, and I can believe this is probably true, but I think the claim ought to be supported by a quantitative value – i.e., the percentage of variance captured by the two previous climatologies – so that readers can compare and see the improvement in this metric. Perhaps these values are in the manuscript somewhere and I overlooked them – in that case I think they should be featured somewhere that is easier to find (e.g., in the abstract or in a table).

Good point.
However, it is hard to do an apple-to-apple comparison. Because we are comparing to raw data, whereas, Kettle et al., 1999 and Lana et al., 2011 interpolated the data, it is hard to extract the raw data information from the climatological map. We thus changed the wording, and weakened the comparison as follow,

"the ability of the ANN to build a nonlinear relationship between DMS and environmental predictors allows it to capture much of the variance"

We also added more comparison to previous model results as shown in Fig. 3. and Fig. 4 in the text, also attached below.

[Figure]

Figure 3. Comparisons of monthly mean DMS concentrations between this study and previous studies (Simó and Dachs, 2002; Vallina and Simó, 2007; Lana et al., 2011; Galí et al., 2018).

[Figure]

Figure 4. Comparisons of zonally mean DMS concentrations between this study and previous studies (Simó and Dachs, 2002; Vallina and Simó, 2007; Lana et al., 2011; Galí et al., 2018).

p. 11: I tested the links for the code and data availability; the data doi link at zenodo works, but the github link does not seem to be available.

The code is previously in a private repository, and now is public (https://github.com/weileiw/ANN-DMS-code). We have also uploaded the corresponding data used to train the model in the following directory: https://zenodo.org/record/3833233#.XsM4cBP0nV4

I also noticed a couple of typos:
p. 2, l. 40: "result" -> "results" or "result[s]"
Corrected, Thank you.
p. 5, l. 31: "deduction" -> "reduction"
Corrected, Thank you.

p. 5, l. 133: "assemble" -> "ensemble" (?)
Corrected, Thank you.
p. 7, l. 189: "wasters" -> "waters"
Corrected, Thank you.

---

## Author Comment (AC4) · 7 Jun 2020

[revised manuscript text omitted]

---

## Referee Report (RR1)

**Second review round of Wang et al. Manuscript submitted to Biogeosciences Discussions**

by Martí Galí

**General comments**
The manuscript has seen some substantial improvements compared to the first version. I appreciated the more comprehensive comparison between DMS fields generated by the ANN and those generated with simpler empirical algorithms, including remote sensing algorithms. The comparison is informative as to the uncertainties in different global DMS products. However, I was disappointed to find that several inconsistencies and inaccuracies remain in the text. These issues have to be addressed and clarified.

In the previous review I prompted the authors to provide a more up-to-date view of DMS(P) cycling processes. However, this did not require adding long discussions, just improving the quality of the information and the accuracy of literature references. Now, in section 3.3, the authors discuss why some environmental variables got selected as ANN predictors in relation to available knowledge. But they do not show any results about the sign of the effects of each predictor variable used in the ANN, or whether the sign changes from one region to another. Thus, the discussion becomes quite pointless (because our understanding does not increase), and could be more succinct. The distinction between "controlling factors" and "predictors" should also be clearer throughout, as this paper does not shed light on the controlling factors.

As in the previous version of the manuscript, the authors make a strong case against the binning of in situ DMS measurements, and present a new figure (Fig. 2) to support their choice. However, I still find that the "binning issue" is treated in an inconsistent manner. For example, despite advocating the importance of using non-binned data, the authors do not use time-resolved satellite Chl as predictor although such data are available for most of the DMS measurements used to train the ANN. Instead, the authors match non-binned DMS data to a heavily time-binned global climatology of satellite Chl from SeaWiFS, which does not even cover the entire ocean color satellite era. Finally, they produce global 1x1 monthly DMS climatologies by applying the ANN, previously trained on non-binned DMS data, to climatological fields of the predictor variables. To be clear, I am not asking the authors to perform strict matchups between DMS and satellite data, which I think is beyond the scope of their study (note, however, that a DMS-satellite match-up database is available for measurements older than 2012: https://doi.org/10.5281/zenodo.2205131). In my view the paper needs clearer and more consistent reasoning and methodology regarding the utility of binned data to produce climatologies, because (1) the whole point of making climatologies is precisely collapsing temporal and spatial variability, and (2) the trained ANN is not implemented to produce non-climatological fields in this study.

In connection with comment above: no quantitative assessment of overfitting is provided in section 3.4 nor figure 2. Just a qualitative assessment based on "how much the training and validation errors overlap". Although the inclusion of figure 2 is welcome, a quantitative assessment would provide stronger support to the authors' choices (ie training the ANN using non-binned data).

**Specific comments**
Line numbers refer to the document bg-2020-72-AC4-supplement.pdf

**Abstract**
The reasoning in the second sentence is backwards, in my opinion. In addition, I think this paper deals with the "predictors", not with the "controlling factors".

**Introduction**

L45: Some phytoplankton do not produce significant amounts of DMS, just its precursor DMSP. Please clarify.

L44-51: This entire paragraph is misleading. Reading it, one may deduce that the inability of biogeochemical models to capture global DMS distributions stems from the excessive simplification of phytoplankton diversity, lumping together very different taxa in a few phytoplankton functional types. Le Clainche et al. (2010) propose otherwise: it is the excessive dependence of DMS on phytoplankton dynamics in sulfur cycling parameterizations what results in poor model skill. And they explicitly suggest that modulation of community DMS production yields by environmental stressors (light, nutrients) must be included in model parameterizations. For example, by Vallina et al. (2008) showed the importance of adding light stress effects on DMS production in the Sargasso Sea using a model that had a single phytoplankton compartment. I refer the authors to my previous review for papers reporting experimental evidence of nutrient and light stress. Finally, note that citation of Le Clainche's paper does not support the statements made in this paragraph.

L57: "Interpolation from neighboring provinces". I suggest adding "weighted" before interpolation. Actually, what Lana et al. (2011) did in provinces with insufficient monthly data substituting their seasonal cycle by that of the "biogeochemically closest" province, weighted by the "local province" average.

L59-69: The message of this paragraph is unclear and the newly added sentences disrupt the flow. In addition, I think the authors should refer to "predictors" instead of "controlling factors". The ANN approach as implemented here does not reveal the controlling factors.

**Methods**
L84-85: With the current writing, it is unclear whether measurements were matched to the 1997-2010 SeaWiFS climatology or to the multiyear time series. In the latter case, the period 2011-present would not be covered by matching satellite data. In the former case, one may wonder if the 1997-2010 climatology is representative of the 2011-present period in all ocean areas. Please clarify and provide a solid argumentation for not matching DMS data to satellite data at the best available resolution.
Note: I found the answer later, in L232. As suspected, DMS data were matched to climatological Chl. Note also that global Chl climatologies from ultiple sensors spanning 1997-2020 are available (e.g. GlobColour, ESA CCI).

L84: What reprocessing? Access date? Same applies to L94.

L114: This contradicts L91, where the authors say that "For consistency, we use only Chl-adata retrieved from SeaWiFS in the following multilinear and network models".
L117: The statement that "there is no available climatological [DMSP] dataset to fill the missing values" is not entirely true. There is no climatological DMSP dataset based on objective interpolation of in situ data. But there is a global sea-surface DMSPt climatology based on the remote sensing algorithm of Galí et al. 2015, available here: https://doi.org/10.5281/zenodo.2558511.

L118 and elsewhere: Please replace SiO by $SiO_4$.

L154-162: It is unclear what the authors did. I could not understand whether data had finally been shuffled or not, and why.

Section 2.4.2: I don't see the point of training the ANN using non-binned data to, afterwards, compute the DMS climatology from 1x1 gridded climatologies of the predictor data. If training the model on non-binned data was so advantageous, I would expect the authors to first compute global gridded DMS time series at the highest possible resolution, and only afterwards collapse the multiyear fields into a climatology. The advantage of training the ANN with non-binned DMS data would be demonstrated if a climatology produced using the minimal possible amount of climatological predictor fields outperformed another one computed directly from climatological predictor fields.

Equation 7: this equation produces negative DMS flux at WS < 1.33. Although such low wind speeds are infrequent in a climatology, did the authors add a correction to avoid potential negative values?

**Results and Discussion**
L224: I suggest replacing "it is relatively easy to parameterize in a biogeochemical model" by something like "it is a priori more amenable to simple parameterizations".

L248: "the model roughly predicts the level of DMS concentrations" sounds a bit vague. Can you please report skill metrics like RMSE or bias in addition to R2?

L257: [the ANN] "also incorporates diurnal and seasonal signals present in the data". Please add something like "but see below", or place this sentence after discussing why time of day does not improve ANN predictions. In fact this sentence is a bit contradictory with the finding described two paragraphs later.

L264: "As shown in Fig. 2… sampling location and date alone can explain 44% of the validation data variance". Figure 2 does not show R2. Please rephrase.

L265: The discussion that starts here is interesting but needs a little revision. The analysis of Vallina & Simó 2007 did not address diel cycles, so I do not agree with this sentence: "Given the strong correlation between solar radiation and265DMS concentration reported by Vallina and Simó (2007), one would expect that adding sampling time would improve the model performance".
The occurrence of predictable diel cycles was assessed by Royer et al. 2015 (Small−scale variability patterns of DMS and phytoplankton in surface waters of the tropical and subtropical Atlantic, Indian, and Pacific Oceans) using continuous underway DMS data collected across the global oligotrophic oceans. They concluded there was no such a universal diel cycle, for the reasons pointed out in the manuscript (different possible outcomes of the balance between DMS sources and sinks over the die cycle).

L281: I do not think the cryoprotectant role has anything to do with global DMS patterns, considering that water temperature in most oceanic regions never decreases to freezing temperatures. Please remove or elaborate a better explanation. In addition, what is the sign of the SST influence on DMS?
On the contrary, I agree with the sentence that concludes the paragraph. SST and MLD have known for decades to define and capture a large deal of the biogeographic patterns of the ocean (e.g., Fay & McKinley, 2014, ESSD).

L290: "rate" should be "rate constant" (= specific rate). Not the same as rate!

L295: "loss" should be "transport" because sometimes turbulent vertical transport can result in net inputs to the upper mixed layer.

L351: Please remove "runoff". I would use expressions like "higher freshwater content" or "salinity stratification". Freshwater may come from ice melt, continental runoff, etc… And in addition much of the stratification in high northern latitudes is due to encounter between fresher Pacific-derived waters and more salty Atlantic waters.

L361: It is risky to use these maxima to illustrate your point. If anything, I would rather use quantiles. Very high Chl from SeaWiFS may result from algorithm artifacts in CDOM-ladden waters, whereas Chl data from PMEL were not quality-controlled and some were collected in estuarine areas (Galí et al., 2015, RSE).

L371-374: In the marginal ice zone, DMS can reach high concentrations during-after ice breakup without any need for sea-ice-algae release. Simply because the biomass of high-DMS(P) phytoplankton (*Phaeocystis*) in the water column can be very high. See the compilation by Levasseur (2013; Nat Geo). Out of the seasonal ice zone, the argument on thhe cryoprotection function does not make sense, since subpolar waters where coccolithophores bloom in booth hemispheres do not reach freezing temperatures, and for example Emiliania huxleyi blooms typically happen late in summer at tempertaures >10C. Please rephrase.

L396: In the text you report a global DMS emission of 15.9 Tg S yr$^{-1}$ using the GM12 parameterization. However, when I look at Fig. 8, I see a mean monthly DMS emission of about 1.5 Tg S yr$^{-1}$ (and, for sure, greater than 1.4). Multiplying 1.5x12, we get an annual emission of 18 Tg S yr$^{-1}$. Therefore, I wonder how do you arrive at the value of 15.9 Tg S yr$^{-1}$.

L430-434: Agree with this point.

**Technical corrections**

L84: Check verb tense in this sentence and concordance with the previous one.
L94: in the paragraph above the resolution was 9.2 km, now it is 93 km. Please check.
L173: coordination or coordinate?
L348: please add "concentrations" after "higher".
L426: principal, not "principle".

---

## Author Response (AR2)

Wang et al. Global ocean dimethyl sulfide climatology estimated from observations and an artificial neural network

Review of revised manuscript by Martí Galí
General comments
The manuscript has seen some substantial improvements compared to the first version. I appreciated the more comprehensive comparison between DMS fields generated by the ANN and those generated with simpler empirical algorithms, including remote sensing algorithms. The comparison is informative as to the uncertainties in different global DMS products.
However, I was disappointed to find that several inconsistencies and inaccuracies remain in the text. These issues have to be addressed and clarified.

We tried out best to make the paper clear and easy to follow. The changes are highlighted in the revised manuscript.

In the previous review I prompted the authors to provide a more up-to-date view of DMS(P) cycling processes. However, this did not require adding long discussions, just improving the quality of the information and the accuracy of literature references. Now, in section 3.3, the authors discuss why some environmental variables got selected as ANN predictors in relation to available knowledge. But they do not show any results about the sign of the effects of each predictor variable used in the ANN, or whether the sign changes from one region to another. Thus, the discussion becomes quite pointless (because our understanding does not increase), and could be more succinct. The distinction between "controlling factors" and "predictors" should also be clearer throughout, as this paper does not shed light on the controlling factors.

We have added sensitivity results to demonstrate how environmental parameters impact the DMS concentration distribution. Meanwhile, we merged part of Sec. 3.3 into Sec. 3.7 Sensitivity test, and shortened the discussion to make it clear and succinct.

[revised manuscript text omitted]

a) $P_{SST} - CTL$

b) $P_{MLD} - CTL$

c) $P_{PAR} - CTL$

d) $P_{SAL} - CTL$

e) $P_{DIP} - CTL$

f) $P_{SiO_4} - CTL$

g) $P_{Chla} - CTL$

*Fig. 9 Differences of annul mean DMS concentration between perturbation models and the control model. Specific figure indexes are listed in the figure, where Pxxx represents a perturbed model and the subscript xxx indicates which parameter is changed. C T L is the control model that is the average of our top 10 model results (Fig. 5)s*

As in the previous version of the manuscript, the authors make a strong case against the binning of in situ DMS measurements, and present a new figure (Fig. 2) to support their choice. However, I still find that the "binning issue" is treated in an inconsistent manner. For example, despite advocating the importance of using non-binned data, the authors do not use time-resolved satellite Chl as predictor although such data are available for most of the DMS measurements used to train the ANN. Instead, the authors match non-binned DMS data to a heavily time-binned global climatology of satellite Chl from SeaWiFS, which does not even cover the entire ocean color satellite era. Finally, they produce global 1x1 monthly DMS climatologies by applying the ANN, previously trained on non-binned DMS data, to climatological fields of the predictor variables. To be clear, I am not asking the authors to perform strict matchups between DMS and satellite data, which I think is beyond the scope of their study (note, however, that a DMS-satellite match-up database is available for measurements older than 2012: https://doi.org/10.5281/zenodo.2205131). In my view the paper needs clearer and more consistent reasoning and methodology regarding the utility of binned data to produce climatologies, because (1) the whole point of making climatologies is precisely collapsing temporal and spatial variability, and (2) the trained ANN is not implemented to produce non-climatological fields in this study.

As we explained in our response to the first reviews, binning the data will reduce data variance. This point has also been demonstrated by Derevianko et al. (2009). For the DMS dataset, sampling location and time are important predictors (Section 3.2 and 3.3). Binning and averaging the data lead to loss of these valuable information. Specific to the ANN method, binning the data leads to overfitting, which is shown in Fig. 2.

We argue that there are differences between how a model is trained and how a model is applied. When you train a model, it is important to make the model be able to generalize. That way, you can apply the model to either a coarse or a fine resolution filed. For example, we trained the network model using the raw data and later we are able to apply the model to the very fine resolution NAAMES fields ($0.0417° \times 0.0417°$) (Bell et al. in preparation). We also published our models and code, therefore, any interested reader can reproduce the results and can also apply the model to their own data, regardless of data resolution.

Regarding Chl a, we compared PMEL in situ Chl a to the satellite derived Chl a. Chl a data from these two sources are well correlated ($R^2 = 0.64$ on logarithm scale). Chl a is not a strong DMS predictor. The ANN model does a fairly good job even without it. Moreover, we do not have time series MLD and nutrient data to strictly match DMS measurements. To match DMS with only Chl a would not be meaningful, because the other parameters are all climatological.

In connection with comment above: no quantitative assessment of overfitting is provided in section 3.4 nor figure 2. Just a qualitative assessment based on "how much the training and validation errors overlap". Although the inclusion of figure 2 is welcome, a quantitative assessment would provide stronger support to the authors' choices (ie training the ANN using non-binned data).

*The quantitative assessments have been added.*
*l. 301-302:*
*For example, the averaged RMSE on natural logarithm scale for the 10 best ANN models is 0.608 for the validating dataset and 0.600 for the training dataset when using the un-binned data, whereas the RMSE is 0.655 (validating) and 0.635 (training) for the model constructed using the binned data (See Fig. 2b).*

Specific comments
Line numbers refer to the document bg-2020-72-AC4-supplement.pdf
Abstract
The reasoning in the second sentence is backwards, in my opinion. In addition, I think this paper deals with the "predictors", not with the "controlling factors".

*The sentence is revised to,*

*Knowledge of the global-scale distribution, seasonal variability, and sea-to-air flux of DMS is needed in order to improve understanding of atmospheric sulfur, aerosol/cloud dynamics and albedo.*

*We have changed "controlling factors" to "predictors".*

Introduction
L45: Some phytoplankton do not produce significant amounts of DMS, just its precursor DMSP. Please clarify.

*Corrected. The new sentence is as follows,*
*l.47-48*
*In these models, phytoplankton are divided into different groups based on their ability to produce DMSP, the precursor of DMS.*

L44-51: This entire paragraph is misleading. Reading it, one may deduce that the inability of biogeochemical models to capture global DMS distributions stems from the excessive simplification of phytoplankton diversity, lumping together very different taxa in a few phytoplankton functional types. Le Clainche et al. (2010) propose otherwise: it is the excessive dependence of DMS on phytoplankton dynamics in sulfur cycling parameterizations what results in poor model skill. And they explicitly suggest that modulation of community DMS production yields by environmental stressors (light, nutrients) must be included in model parameterizations. For example, by Vallina et al. (2008) showed the importance of adding light stress effects on DMS production in the Sargasso Sea using a model that had a single phytoplankton compartment. I refer the authors to my previous review for papers reporting experimental evidence of nutrient

and light stress. Finally, note that citation of Le Clainche's paper does not support the statements made in this paragraph.

We have rephrased the last two sentences to make our statement clearer.
l.53-55
*Despite this level of modeling detail, there are still large discrepancies between the model simulations and in situ measurements (Tesdal et al., 2016). Le Clainche et al. (2010) suggested that environmental conditions should be included in future model development because DMS cycling depends strongly on phytoplankton dynamics.*

L57: "Interpolation from neighboring provinces". I suggest adding "weighted" before interpolation. Actually, what Lana et al. (2011) did in provinces with insufficient monthly data substituting their seasonal cycle by that of the "biogeochemically closest" province, weighted by the "local province" average.
Corrected. Thanks.

L59-69: The message of this paragraph is unclear and the newly added sentences disrupt the flow. In addition, I think the authors should refer to "predictors" instead of "controlling factors". The ANN approach as implemented here does not reveal the controlling factors.
We have rephrased the paragraph for clarity. The revised section is as follows,
l. 69-74
*In this study, we explore the relationships between DMS and environmental parameters using a machine learning method. Such relationships are hard to detect using traditional linear regression methods, because environmental parameters do not directly influence DMS concentration. They control the distribution of marine algae that determines the distribution of DMSP (a precursor of DMS) and its conversion to DMS (Kiene et al., 2000; Simó, 2001). The objective of this paper is to discover the relationships between DMS and environmental variables, with the goal of constructing a novel monthly-resolved DMS climatology.*

We have added sensitivity tests that reveals how each environmental parameter can influence DMS distribution, and thus we believe the use of controlling factors are appropriate.

Methods
L84-85: With the current writing, it is unclear whether measurements were matched to the 1997- 2010 SeaWiFS climatology or to the multiyear time series. In the latter case, the period 2011- present would not be covered by matching satellite data. In the former case, one may wonder if the 1997-2010 climatology is representative of the 2011-present period in all ocean areas. Please clarify and provide a solid argumentation for not matching DMS data to satellite data at the best available resolution.
Note: I found the answer later, in L232. As suspected, DMS data were matched to climatological Chl. Note also that global Chl climatologies from multiple sensors spanning 1997-2020 are available (e.g. GlobColour, ESA CCI).

We have made the Chl *a* data source clear in Sec. 2.1. The sentence is in line 89-90.

L84: What reprocessing? Access date? Same applies to L94.
Information added. Thank you.

L114: This contradicts L91, where the authors say that "For consistency, we use only Chl-a data retrieved from SeaWiFS in the following multilinear and network models".

There is no contradiction.

L114 describes how we run linear regression model. In the linear regression model, we use both in situ and satellite Chl a data.
To make it clear, we have added the following text in the manuscript.

l.118-119
*In a third step, to keep Chl a data sources consistent as described previously, we use satellite Chl a;*

L117: The statement that "there is no available climatological [DMSP] dataset to fill the missing values" is not entirely true. There is no climatological DMSP dataset based on objective interpolation of in situ data. But there is a global sea-surface DMSPt climatology based on the remote sensing algorithm of Galí et al. 2015, available here: https://doi.org/10.5281/zenodo.2558511.
Thank you for pointing this out. We have rephrased the sentence.

l. 122-123
*DMSPt is not included, because there is no observation based climatological dataset to fill the missing values.*

The DMSP database that the reviewer referred to is a valuable resource, but we do not intend to include that in our model, so that we avoid building our model based on another model's results.

L118 and elsewhere: Please replace SiO by $SiO_4$.
Thank you, SiO has been replaced with $SiO_4$.

L154-162: It is unclear what the authors did. I could not understand whether data had finally been shuffled or not, and why.
We have rephrased the corresponding sentence to make it clear as follows,

l.160-163
*The data was split into the above sets manually rather than automatically. This is because data collected from the same cruise are highly intercorrelated. The common practice of shuffling and randomly splitting the data produces an over-fitted model because the validating data can be predicted using near-neighbor values. This kind of apparent skill does not generalize to regions with large data gaps, which we need for constructing a robust climatology.*

Section 2.4.2: I don't see the point of training the ANN using non-binned data to, afterwards, compute the DMS climatology from 1x1 gridded climatologies of the predictor data. If training the model on non-binned data was so advantageous, I would expect the authors to first compute global gridded DMS time series at the highest possible resolution, and only afterwards collapse the multiyear fields into a climatology. The advantage of training the ANN with non-binned DMS data would be demonstrated if a climatology produced using the minimal possible amount of climatological predictor fields outperformed another one computed directly from climatological predictor fields.
Please see our reply to general comment #2;

Equation 7: this equation produces negative DMS flux at WS < 1.33. Although such low wind speeds are infrequent in a climatology, did the authors add a correction to avoid potential negative values?
Yes, negative $K_{w,660}$ was set to zero in our algorithm. We have made it clear in the text.

Results and Discussion
L224: I suggest replacing "it is relatively easy to parameterize in a biogeochemical model" by something like "it is a priori more amenable to simple parameterizations".
The sentence has been modified. Thanks.

L248: "the model roughly predicts the level of DMS concentrations" sounds a bit vague. Can you please report skill metrics like RMSE or bias in addition to R2?
$R^2$ are added. Thanks.

L257: [the ANN] "also incorporates diurnal and seasonal signals present in the data". Please add something like "but see below", or place this sentence after discussing why time of day does not improve ANN predictions. In fact this sentence is a bit contradictory with the finding described two paragraphs later.
We changed the wording as follows,

l.257-258
*The ANN model can also incorporate sampling time and coordinate signals present in the data (see below).*

L264: "As shown in Fig. 2... sampling location and date alone can explain 44% of the validation data variance". Figure 2 does not show R2. Please rephrase.
We have removed "As shown in Fig. 2,", because we realized that adding $R^2$ values in Fig. 2 is too messy because there are too many $R^2$ values.

L265: The discussion that starts here is interesting but needs a little revision. The analysis of Vallina & Simó 2007 did not address diel cycles, so I do not agree with this sentence: "Given the strong correlation between solar radiation and DMS concentration reported by Vallina and Simó (2007), one would expect that adding sampling time would improve the model performance".

The occurrence of predictable diel cycles was assessed by Royer et al. 2015 (Small–scale variabilityscale variability patterns of DMS and phytoplankton in surface waters of the tropical and subtropical Atlantic, Indian, and Pacific Oceans) using continuous underway DMS data collected across the global oligotrophic oceans. They concluded there was no such a universal diel cycle, for the reasons pointed out in the manuscript (different possible outcomes of the balance between DMS sources and sinks over the die cycle).

We agree that Vallina and Simó 2007 is not appropriate to support DMS diurnal cycle, we thus have deleted the reference and change the wording. The revised text is as follows,

l.266-167
*Time of day can be another possible predictor if DMS concentration varies diurnally. However, adding time of day to the model increases RMSE slightly (Fig. 2a).*

L281: I do not think the cryoprotectant role has anything to do with global DMS patterns, considering that water temperature in most oceanic regions never decreases to freezing temperatures. Please remove or elaborate a better explanation. In addition, what is the sign of the SST influence on DMS?

We have removed the corresponding sentence.

We have added sensitivity tests, DMS and SST do not change unidirectionally (please see our earlier reply to the general comments).

On the contrary, I agree with the sentence that concludes the paragraph. SST and MLD have known for decades to define and capture a large deal of the biogeographic patterns of the ocean (e.g., Fay & McKinley, 2014, ESSD).

Thanks.

L290: "rate" should be "rate constant" (= specific rate). Not the same as rate!

Done, thank you.

L295: "loss" should be "transport" because sometimes turbulent vertical transport can result in net inputs to the upper mixed layer.

Done, Thanks.

L351: Please remove "runoff". I would use expressions like "higher freshwater content" or "salinity stratification". Freshwater may come from ice melt, continental runoff, etc...
And in addition, much of the stratification in high northern latitudes is due to encounter between fresher Pacific-derived waters and more salty Atlantic waters.

Done, Thanks.

L361: It is risky to use these maxima to illustrate your point. If anything, I would rather use quantiles. Very high Chl from SeaWiFS may result from algorithm artifacts in

CDOM-ladden waters, whereas Chl data from PMEL were not quality-controlled and some were collected in estuarine areas (Galí et al., 2015, RSE).

In the manuscript, we stated that in situ Chl a has greater variance compared to satellite Chl a, whereas, the reviewer indicated the opposite. In addition, coastal data are filtered out according to the reviewer's suggestion in the first round of revision.

We anyway changed the expression and used percentile to demonstrate our idea.

l. 325-327
*Extreme conditions are smoothed out in climatological data, e.g. in the DMS database the 99 percentiles of in situ Chl-a concentration is 12.58 mg/m$^3$, whereas it is only 6.85 mg/m$^3$ in the SeaWiFS climatology.*

L371-374: In the marginal ice zone, DMS can reach high concentrations during-after ice breakup without any need for sea-ice-algae release. Simply because the biomass of high-DMS(P) phytoplankton (Phaeocystis) in the water column can be very high. See the compilation by Levasseur (2013; Nat Geo). Out of the seasonal ice zone, the argument on the cryoprotection function does not make sense, since subpolar waters where coccolithophores bloom in both hemispheres do not reach freezing temperatures, and for example Emiliania huxleyi blooms typically happen late in summer at tempertaures >10C. Please rephrase.

We have changed the wording to emphasize on the main cause of high DMS in high latitude. We have also put cryoprotection and release of ice algae in the context with ice edge zone.
The revised text is as follows.

l.338-344
*The high DMS concentration during the summertime at high latitudes is believed to accompany blooms of coccolithophores and Phaeocystis, which are strong DMSP producers (Neukermans et al., 2018; Wang et al., 2015). The shoaling mixed layer depth during the summer provides favorable conditions, i.e. stable and warm, with adequate irradiation for coccolithophores and Phaeocystis growth (Galí et al., 2019). Additionally, high DMS concentrations at ice edge zones have also been observed. These high concentrations are due to the release of ice algae that are prolific DMSP producers (Stefels et al., 2012; Webb et al., 2019). As an important cryoprotectant and osmolyte, DMSP helps ice algae to cope with the low temperature and high salinity conditions (Thomas and Dieckmann, 2002).*

L396: In the text you report a global DMS emission of 15.9 Tg S yr-1using the GM12 parameterization. However, when I look at Fig. 8, I see a mean monthly DMS emission of about 1.5 Tg S yr-1 (and, for sure, greater than 1.4). Multiplying 1.5x12, we get an annual emission of 18 Tg S yr-1. Therefore, I wonder how do you arrive at the value of 15.9 Tg S yr$^{-1}$.
Thank you for pointing this out. We accidently plotted the flux based on N00, whereas used GM12 in the caption. We have updated the figure to make the figure and caption consistent.

L430-434: Agree with this point.
Thanks.

Technical corrections
L84: Check verb tense in this sentence and concordance with the previous one.
Done, are --> were

L94: in the paragraph above the resolution was 9.2 km, now it is 93 km. Please check.
We have double checked the resolution; it is 9.2 km. Thank you for catching it.

L173: coordination or coordinate?
Corrected, thank you.

L348: please add "concentrations" after "higher".
The word has been added.

L426: principal, not "principle".
Corrected. Thanks.

---

## Author Response (AR3)

Dear editor,

We would like to thank you and the reviewers for your valuable and constructive suggestions, which help to substantially improve the quality of the study. We have addressed the following editorial comments in the revised manuscript.

If you have any further suggestions/comments, please kindly let us know.

Sincerely,
Wei-Lei Wang

Assistant Project Scientist
Earth System Sciences
University of California, Irvine
Irvine, CA 92697
cell: 631.988.2397
weilei.wang@gmail.com
weileiw@uci.edu

L14: Remove "(Lana et al., 2011)" from the abstract.
Done. Thank you.

L24: Romove the parentheses for e.g. Stefels et al., 2007, that is, not (DMSP; (Stefels et al, 2007)), but (DMSP; Stefels., 2007).
Done. Thank you.

L28 chlorophyll a (Chl a; a proxy for phytoplankton biomass). The "a" should be italic.
Done. Thank you.

L 39 and hereafter: "Chl-a" is sometimes used, instead of "Chl a". Please amend it throughout the manuscript including the figure 6 legend.
We think that the rule of using/not using the hyphen between Chl and *a* is that if Chl *a* appears before a noun then a hyphen is used (e.g., Chl-*a* concentration), otherwise, a hyphen is not used.

If the journal has a different rule, we would like to follow it. Please kindly let us know.

L.95: fluorometric method
Done. Thank you.

L111: not mg/m3, but mg m-3;
Done. Thank you.

L160: The data were
Done. Thank you.

L370 and hereafter: N. hemisphere? In the text, northern (or southern) is not abbreviated. Please check these throughout the text.
To keep them consistent, we have spelled out all such abbreviations.

L399: Revolve a space immediately after "ocean".
The space has been removed. Thank you.

L476-606: For the abbreviation of journal names, please follow the Caltech Library as mentioned in the BG manuscript preparation.
Thank you. We have updated the abbreviations for journal names according to Caltech Library.

Table A1: Please amend the format of References: for example, not "(Kettle et al., 1999)", but "Kettle et al. (1999)".
Done. Thank you.